# Accelerated functional brain aging in pre-clinical familial Alzheimer's disease

Julie Gonneaud [1,2 ✉], Alex T. Baria[1], Alexa Pichet Binette [1,2], Brian A. Gordon [3], Jasmeer P. Chhatwal [4], Carlos Cruchaga [3], Mathias Jucker [5], Johannes Levin [6], Stephen Salloway [7], Martin Farlow[8], Serge Gauthier[1], Tammie L. S. Benzinger [3], John C. Morris[3], Randall J. Bateman [3], John C. S. Breitner[1], Judes Poirier [1], Etienne Vachon-Presseau [9,10,11,105], Sylvia Villeneuve [1,2,105 ✉], Alzheimer's Disease Neuroimaging Initiative (ADNI)*, Dominantly Inherited Alzheimer Network (DIAN) Study Group* & Pre-symptomatic Evaluation of Experimental or Novel Treatments for Alzheimer's Disease (PREVENT-AD) Research Group*

Resting state functional connectivity (rs-fMRI) is impaired early in persons who subsequently develop Alzheimer's disease (AD) dementia. This impairment may be leveraged to aid investigation of the pre-clinical phase of AD. We developed a model that predicts brain age from resting state (rs)-fMRI data, and assessed whether genetic determinants of AD, as well as beta-amyloid (Aβ) pathology, can accelerate brain aging. Using data from 1340 cognitively unimpaired participants between 18–94 years of age from multiple sites, we showed that topological properties of graphs constructed from rs-fMRI can predict chronological age across the lifespan. Application of our predictive model to the context of pre-clinical AD revealed that the pre-symptomatic phase of autosomal dominant AD includes acceleration of functional brain aging. This association was stronger in individuals having significant Aβ pathology.

[1] Douglas Mental Health University Institute, McGill University, Montreal, QC, Canada. [2] McConnell Brain Imaging Center, Montreal Neurological Institute, McGill University, Montreal, QC, Canada. [3] Knight Alzheimer Disease Research Center, Washington University School of Medicine, St. Louis, MO, USA. [4] Brigham and Women's Hospital–Massachusetts General Hospital, Boston, MA, USA. [5] Hertie-Institute for Clinical Brain Research, University of Tübingen, Tübingen, Germany. [6] Ludwig-Maximilians-Universität München, German Center for Neurodegenerative Diseases and Munich Cluster for Systems Neurology (SyNergy), Munich, Germany. [7] Butler Hospital, Providence, RI, USA. [8] Department of Neurology, Indiana University School of Medicine, Indianapolis, IN, USA. [9] Department of Anesthesia, Faculty of Medicine, McGill University, Montreal, QC, Canada. [10] Faculty of Dentistry, McGill University, Montreal, QC, Canada. [11] Alan Edwards Centre for Research on Pain, McGill University, Montreal, QC, Canada. [105]These authors contributed equally: Etienne Vachon-Presseau, Sylvia Villeneuve. *Lists of authors and their affiliations appear at the end of the paper. ✉email: julie.gonneaud@mail.mcgill.ca; sylvia.villeneuve@mcgill.ca

The brain shows major changes over the course of aging. It is not fully understood how neurodegenerative diseases affect brain regions and networks that are also affected by normal aging. However, increasing evidence suggests that neural systems vulnerable to age are also vulnerable to Alzheimer's disease (AD) and other neurodegenerative diseases[1]. Recent availability of large-scale neuroimaging datasets has facilitated the application of machine learning techniques and enabled development of models that can predict behavior and characteristics of brain structure and function known to change with age[2–10]. We investigated whether predicted brain age may be a relevant biomarker of neurodegenerative disease[2], inasmuch as disease may cause deviations from normal aging trajectories, and the factors that influence these deviations may be studied. As an example, brain age predictive models using data from structural magnetic resonance imaging (MRI) have shown accelerated biological aging in individuals who develop AD dementia[11–14]. Similar phenomena are already apparent in others who have mild cognitive impairment (MCI) that progresses to dementia[12,15]. Such inter-individual differences between predicted biological and chronological age have been studied in relation to lifestyle variables[16–19] and to genetic determinants[14,17,20]. It is currently unknown, however, whether accelerated brain aging precedes evidence of cognitive decline, and whether it can be detected in the pre-clinical phase of AD.

The dementia of AD is characterized by progressive cognitive decline that becomes sufficient to impair activities of daily living[21]. Prior work has shown that brain changes characteristic of an AD process can be demonstrated two or three decades before symptom onset[22,23]. Typically, this sequence begins with the accumulation of cerebral beta-amyloid (Aβ), followed by the deposits of hyperphosphorylated *tau* (neurofibrillary tangles), metabolic brain alterations, and other evidence of neurodegeneration that precede cognitive and functional symptoms[22,24]. Functional brain alterations revealed by MRI measures of resting state connectivity (rs-fMRI) become detectable almost synchronously with Aβ and *tau* measured by positron emission tomography (PET) and are therefore evident several years before atrophy can be detected by structural MRI[25,26]. Conjunction of such functional and biological changes appears to extend throughout the development of AD from its pre-clinical to its dementia stages[24]. These findings suggest that MRI measures of resting state functional connectivity may be a more sensitive modality than structural imaging for detection of brain changes in pre-clinical AD.

AD dementia symptoms appear only after massive, evidently irreversible brain changes. Therefore, a more promising approach, at least in theory, is to prevent such changes. However, AD prevention requires improved understanding of the pre-clinical phase of AD[27]. Identification of individuals in this clinically silent phase of the disease is challenging because it is mostly unknown who will develop dementia during the lifespan. One way to circumvent this problem is the study of autosomal dominant AD (ADAD), a group of rare genetically determined variants of AD caused by mutations in the amyloid precursor protein (*APP*), presenilin 1 (*PSEN1*) or presenilin 2 (*PSEN2*) genes, all involved in Aβ production[22,28]. Because these mutations are fully penetrant, progression to disease is predictable, making ADAD an ideal model for the study of the pre-clinical (*i.e.*, pre-symptomatic) phase of AD.

Although it is impossible to determine with certainty who will develop dementia due to sporadic AD (sAD), some factors are known to increase the risk of its development. Prominent among these is the ε4 allele at the polymorphic *APOE* locus that encodes apolipoprotein E, known to be involved in Aβ clearance[28,29]. More broadly, a strong family history of sAD dementia has also been associated with a 2- to 4-fold increased incidence of dementia[30,31]. Individuals whose brains show Aβ pathology are also known to experience brain changes and related cognitive decline over time[32,33]. Thus, asymptomatic individuals can be classified as being in the pre-clinical phase of the disease if they have Aβ pathology[27]. Likewise, their risk of dementia is increased if they carry an *APOE* ε4 allele or other known genetic risk factor and/or if they have a strong family history of the disease[34,35]. Here, we tested whether individuals in the pre-clinical phase of ADAD, or at risk of pre-clinical sAD, show evidence of accelerated brain aging prior to the symptoms predicted by their genetic risk or Aβ status.

We studied 1624 cognitively unimpaired participants between 18 and 94 years of age, recruited and scanned in different studies and centers. Within these, we developed a method that predicts brain age from rs-fMRI. We relied on measures of network integration and segregation, known as graph metrics[36], to represent global brain functioning and developed a neural net. Briefly, we trained this model initially on a cohort of cognitively unimpaired individuals ranging in age from 18 to 90 years old. We then validated its generalizability in another group of cognitively unimpaired individuals (in age from 19 to 79 years old) from another study/site. After such validation, we tested whether individuals with pre-clinical ADAD showed accelerated functional brain aging in comparison with their age-matched relatives without a causal mutation. Importantly, none of these latter participants had been involved in the development or validation of the brain age model. In these same individuals, we also tested whether Aβ pathology was a further predictor of brain age. Finally, in a cohort of asymptomatic individuals having a parental or other strong family history of sAD, we tested whether *APOE* ε4 and/or Aβ pathology were associated with predicted functional brain age.

Our results showed, first, that pre-symptomatic carriers of ADAD mutations (DIAN cohort) had evidence of accelerated functional brain aging. Importantly, this finding was stronger in individuals who already accumulated significant Aβ pathology as evidenced by PET imaging. In the cohort at elevated risk of sAD (PREVENT-AD cohort), neither *APOE* ε4 status nor PET evidence of Aβ pathology was associated with apparent accelerated brain aging but individuals closer to their parental age of onset tended to show accelerated brain aging. Secondary analyses in a third independent cohort including a small subset of individuals diagnosed with either sAD dementia or MCI (ADNI cohort) confirmed the expected acceleration in functional brain aging in patients vs. cognitively normal older adults, suggesting that functional brain age is accelerated in cognitively impaired individuals with sAD and therefore validating the sensitivity of our model to sporadic AD-related processes. We conclude that asymptomatic persons with strong genetic determinants show a characteristic pattern of functional brain changes that are associated with accelerated biological brain aging. The biological development of AD is therefore characterized by a pattern of advanced brain aging that can be detected prior to symptom onset, at least in individuals having rare genetic mutations that cause AD and significant Aβ pathology.

## Results

**Separation of the multisite data into a training, validation, and test sets.** We gathered rs-fMRI data from 1624 cognitively unimpaired participants between 18 and 94 years of age, provided by the Dominantly Inherited Alzheimer Network (DIAN), the Pre-symptomatic Evaluation of Experimental or Novel Treatments for Alzheimer's Disease cohort (PREVENT-AD), the Cambridge Centre for Ageing and Neuroscience (CamCAN), the

**Table 1 Dataset characteristics.**

| Cohorts | | Training set | Validation set | Test set |
|---|---|---|---|---|
| DIAN non-carriers | N | 105 | — | 29 |
| | Age [range] | 38.70 ± 11.41 [19–69] | | 38.90 ± 11.55 [20–58] |
| | Sex ratio F/M [F] | 57/48 [54%] | | 18/11 [62%] |
| DIAN carriers | N | — | — | 125 |
| | Age [range] | | | 34.33 ± 9.66 [18–61] |
| | Sex ratio F/M [F] | | | 68/57 [54%] |
| PREVENT-AD | N | 36 | — | 256 |
| | Age [range] | 63.5 ± 5.08 [55–78] | | 63.51 ± 5.37 [55–84] |
| | Sex ratio F/M [F] | 25/11 [69%] | | 189/67 [74%] |
| CamCAN | N | 408 | — | 96 |
| | Age [range] | 51.12 + 18.27 [18–87] | | 55.80 ± 19.30 [18–88] |
| | Sex ratio F/M [F] | 208/200 [51%] | | 40/56 [42%] |
| ADNI | N | 29 | — | 15 |
| | Age [range] | 76.41 ± 6.60 [66–94] | | 72.73 ± 6.70 [65–90] |
| | Sex ratio F/M [F] | 17/12 [59%] | | 10/5 [67%] |
| FCP-Cambridge | N | 195 | — | — |
| | Age [range] | 21.04 ± 2.33 [18–30] | | |
| | Sex ratio F/M [F] | 122/73 [63%] | | |
| ICBM | N | — | 46 | — |
| | Age [range] | | 42.28 ± 19.31 [19–79] | |
| | Sex ratio F/M [F] | | 29/17 [63] | |
| Total sample | N | 773 | 46 | 521 |
| | Age [range] | 43.37 ± 20.45 [18–94] | 42.28 ± 19.31 [19–79] | 54.49 ± 16.25 [18–90] |
| | Sex ratio F/M [F] | 429/344 [55%] | 29/17 [63%] | 325/196 [62%] |

Demographic information by cohort and set.
F Female, M male, DIAN Dominantly Inherited Alzheimer Network, PREVENT-AD Pre-symptomatic Evaluation of Experimental or Novel Treatments for Alzheimer's Disease cohort, CamCAN Cambridge Centre for Ageing and Neuroscience, FCP-Cambridge 1000-Functional Connectomes Project—Cambridge site, ADNI Alzheimer's Disease Neuroimaging Initiative, ICBM the International Consortium for Brain Mapping.

1000-Functional Connectomes Project—Cambridge site (FCP-Cambridge), the Alzheimer's Disease Neuroimaging Initiative (ADNI) and the International Consortium for Brain Mapping (ICBM) cohorts, to build a "brain age" predictive model (Table 1). Considering our focus on the pre-clinical phase of AD, individuals with mild cognitive impairment (MCI) or AD dementia were excluded from the main analyses. In secondary analyses, we nevertheless tested whether cognitively impaired individuals with sAD evidenced accelerated brain aging using our functional predictive model that was built solely on cognitively unimpaired individuals.

After processing and quality control, 1340 cognitively unimpaired individuals remained for the analyses. These were divided into a training set of 773 persons (large multi-cohorts dataset covering the lifespan used to build the predictive models), a validation set (independent lifespan dataset of 46 persons from ICBM used to test the generalizability of the developed models and select the final model), and one multi-cohort test set (125 DIAN mutation carriers and 29 without a mutation, 256 PREVENT-AD individuals thought to be at enhanced genetic risk of sAD, 96 from CamCAN, and 15 cognitively normal individuals from ADNI). A harmonized pre-processing pipeline was applied to all individuals, and 26 graph metrics were chosen based on their ability to quantify whole-brain connectivity and extracted from each participant's correlation matrix (see Material and Methods for details). Further details are shown in Table 1 and Fig. 1.

**Feature ranking as a step for reducing the number of features in the final model.** First, to reduce the number of inputs of the model, we searched for graph metrics that most reliably predicted chronological age[10]. To do so, training set data was entered in parallel in support vector machine (SVM) and regression tree ensemble models to identify graph metrics with highest weights.

The root mean squared error (rmse) for predicted chronological age in SVM and the tree ensemble were 16.45 and 16.08, respectively. Graph metrics were then ranked separately by order of SVM weights and ensemble model importance (i.e., highest load corresponding to the most important). We used the average rank from both models to determine the overall importance of each metric, as presented in Fig. 2a. Feature rank determined which metrics would be used as input into the subsequent neural network models, to build our predictive brain age model.

**Building the Brain Age model and improving its generalizability.** We chose the optimal neural net architecture after having built different neural networks with increasing complexity, varying in number of input features (5, 10, 15, 20, or 25 most-important graph metrics, ranked as described previously), hidden layers, and hidden layer units. Importantly, each graph metric was only entered once as input for each neural network architecture tested, and the inputs were kept constant across the model's iterations, such that features of more complex models always included the features of the simpler ones. We used an average of three determinations of rmse to assess the performance for each model. The different neural networks were applied separately in the training and validation set (Fig. 2b). To test the relevance of the metrics' ranking, we assessed also the performance of neural nets on the training data when including the metrics randomly (null model, see Fig. 2b, right panel) and compared it to the models created based on ranked metrics (Fig. 2b, left panel). This null model suggested that the neural network performed better when features were ordered using SVM weights and ensemble feature importance, at least in simpler models.

To select the optimal neural network architecture for our brain age model, we generated the different models using the training set and evaluated which of these provided the best generalizability

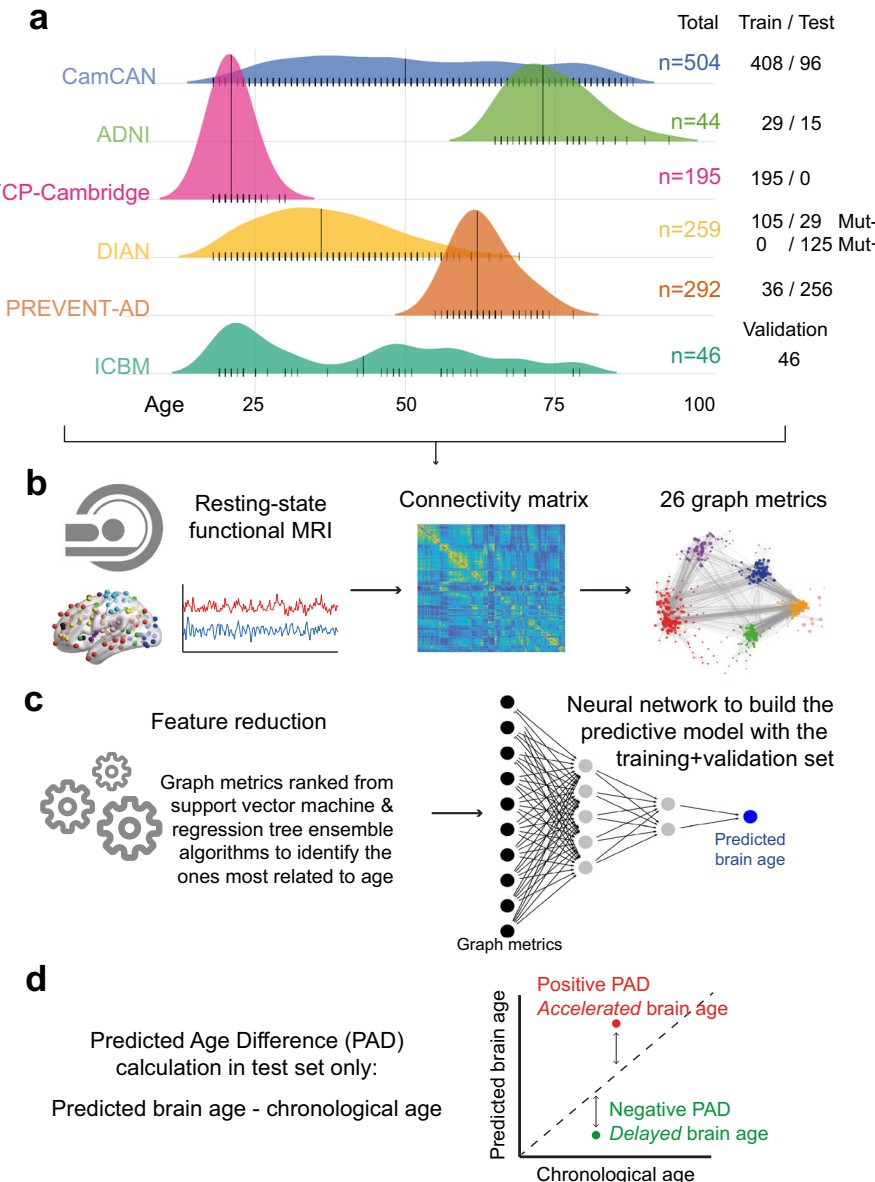

**Fig. 1 Methodology overview. a** Multiple cohorts covering the lifespan were included in the study. They were separated into a training and validation set, both used to develop the predictive brain age model, and a test set in which our model was applied. **b** All participants underwent resting state functional magnetic resonance imaging that was processed with a uniform pipeline. Functional connectivity matrices were generated from the Power atlas[82], from which graph metrics were calculated. Graph metrics were the input in our brain age model, and thus all possible metrics were of interest. **c** The first step toward building the model was to rank the different graph metrics from the most to least related to aging in our training set, to determine an order of importance to our model inputs using both support vector machine and regression tree ensemble algorithms. Neural networks were then tested to identify the best brain age model. Different architectures were tested, and the model applied in the training set that best generalized to the validation set was chosen as the final model (see Fig. 2). **d** The model was applied to the left-out test set and our measure of interest was the predicted age difference (PAD). Mut−: mutation non-carriers, Mut+: mutation carriers, MRI: magnetic resonance imaging, PAD: predicted age difference.

to the validation set (i.e., avoiding overfitting). Thus, the validation set helped us to determine the best balance between improved age prediction and good generalizability. In general, increasing model complexity (more features and hidden layers/units) led to better performance in the training set (Fig. 2b, left panel). However, as expected, too much complexity resulted in overfitting as evidenced by improved performance in the training set, resulting in poorer fit in the validation set (Fig. 2b, middle panel). The model that produced the lowest rmse in the validation set (averaged rmse over 3 iterations = 13.89) had 10 inputs (i.e., the 10 most important metrics, see Fig. 2a) and 2 hidden layers (5 units in the first layer and 2 in the second). The performance of

this specific model was similar to that obtained in the training set (averaged rmse over 3 iterations = 13.75). This model was thus applied to the remaining unseen data (test set) to test whether genetics or AD pathology accelerated apparent functional brain aging. Subgraph centrality, clustering/modularity coefficients, and small-worldness were among the selected graph metrics (10 first metrics Fig. 2a). For reference, the covariance matrix of the 10 selected graph metrics is presented as Supplementary material (Supplementary Fig. 1).

**Performance of the final brain age model.** We show the association between chronological age and the model-predicted brain

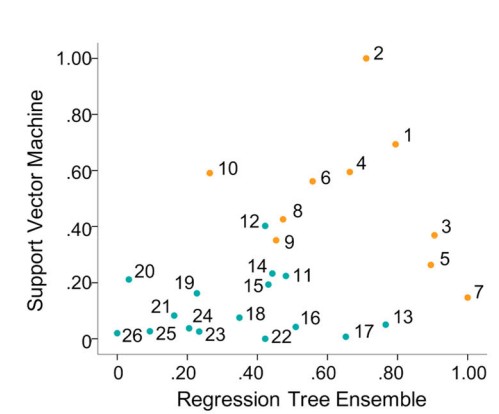

1. Sub-graph centrality
2. Positive-weighted clustering coefficient
3. Binarized assortativity
4. Weighted modularity coefficient
5. Small worldness
6. Flow coefficient
7. Betweenness centrality
8. Negative-weighted participation coefficient
9. Positive-weighted diversity coefficient
10. Negative-weighted clustering coefficient
11. Eigen vector centrality
12. Binarized efficiency
13. Weighted assortativity
14. Negative-weighted gateway coefficient
15. Binarized clustering coefficient
16. Resilience
17. Binarized modularity coefficient
18. Participation coefficient
19. Negative-weighted diversity coefficient
20. Positive-weighted participation coefficient
21. Characteristic path diameter
22. Characteristic path eccentricity
23. Characteristic path radius
24. Positive-weighted gateway coefficient
25. Characteristic path lambda
26. Characteristic path efficiency

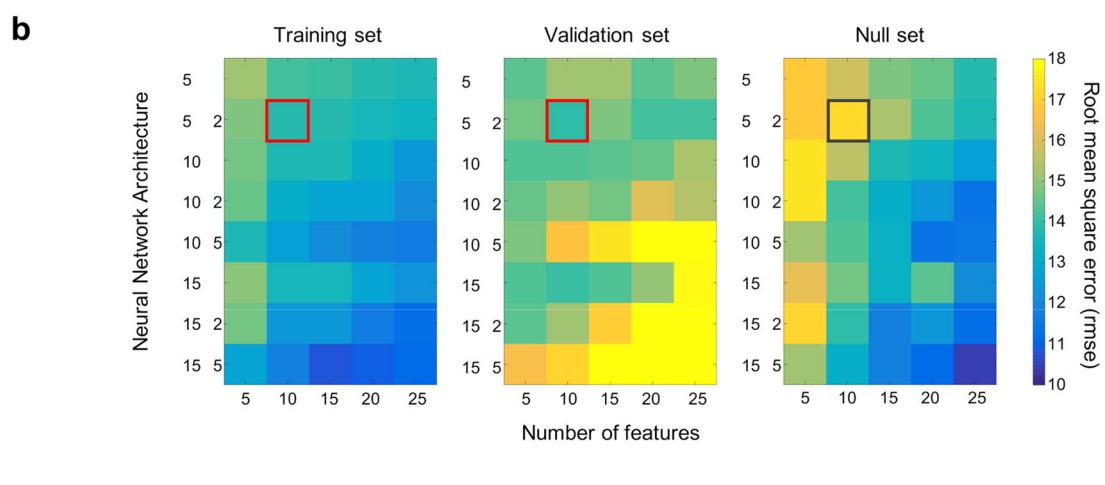

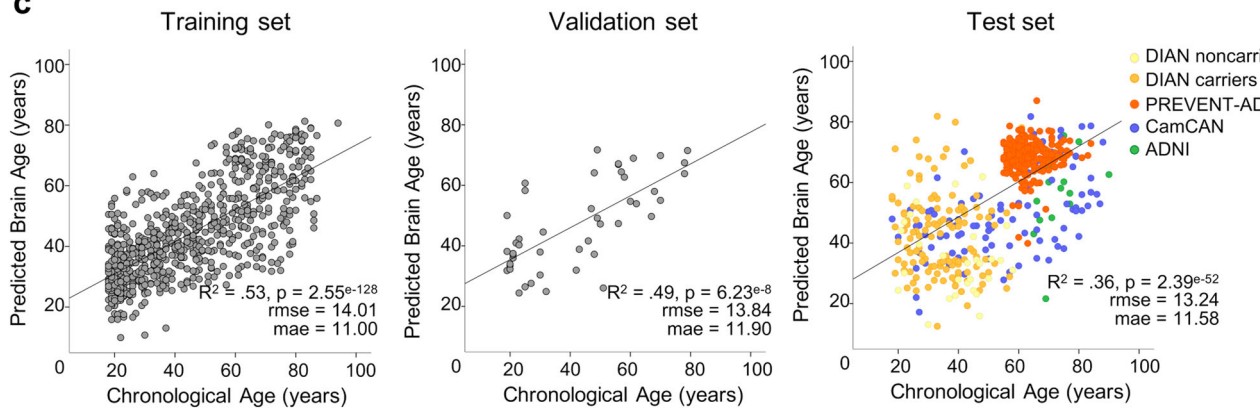

**Fig. 2 Features ranking and neural networks performance. a** Scatter plots of SVM model weights (*y*-axis) and ensemble tree feature importance (*x*-axis). Model weights are absolute value, and normalized such that 1 indicates highest importance. Numbers next to data points indicate their rank (i.e., 1 = highest average rank between both SVM and ensemble models; orange dots correspond to the top 10 features, blue dots represent lower-ranked features). **b** Root mean square error of different neural network models with inputs sorted according to rank for the training set (left), and the validation set (middle). Values were averaged over 3 iterations of the models. Neural networks trained with randomly-ranked inputs served as our null models (right). The *x*-axis indicates the number of inputs into the model (number of graph metrics) while the *y*-axis indicates the network architecture. For example, 5 means 1 hidden layer with 5 units, 5 2 means 2 hidden layers, the first one with 5 units and the second with 2 units. Darker (blue) colors indicate higher accuracy, while lighter (yellow) colors indicate lower accuracy. The red square identifying the model that provides the better generalizability in the validation set (lowest rmse) contains 2 hidden layers of 5 and 2 units, and uses the 10 highest-ranked graph metrics as input. The same neural network trained on randomly-ranked inputs (null model, gray square) provides lower accuracy. **c** Brain age model performance across datasets. Correlations between chronological age (*x*-axis) and age predicted by the neural network (*y*-axis) are represented for the training (*n* = 773), validation (*n* = 46) and test (*n* = 521) sets. Statistical values (**c**) were obtained from Pearson's correlations (two-sided test, with no adjustment). Source data are provided as a Source data file. SVM: support vector machine, rmse: root mean square error, mae: mean absolute error.

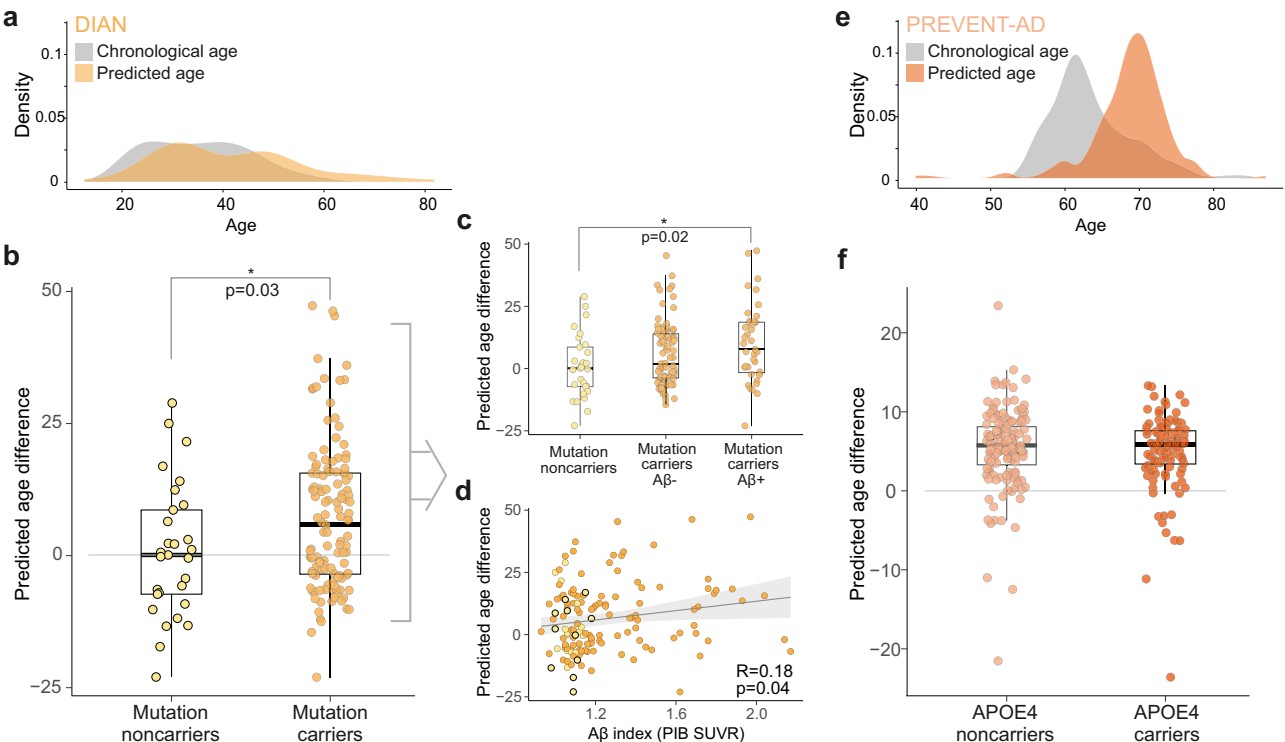

**Fig. 3 Predicted age difference in DIAN and PREVENT-AD.** Density plot of chronological age vs predicted age in the test set participants in DIAN ($n = 154$) (**a**). Brain age is overestimated in autosomal dominant mutation carriers ($n = 125$) compared to non-carriers ($n = 29$) (**b**). The overestimation in mutation carriers is in part due to Aβ, with a difference between mutation noncarriers ($n = 29$) and Aβ+ mutation carriers ($n = 39$) only (Aβ− mutation carriers [$n = 75$] did not differ from the other groups) (**c**), and an association between Aβ load and predicted age difference across the whole cohort ($n = 154$) (**d**). Light (yellow) colors represent DIAN mutation non-carriers and darker (orange) colors represent DIAN mutation carriers. Density plot of chronological age vs predicted age in the test set participants in PREVENT-AD ($n = 256$) (**e**). In individuals at risk of sporadic Alzheimer's disease, brain age is overestimated irrespectively of *APOE* ε4 genotype (**f**). Light (salmon) colors represent PREVENT-AD *APOE* ε4 non-carriers ($n = 147$) and darker (dark orange) colors represent PREVENT-AD *APOE* ε4 carriers ($n = 108$). For **b**, **c** and **f** the interquartile range (25th Percentile, Median and 75th Percentile), the whiskers (lines indicating variability outside the upper and lower quartiles minimum value) and the individual dots are presented. For **d**, shaded (gray) area represents confidence intervals (95%). Statistical values were obtained from general linear models (**b**, **c**, **f**) or partial Pearson's correlations (**d**), controlling for chronological age, without further adjustment (two-sided tests). Aβ: beta-amyloid, Aβ−: amyloid-negative, Aβ+: amyloid-positive; *APOE4*: apolipoprotein E4, PIB: Pittsburgh compound B, SUVR: standardized uptake value ratio. Source data are provided as a Source data file.

age for each dataset in Fig. 2c. As expected, predicted age was correlated with chronological age in the training set ($R^2 = 0.53$, $p < 0.0001$; rmse = 14.01, mean absolute error [mae] = 11.00; Fig. 2c left) and the validation set ($R^2 = 0.49$, $p < 0.0001$; rmse = 13.84; mae = 11.90; Fig. 2c middle). Of note, the neural net model outperformed the simpler models used in our feature ranking step (rmse = 16.45 for SVM and 16.45 for tree ensemble, see above). Importantly, the model was able to predict chronological age from functional brain properties in the test set ($R^2 = 0.36$, $p < 0.0001$; rmse = 13.24; mae = 11.58; Fig. 2c right). Notably, the same was true when restricting the analyses to the CamCAN cohort, considered as a lifespan dataset representative of healthy aging ($R^2 = 0.26$; $p < 0.001$; rmse = 16.70; mae = 14.32).

**Functional brain aging and pre-clinical Alzheimer's disease.** To assess the characteristics of functional brain aging in pre-clinical AD and evaluate whether genetic determinants/risk and Aβ pathology were related to accelerated brain aging, we calculated the predicted age difference or PAD (Fig. 1d). This was computed as predicted brain age minus chronological age for each participant in the test set[3]. PAD deviation from zero should not be interpreted in isolation due to the potential existence of site/cohort effects, and we, therefore, interpret group comparisons only within cohorts.

Analyses considered DIAN (Fig. 3a–d) and PREVENT-AD (Fig. 3e, f) participants from the test set (Table 2). We tested whether genes predisposing to AD, either the ADAD mutation carriers or the broader familial risk of sAD, were associated with accelerated brain aging. To do so, we compared PAD between mutation carriers vs non-carriers from DIAN, and *APOE* ε4 carriers *vs* non-carriers from PREVENT-AD. Considering the tendency of the model to overestimate younger ages and underestimate older ages, all subsequent analyses were controlled for chronological age (see ref. [14] for a similar procedure). The model's prediction in DIAN mutation carriers overestimated their chronological age (i.e., positive PAD = 8.19 years) in contrast to mutation non-carriers (i.e., negative PAD = −3.54 years; $F_{1,152} = 4.88$, $p = 0.03$; Table 3 and Fig. 3a, b). Overall, the predicted age in the PREVENT-AD cohort overestimated the chronological age by ~5 years (Fig. 3e), but *APOE* ε4 status was not associated with differences in this PAD ($F_{1,253} < 1$; $p = 0.49$, Table 3 and Fig. 3f).

Given the importance of Aβ deposition in the cascade of events leading to AD dementia, we investigated whether Aβ burden is related to functional brain aging. We assessed the effect of Aβ deposition, measured by PET, on the PAD in both the DIAN and PREVENT-AD cohorts. Aβ-PET was acquired using [11]C-PIB in DIAN and [18]F-NAV4694 in PREVENT-AD, and Aβ burden was determined for each cohort according to their own processing

**Table 2 DIAN and PREVENT-AD test set characteristics.**

|  | DIAN mutation non-carriers | DIAN mutation carriers | PREVENT-AD |
|---|---|---|---|
| N | 29 | 125 | 256 |
| Chronological Age (years; mean ± SD) | 38.90 (±11.55) | 34.33 (±9.66) | 63.51 (±5.37) |
| Sex Ratio F/M (%) | 18/11 (62%/38%) | 68/57 (54%/46%) | 189/67 (74%/26%) |
| Education (years; mean ± SD) | 14.41 (2.13) | 14.89 (±3.10) | 15.65 (±3.51) |
| EYO (years; mean ± SD)[a] | −8.56 (±10.85) | −14.18 (±8.94) | −10.42 (±7.21) |
| APOE4 carriers (%)[b] | 11 (38%) | 36 (29%) | 108 (42%) |
| Aβ-positive (%)[c] | 0 (0%) | 39 (34%) | 14 (22%) |
| MMSE or MoCA (mean ± SD)[d] | 29.36 ± 1.03 | 29.02 ± 1.27 | 28.11 ± 1.52 |

*Aβ beta-amyloid, Aβ− amyloid-negative, Aβ+ amyloid-positive; APOE apolipoprotein E, EYO Estimated Years to Symptom Onset, MMSE Mini–Mental State Examination, MoCA Montreal Cognitive Assessment, SD standard deviation.*
[a]EYO in DIAN was calculated based on the parental age at onset. EYO of PREVENT-AD was calculated only for individuals with a parental history of AD, data was available for 241 participants.
[b]APOE genotyping was missing for 1 PREVENT-AD participant.
[c]Aβ-PET data was missing for 2 mutation non-carriers and 11 mutation carriers in DIAN and 192 PREVENT-AD participants.
[d]Global cognitive functioning was assessed using MMSE in DIAN and MoCA in PREVENT-AD.

pipelines and methods (see Methods section for Aβ measurements details). We assessed the influence of Aβ burden on functional brain aging by comparing Aβ-positive to Aβ-negative individuals. We also explored the possible influence of Aβ as a continuous variable by assessing the partial correlation between PAD and Aβ load. All analyses were controlled for chronological age.

In DIAN, we found a grading effect of (quasi-continuous) Aβ on PAD. Higher PAD was observed in Aβ-positive mutation carriers when compared to the group of non-carriers ($F_{1,65} = 6.9$, $p = 0.02$; Fig. 3c). However, the PAD in Aβ-negative carriers compared to non-carriers was only marginally higher ($F_{1,101} = 2.73$, $p = 0.10$; Fig. 3c). There were no significant differences between DIAN Aβ-positive and Aβ-negative mutation carriers ($F_{1,111} = 1.93$, $p = 0.17$; Fig. 3c, Table 3). Partial Pearson correlations showed that accelerated brain age was associated with increased fibrillar Aβ load in the entire DIAN cohort ($r_{140} = 0.18$, $p = 0.04$; Fig. 3d), a finding that was no longer significant when the analysis was restricted to mutation carriers ($r_{111} = 0.14$, $p = 0.14$).

In PREVENT-AD, among the 64 individuals who underwent Aβ-PET imaging (test set only), 50 were Aβ-negative and 14 were Aβ-positive (Table 3). The PAD was not associated with Aβ burden, either when looking at Aβ-positivity ($F_{1,61} < 1$; $p = 0.33$) or the influence Aβ load ($r_{61} = 0.12$; $p = 0.35$). Adding the delay between PET and rs-fMRI assessments as a covariate provided similar results ($F_{1,60} < 1$; $p = 0.36$ using Aβ-status and $r_{61} = 0.12$; $p = 0.37$ for the partial correlation with Aβ load).

In supplementary analyses, we explored the association between PAD and estimated years to symptom onset (EYO). EYO has been widely used as an estimate of disease progression in DIAN[22,37], and it has been associated with amyloid pathology in individuals having a parental history of sporadic AD[38,39]. This index, calculated as the difference between parental age at symptom onset and participant's chronological age, estimates each individual's proximity to symptom onset (see Supplementary Methods for details). A weak but positive association was found between EYO and PAD in the PREVENT-AD ($r = 0.13$, $p = 0.05$), such that individuals that had higher PAD tended to also be closer to their expected age of onset. No such association was found in DIAN mutation carriers ($r = −0.13$, $p = 0.16$), or non-carriers ($r = −0.23$, $p = 0.23$).

Finally, we performed additional *post hoc* analyses to test whether sAD symptomatic individuals (MCI and dementia) had a higher PAD than asymptomatic individuals at risk of sAD (APOE ε4 carriers). This analysis was not initially planned, and was conducted only in a small subsample of the ADNI dataset (15 asymptomatic APOE ε4 carriers from the test set and 100 symptomatic individuals). The findings do suggest, as expected, increased PAD among individuals with cognitive impairment as compared with asymptomatic individuals at risk of sAD (using parametric, $F_{1,112} = 2.85$, $p = 0.047$, or non-parametric Mann–Whitney-$U = 965$, $p = 0.04$, one-tailed test).

## Discussion

Variation in notional biological aging has been proposed to account for inter-individual differences in the way people age[40]. Combined with larger and more available datasets, machine learning methods can improve our understanding of brain function and our ability to predict health trajectories from brain properties. Previous models of brain aging have been informed primarily by characteristics of brain structure[41]. Accelerated structural brain aging has been found in individuals with MCI and AD dementia[12,14,15]. However, functional brain abnormalities are generally detectable prior to structural changes in the AD continuum, the latter being typically more proximate to the expression of clinical symptoms[25,42,43]. Here we developed a model that could evidently predict brain age across the entire adult human lifespan (ages 18–94). This model relied on topological properties of graphs constructed from rs-fMRI and demonstrates the feasibility of predicting brain age from rs-fMRI using global measures of network integration and segregation[6,36]. Applying our predictive functional model to ADAD in the DIAN cohort, we observed that brain aging was apparently accelerated in individuals with pre-clinical ADAD. This association was especially clear in individuals who had PET evidence of Aβ deposition. Among individuals at elevated risk of sAD (PREVENT-AD cohort), neither APOE ε4 nor Aβ was associated with accelerated brain aging. However, asymptomatic individuals who were closer to their expected age of symptom onset tended to show accelerated brain aging. The latter observation was corroborated by observations that symptomatic individuals with sAD showed accelerated brain aging when compared to asymptomatic individuals at risk (ADNI cohort, secondary analyses).

We developed the described model in participants from different cohorts and sites, and validated its generalizability in an independent monocentric dataset. While there is undoubtedly a cost to (internal) accuracy when optimizing model (external) generalizability, this external validation step is a major strength of this work. While modest in size, the validation set represented a completely independent dataset that covers the entire adult lifespan. Although we cannot exclude the possibility that a larger multicenter validation cohort might have led to selection of a slightly different network architecture, we note that our model's rmse was very similar between the validation and the test sets. Importantly, the test set was never used in the development/

**Table 3 Model's prediction according to the presence of genetic mutation/risk and Aβ pathology in DIAN and PREVENT-AD cohorts (test set only).**

| | DIAN | | | | | PREVENT-AD | | | |
|---|---|---|---|---|---|---|---|---|---|
| | Mutation NonCarriers | Mutation Carriers | Mutation NonCarriers Aβ− | Mutation Carriers Aβ− | Mutation Carriers Aβ+ | APOE4 NonCarriers | APOE4 Carriers | Aβ− | Aβ+ |
| N | 29 | 125 | 29[a] | 75 | 39 | 147 | 108 | 50 | 14 |
| Chronological Age in years (±SD) | 38.90 (±11.55) | 34.35[b] (±9.66) | 38.90 (±11.55) | 33.15[b] (±9.09) | 38.18[b] (±10.04) | 63.89 (±5.74) | 63.05 (±4.77) | 62.90 (±4.28) | 64.71 (±5.69) |
| Predicted Age in years (±SD) | 35.35 (±12.90) | 42.51[b] (±13.90) | 35.35 (±12.90) | 41.55[b] (±13.19) | 41.18[b] (±14.09) | 68.94 (±5.14) | 68.46 (±5.01) | 68.80 (±6.08) | 70.37 (±2.97) |
| PAD in years (±SD) | −3.54 (±18.99) | 8.19[b] (±17.63) | −3.54 (±18.99) | 8.40[b] (±17.21) | 6.62[b] (±18.09) | 5.04 (±7.62) | 5.41 (±6.79) | 5.90 (±7.60) | 5.66 (±6.56) |

Aβ− amyloid-negative, Aβ+ amyloid-positive, PAD predicted age difference, SD standard deviation.
[a]Aβ-PET data was missing for 2 mutation non-carriers. To avoid further reducing the number of individuals in this group, they were both exceptionally kept for the corresponding analysis and their Aβ negativity was determined based on (i) their CSF data (available for both of them) and (ii) the unlikelihood that they have detectable Aβ pathology on PET considering their young age and the absence of ADAD mutation.
[b]indicates a significant difference from DIAN mutation noncarriers (p < .05); of note, no differences were found between DIAN mutation carriers Aβ− and DIAN mutation carriers Aβ+, neither in PREVENT-AD when comparing APOE4 noncarriers to APOE4 carriers or Aβ− to Aβ+. Statistical values were obtained using general linear models, including group as a predictor, without adjustment (two-sided).

validation of the brain aging model. Also, the model was not modified any further after the hypotheses were tested, i.e., hypotheses were only tested once using a model that appeared (from development and validation work) to be optimal. This approach ensured that our results regarding brain aging in pre-symptomatic AD were independent of the way the model was built.

To assess information integration in the brain, we relied on global brain function while applying graph metrics[6,36]. This approach provides a holistic view of brain function that has been shown previously to change through aging and AD[44]. Graph theory has the advantage that it quantifies and simplifies the many "moving parts" of dynamic systems inasmuch as every connection is defined by its relation to all others. We also used feature selection as an intermediate step to simplify the final model. We suggest that our approach using graph theory and feature selection are steps in the right direction toward interpretability of complex models. We are encouraged that the 10 graph metrics suggested as most important by these algorithms provided much lower error in our final neural network model in comparison to random choice of graph metrics. Of note, models using individual functional connections as inputs are also possible, but such models have been shown to require multiple dozens[45] or hundreds[10] of functional connections whose interrelationships are not defined.

Compared with structural predictive models, previous modeling approaches using rs-fMRI data have found higher error[7,19,46,47]. These observations could partly be attributable to known characteristics of rs-fMRI data. Such data are typically noisier and experience more dynamic changes than structural data, and they may be more sensitive to multi-site effects. Despite these difficulties, we attempted to derive our brain age model from rs-fMRI because this modality appears better suited to study of the pre-clinical phase of AD. An extensive literature suggests that connectivity disruption appears early in the course of sAD as well as in "normal" aging[48–53]. Moreover, training, validating, and testing our predictive model across multiple cohorts also increased the error of our model compared to the previous studies[3,7,11,13,14]. Yet, inclusion of data from different sites should logically improve the generalizability of the model, a key strength when the model is applied to new data from different cohorts[7,54]. Finally and importantly, brain age models tend to overestimate younger ages and underestimate older ages[19,55]. While some researchers apply an age-bias correction procedure to their model[19], we are showing the non-adjusted model and used chronological age as a nuisance variable in our PAD analyses instead of applying this correction prior to the PAD calculation. In sum, while we recognize that the error of our model is higher than most previous brain age models, it was derived from rs-fMRI data, no age-bias correction was applied to test the model accuracy, and we suggest that it is more generalizable than previous models. Crucially, it also appears to be sensitive to the questions of interest here.

Applying our model in the context of AD, we found evidence of accelerated functional brain aging in individuals in the pre-clinical phase of dominantly inherited AD. ADAD is widely believed to be a disease caused by overproduction of Aβ, and studies of ADAD have shown that biomarkers such as CSF-Aβ, start changing in mutation carriers as early as 25 years before symptom onset[22]. This is followed by the accumulation of fibrillar Aβ deposition as measured by PET imaging, alongside changes in concentrations of *tau* in the CSF and cerebral atrophy. Later changes include glucose hypometabolism and episodic memory decline and global cognitive decline[22]. Studies employing rs-fMRI in this disease are relatively rare, however, and have considered the entire ADAD spectrum[48,56–58]. One of these studies

compared asymptomatic mutation carriers and non-carriers and suggested reduction in Default Mode Network functional connectivity among asymptomatic carriers[56]. This finding is concordant with literature on sAD suggesting that change in rs-fMRI is one of the earliest biomarkers of the disease[25,26]. Our rs-fMRI predictive model implied that functional brain age of ADAD pre-symptomatic mutation carriers (DIAN) exceeded their chronological age by about 10 years (based on the findings in non-carriers). This observation alone suggests that the pre-symptomatic phase of ADAD is accompanied by accelerated brain aging. The relative importance of Aβ burden on accelerated brain aging was less clear. While no association was found between Aβ burden when restricted to DIAN mutation carriers, the difference between mutation carriers and non-carriers was stronger (i.e., significant only) in those with fibrillar Aβ as detected with PET imaging. The observations of accelerated brain aging in carriers may therefore not be entirely attributable to the accumulation of Aβ. While Aβ is often hypothesised to be the starting point of the AD neuropathological cascade[59], *tau* is believed to be more toxic[60,61] and might therefore be more closely associated with accelerated aging. Mutated genes in ADAD could also have life-long effects on the brain that are not fully dependent on Aβ accumulation. Consistent with this view, a previous study in *PSEN1* mutation carriers from the Columbian cohort showed early changes in brain function before evidence of cerebral Aβ plaque accumulation[62]. Finally, we cannot exclude the possibility that some Aβ-negative individuals would in fact be Aβ accumulators[63,64], or present other forms of Aβ that cannot be detected through PET. What seems to be clear is that AD genetic mutations influence functional brain properties in pre-clinical ADAD. The exact mechanisms that drive this accelerated brain aging will need further investigation.

When investigating the characteristics of PAD in individuals with a family history of sAD (PREVENT-AD), we did not find differences between *APOE* ε4 carriers and non-carriers, nor associations with Aβ burden (for similar results with structural and metabolic brain age, see refs. [11,65]). These findings do not necessarily contradict an extensive literature suggesting association between *APOE* ε4 status, Aβ, and rs-fMRI[44,66–69]. They seem instead to suggest that we are capturing different constructs. While the previous studies tested the direct effect of these two factors on rs-fMRI metrics, we tested the associations between these AD risk factors and a proxy of biological aging derived from rs-fMRI. While we found no association between Aβ and PAD, we did observe, however, an association between EYO and PAD such that PREVENT-AD participants who were closer to their parents' age of onset tended to have older predicted brain age when compared with others. In the same cohort, EYO had previously been associated with functional changes mimicking brain changes characteristic of AD dementia[70].

While our focus was the pre-clinical phase of the disease, we performed *post-hoc* analyses using rs-fMRI data from a small subset of ADNI patients. We found accelerated functional aging in persons with symptomatic sAD (MCI or dementia) when compared with others who were asymptomatic, but at increased risk of sAD (*APOE* ε4 participants from our test set). These additional analyses suggest accelerated functional brain aging in individuals with clinical sAD and further confirm the validity of our brain age model.

Several limitations should be mentioned. These relate both to the model and to the cohorts used to test our hypotheses. First, our choice not to update or tweak the model after it was used to test our hypotheses (a main strength of our approach) left us with a few small errors when constructing the model (e.g., two PREVENT-AD*APOE* ε4 carriers were included in the training set). While these oversights were unlikely to have affected the

final results (*APOE* ε4 carriers from other cohorts without genotype data were presumably included in the training set), they nevertheless pose a small threat to the integrity of the model. Second, we also cannot exclude the possible influence of collinearity when determining the age predictive graph metrics. The SVM and the tree ensemble were however mostly in agreement, and it's unlikely that multicollinearity would have had equal influence on these two very different algorithms. Also, while we made great efforts to increase the generalizability of our predictive model, most of the participants included in this study were Caucasian (see Supplementary Methods), stressing the need to increase diversity in both lifespan and AD cohorts. Functional brain age was also found to exceed chronological age in the PREVENT-AD cohort, while this was not the case in other sites/cohorts of similar ages. While it is tempting to interpret these results as resulting from the participants' family history, we think it reflects largely a site effect. To minimize such possible site effects, we drew on data from a variety of cohorts and sites, validated the model on a completely independent validation set (new site) and applied similar processing methods to all data. No further harmonization procedure was applied. The site effects are inherently related to the different age composition of the sites (or cohorts, see Fig. 1), and thus harmonizing by sites would have removed the age difference between participants (see Supplementary Fig. 2 for an example of sites correction using ComBat; https://github.com/Jfortin1/ComBatHarmonization). While the possibility of site effects limits our ability for direct comparisons across cohorts, it cannot reasonably threaten the integrity of our main findings, which resulted from within-cohort comparisons. One obvious limitation of inference from the PREVENT-AD data compared to those from DIAN, is that we cannot know which participants will later develop AD dementia. The lack of evidence for accelerated brain aging in PREVENT-AD*APOE* ε4 carriers (vs non-carriers) might reflect nothing more than the known fact that not all *APOE* ε4 carriers will develop AD dementia (i.e., are in the pre-clinical phase of the disease) while some non-carriers will develop the disease. The subsample of PREVENT-AD participants having Aβ pathology in the test set was also relatively small, which could likely limit inference.

In sum, using rs-fMRI graph metrics, we developed a model that can predict brain age across the whole human lifespan. Applying this model to predict brain aging in the context of pre-clinical AD revealed that the pre-symptomatic phase of ADAD is characterized by accelerated functional brain aging. Whether a similar relationship holds for pre-clinical sAD and by which underlying mechanisms AD accelerates brain aging will require further evaluation.

## Methods
### Cohorts and participants
*Dominantly Inherited Alzheimer Network—DIAN*. DIAN is a multisite longitudinal study[71], which enrolls individuals age 18 and older who have a biological parent that carries a genetic mutation responsible for ADAD. They all underwent clinical and cognitive assessments, genetic testing, and imaging (magnetic resonance imaging [MRI] and amyloid-positron emission tomography [PET]). Data has been obtained after request and IRB approval (information can be found at dian.wustl.edu/our-research/observational-study/). Baseline data from cognitively unimpaired mutation carriers and non-carriers archived in the DIAN data freeze 10 (January 2009 to May 2016) were used in the present study. All selected individuals had a Clinical Dementia Rating (CDR)[72] scale of 0. Baseline data from 280 cognitively unimpaired individuals (mutation carriers and non-carriers) aged between 18 and 69 years old, for whom structural MRI and rs-fMRI data were available, have been included.

*Pre-symptomatic Evaluation of Experimental or Novel Treatments for Alzheimer's Disease—PREVENT-AD*. The PREVENT-AD (Douglas Mental Health University Institute, Montréal) is a monocentric longitudinal cohort[73]. Briefly, 399 cognitively unimpaired older individuals with a family history of sAD (at least one parent or multiple siblings) were enrolled between September 2011 and November 2017.

Inclusion criteria included (i) being 60 or older; 55–59 for individuals who were less than 15 years from the age of their relative at symptom onset, (ii) being cognitively normal and (iii) no history of major neurological or psychiatric disease. Normal cognition was defined as CDR of 0 and a Montreal Cognitive Assessment (MoCA)[74] ≥24. In the few cases of ambiguous results (3 participants having a CDR of 0.5 and 1 participant with a MoCA of 23 in the present sample), participants were further evaluated with a more extensive neuropsychological test battery, which was carefully reviewed by neuropsychologists and physicians to ensure normal cognition. Participants underwent clinical and cognitive examinations, blood tests, and MRI annually. Data from the present study were archived in the Data Release 5.0 and are partially available at https://openpreventad.loris.ca/. PET scans were acquired in a subset of participants between February 2017 and July 2019. Three hundred and fifty-three participants, aged 55–84, for whom baseline structural MRI and rs-fMRI were available were included in the present study.

*Cambridge Centre for Ageing and Neuroscience—CamCAN.* The Cambridge Centre for Ageing and Neuroscience (Cam-CAN; http://www.cam-can.org/) is a large-scale collaborative research project, launched in October 2010, using epidemiological, behavioral, and neuroimaging data to characterize age-related changes in cognition and brain structure and function, and to uncover the neurocognitive mechanisms that support healthy cognitive ageing[75]. In the present study, 648 individuals aged between 18 and 88, with structural MRI and rs-fMRI data were included.

*Alzheimer's Disease Neuroimaging Initiative—ADNI.* ADNI data used in the preparation of this article were obtained from the Alzheimer's Disease Neuroimaging Initiative (ADNI) database (adni.loni.usc.edu)[76]. ADNI was launched in 2003 as a public-private partnership, led by Principal Investigator Michael W. Weiner, MD. The primary goal of ADNI has been to test whether serial MRI, PET, other biological markers, and clinical and neuropsychological assessment can be combined to measure the progression of MCI and early AD. Considering the focus on preclinical AD, forty-nine cognitively unimpaired individuals with structural MRI and rs-fMRI data were included in the present study. An additional 106 (100 after quality control) individuals with MCI or dementia and structural MRI and rs-fMRI data were included in *post hoc* analyses to validate the model in cognitively impaired sAD individuals.

*1000-Functional Connectomes Project (Cambridge site)—FCP-Cambridge.* The 1000-Functional connectomes project (FCP) is a large initiative that gathers functional data from cognitively unimpaired adults recruited worldwide (33 sites) and makes it publicly available to facilitate discovery science of brain function (http://fcon_1000.projects.nitrc.org/fcpClassic/FcpTable.html)[77]. We used the large dataset from Cambridge-Buckner that includes 198 subjects between 18 and 30 years old, all collected at the Cambridge site ([FCP-Cambridge], PI: Buckner, R.L.).

*International Consortium for Brain Mapping—ICBM.* The ICBM dataset[78] is publicly available as part of the 1000-FCP repository (see above; see also ref. [79] for details). The dataset is constituted of 86 cognitively unimpaired older adults from 19 to 95 years old who underwent structural MRI and rs-fMRI at the same site (Montreal Neurological Institute, Canada).

For the purpose of the brain age model, participants were divided into training, validation, and test sets. In order to reach the most accurate model of "healthy" brain aging from our data, cognitively unimpaired individuals from the different cohorts were assigned randomly to the training set, except when their genetic status was available (DIAN, PREVENT-AD, and ADNI), in which case only individuals with no genetic predisposition for AD were included in the training set; the remaining data (including individuals at increased risk of AD) was assigned to the test set. Thus, mutation non-carriers from DIAN (~80% of DIAN non-carriers, randomly selected) were assigned to the training set, along with ADNI *APOE4* non-carriers, individuals from FCP-Cambridge, and ~80% of the cognitively unimpaired individuals selected randomly from CamCAN. While the PREVENT-AD cohort has an increased risk of sAD, a few individuals from this cohort (~10%) were assigned to the training set to expose the model to this site's characteristics; these individuals were randomly selected from the subsample of *APOE4* non-carriers (with the exception of two *APOE4* carriers who were included in the training set by mistake). ICBM was used as an independent sample of healthy individuals to assess the generalizability of the brain age model to other datasets (validation set). Finally, the test set included our population of interest (DIAN mutation carriers, most PREVENT-AD participants) and the remaining asymptomatic individuals from the other cohorts (remaining ~20% DIAN mutation non-carriers and CamCAN participants, along with ADNI *APOE4* carriers).

**Standard protocol approvals, registrations, and participants consents.** All studies were approved by study sites' respective regional ethics committees.

More specifically, DIAN study procedures were approved by the Washington University Human Research Protection Office and the local institutional review boards of the participating sites.

PREVENT-AD study was approved by the Research, Ethics and Compliance Committee of McGill University (Montréal, Canada).

The Ethics committees/institutional review boards that approved the ADNI study are: Albany Medical Center Committee on Research Involving Human Subjects Institutional Review Board, Boston University Medical Campus and Boston Medical Center Institutional Review Board, Butler Hospital Institutional Review Board, Cleveland Clinic Institutional Review Board, Columbia University Medical Center Institutional Review Board, Duke University Health System Institutional Review Board, Emory Institutional Review Board, Georgetown University Institutional Review Board, Health Sciences Institutional Review Board, Houston Methodist Institutional Review Board, Howard University Office of Regulatory Research Compliance, Icahn School of Medicine at Mount Sinai Program for the Protection of Human Subjects, Indiana University Institutional Review Board, Institutional Review Board of Baylor College of Medicine, Jewish General Hospital Research Ethics Board, Johns Hopkins Medicine Institutional Review Board, Lifespan—Rhode Island Hospital Institutional Review Board, Mayo Clinic Institutional Review Board, Mount Sinai Medical Center Institutional Review Board, Nathan Kline Institute for Psychiatric Research & Rockland Psychiatric Center Institutional Review Board, New York University Langone Medical Center School of Medicine Institutional Review Board, Northwestern University Institutional Review Board, Oregon Health and Science University Institutional Review Board, Partners Human Research Committee Research Ethics, Board Sunnybrook Health Sciences Centre, Roper St. Francis Healthcare Institutional Review Board, Rush University Medical Center Institutional Review Board, St. Joseph's Phoenix Institutional Review Board, Stanford Institutional Review Board, The Ohio State University Institutional Review Board, University Hospitals Cleveland Medical Center Institutional Review Board, University of Alabama Office of the IRB, University of British Columbia Research Ethics Board, University of California Davis Institutional Review Board Administration, University of California Los Angeles Office of the Human Research Protection Program, University of California San Diego Human Research Protections Program, University of California San Francisco Human Research Protection Program, University of Iowa Institutional Review Board, University of Kansas Medical Center Human Subjects Committee, University of Kentucky Medical Institutional Review Board, University of Michigan Medical School Institutional Review Board, University of Pennsylvania Institutional Review Board, University of Pittsburgh Institutional Review Board, University of Rochester Research Subjects Review Board, University of South Florida Institutional Review Board, University of Southern, California Institutional Review Board, UT Southwestern Institution Review Board, VA Long Beach Healthcare System Institutional Review Board, Vanderbilt University Medical Center Institutional Review Board, Wake Forest School of Medicine Institutional Review Board, Washington University School of Medicine Institutional Review Board, Western Institutional Review Board, Western University Health Sciences Research Ethics Board, and Yale University Institutional Review Board.

The CamCAN study has been approved by the local ethics committee, Cambridgeshire 2 Research Ethics Committee.

For the 1000-Functional Connectomes Project (ICBM and FCP-Cambridge), each contributor's respective ethics committee approved submission of deidentified data. The institutional review boards of NYU Langone Medical Center and New Jersey Medical School approved the receipt and dissemination of the data.

All participants gave written informed consent prior to participation.

**MRI acquisition and processing.** DIAN: DIAN imaging data was acquired at multiple sites on 3T scanners by applying ADNI parameters and procedures[22,71]. T1-weighted MRI (used for rs-fMRI processing) were acquired with the following parameters: repetition time (TR) = 2400 ms, echo time (TE) = 16 ms, flip angle = 8°, acquisition matrix = 256 × 256, voxel size = 1 × 1 × 1 mm. Eyes-open rs-fMRI images were acquired using the following parameters: TR = 2230 ms or 3000 ms; TE = 30 ms, flip angle = 80°, voxel-size = 3.3 × 3.3 × 3.3 mm, field of view (FOV) = 212, 140 volumes; acquisition lasting ~5 min or 7 min.

PREVENT-AD: MRI data were acquired on a 3T Magnetom Tim Trio (Siemens) scanner. T1-weighted images were obtained using a GRE sequence with the following parameters: TR = 2300 ms; TE = 2.98 ms; flip angle = 9°; matrix size = 256 × 256; voxel size = 1 × 1 × 1 mm; 176 slices. For resting state fMRI scans, two consecutive functional T2*-weighted scans were collected eyes-closed with a blood oxygenation level-dependent (BOLD) sensitive, single-shot echo planar sequence with the following parameters: TR = 2000ms; TE = 30 ms; flip angle = 90°; matrix size = 64 × 64; voxel size = 4 × 4 × 4 mm; 32 slices; 150 volumes, acquisition time = 5min45s. For consistency with the other cohorts that only had one run, only the first run was considered for each participant.

CamCAN: Images were acquired on a 3T Magnetom Tim Trio (Siemens). T1-weighted MRI were acquired using the following parameters: 3D MPRAGE GRAPPA = 2, TR = 2250 ms, TE = 2.99 ms, TI = 900 ms; flip angle = 9°; voxel-size 1 mm isotropic; FOV = 256 × 240 × 192 mm; acquisition time = 4 min 32 s. Rs-fMRI data were acquired eyes closed using a T2* GE EPI sequence with the following parameters: TR = 1970 ms; TE = 30 ms, flip angle = 78°; voxel-size = 3 × 3 × 4.44 mm, FOV = 192 × 192; 261 volumes of 32 axial slices 3.7 mm thick with a 0.74 mm gap, acquisition time = 8 min 40 s.

ADNI: Data were acquired at multiple sites, following the ADNI protocol[80]. Structural images were acquired using a 3D MPRAGE T1-weighted sequence with the following parameters: TR = 2300 ms; TE = 2.98 ms; TI = 900 ms; flip angle = 9°; voxel size=1.1 × 1.1 × 1.2 mm³; FOV = 256 × 240 mm²; 170 slices. The rs-fMRI images were obtained, eyes open, using a T2 weighted echo-planar imaging sequence with the following parameters: TR = 3000 ms; TE = 30 ms; flip angle = 80°; 48 slices of 3.3 mm; 140 volumes; acquisition lasting ~5 min.

FCP-Cambridge: Images were acquired using a Siemens 3T Trio scanner. High-resolution T1-weighted images were acquired as follows: MP-RAGE TR = 2200 ms, TE = 1.04–7.01 ms, flip angle = 7°, voxel size = 1.2 × 1.2 × 1.2 mm, FOV = 230 mm, 144 sagittal slices. Rs-fMRI were collected, eyes open, with the following parameters: EPI TR = 3000 ms, TE = 30 ms, flip angle = 85°, voxel size = 3 × 3 × 3 mm, FOV = 216 mm, 47 axial slices, 124 volumes, lasting ~6 min.

ICBM data was acquired on a Siemens Sonata 1.5 T MR scanner at the MNI. T1-weighted scan was acquired as follows: TR = 2200 ms, TE = 92 ms, flip angle = 30°, 256 × 256 matrix with a 1 × 1 mm² resolution, 176 contiguous sagittal slices covering the whole-brain, slice thickness = 1 mm. Three rs-fMRI runs were acquired eyes-closed with the following parameters: 2D echoplanar BOLD MOSAIC sequence, TR = 2000 ms, TE = 50 ms, flip angle = 90°, 64 × 64 matrix with a 4 × 4 mm² resolution, 23 contiguous axial slices covering the cortex but not the cerebellum, slice thickness = 4 mm, 138 volumes; each run lasting ~4 min 30 s. For consistency with the other cohorts that only had one run, only the first run was considered for each participant.

**Rs-fMRI processing**. In order to limit site effects, all functional images were processed in our laboratory (by APB) applying the exact same pipeline and processing steps. The NeuroImaging Analysis Kit version 0.12.4 (NIAK; http://niak.simexp-lab.org/) was used for rs-fMRI preprocessing, following the procedure applied in the previous publications[54,70]. Briefly, images underwent slice timing correction, and rigid-body motion parameters were estimated. T1-weighted images were linearly and non-linearly normalized to the MNI space. After coregistration to structural scans, functional images were normalized to the MNI space by applying parameters from the T1-weighted images and resampled to 2 mm isotropic. Slow time drifts, average white matter and cerebrospinal fluid signal and motion artifacts (first principal components of the six realignment parameters, and their squares) were regressed out from the rs-fMRI time series. Finally, fMRI volumes were smoothed with a 6 mm Gaussian kernel. Frame displacement was calculated and those exhibiting displacement >0.5 were removed (scrubbed), along with one adjacent frame prior, and two consecutive frames after[81]. Images with less than 40% of their original data after scrubbing were discarded (see Supplementary Table 1 for the percentage of frames retained in each cohorts).

Overall, 266 individuals (16 DIAN, 60 PREVENT-AD, 130 CamCAN, 4 ADNI, 1 FCP-Cambridge, and 39 ICBM, as well as 6 ADNI patients [included in secondary analyses]) were discarded due to failing preprocessing standards or having insufficient data after scrubbing.

Average BOLD signals were extracted from 272 regions corresponding to the Power and Petersen functional atlas[82], to which key regions of the limbic system were added[83]. Regions labeled as "uncertain", or with weak or non-existent signal in any one image were excluded from all images, resulting in 238 total regions (see Supplementary Table 2 for the total listing of the regions). For each subject, BOLD activity time series from these regions were used to construct a 238 × 238 Pearson correlation matrix, which was then Fisher's Z-transformed.

Motion-related noise was further mitigated using the mean regression (MR) technique as outlined previously[84]. Briefly, the average of all correlation values within the upper diagonal of the correlation matrix was calculated for each subject in the training data. A linear fit between these across-subject average values and the across-subject value at each element of the correlation matrix was generated, creating a slope and intercept term associated with each element of the matrix. The final value used in each element of the correlation matrix was equal to the residual between the MR-model fit and the original correlation value. Importantly, the MR model was created with only the training data.

For each subject, 26 graph metrics, chosen based on their ability to quantify whole-brain connectivity, were extracted from the correlation matrix using the Brain Connectivity Toolbox (https://sites.google.com/site/bctnet/)[36], in Matlab. Both weighted and unweighted metrics were calculated, if applicable. Graph metrics were chosen because they outperformed models trained directly on the weighted edges of the matrices. In the case of unweighted metrics, correlation matrices were thresholded at 5% link density, which ensured only the top 5% strongest correlation values were counted as connections in the matrix[85]. Only 5 out of the 26 metrics used binarized matrices and out of those 5, only one was retained in the final model (i.e., weighted modularity coefficient). One global value was extracted for each graph metric. In cases where a metric was outputted for each region (e.g., subgraph centrality), the median or median of log values was used as a global estimate. Small-worldness and resilience metrics, not included in the toolbox but both shown to be strong indicators of age, were calculated as previously determined (see Supplementary Methods for details)[6]. Briefly, small-worldness was calculated as the averaged clustering coefficient of the correlation matrix divided by the averaged clustering coefficient of a random network with same node-edge count, which was divided by the averaged efficiency of a random network divided by the averaged efficiency of the correlation matrix. In graph theory, resilience of

network G is defined as the relative number of edges that must be removed for the network to lose property P, and is a measure of the network's robustness to targeted or random attacks. Here, resilience is calculated as the slope of the log-log degree distribution. Subjects with any graph metric that was 5 standard deviations beyond the training set group mean was removed from the analysis entirely. A total of 15 individuals from the training set (1 DIAN mutation non-carrier, 11 CamCAN, 2 FCP-Cambridge, 1 ADNI), 1 from the validation set (ICBM) and 8 from the test set (1 DIAN mutation non-carriers, 3 DIAN mutation carriers, 1 PREVENT-AD, 3 CamCAN) were excluded.

**Brain age model**. The general procedure for iterating through different models included 5-fold cross-validation within the training data, and a second validation with an independent, external-site dataset. Models with the lowest error in predicting age on this validation set then served as candidates for the final model. Once the final model was determined, our hypotheses were then tested on the test set. Of importance, the test set was composed of unseen data that were not used to create, optimize, or validate the model. Neither the model nor the hypotheses were modified after the model was considered final and ready for hypothesis testing.

First, in order to reduce the number of inputs to the model, we searched for the graph metrics that were the most reliably predictive of age. To do so, the training set data was entered in a support vector machine (SVM) and a regression tree ensemble model to estimate which graph metrics were the most important to predict chronological age (i.e., highest weights). For the SVM model, the features (i.e., the 26 metrics) were standardized by subtracting the mean and dividing by the standard deviation of the training group. SVM was implemented with the fitrlinear function using a linear kernel, and Bayesian-optimized ridge regularization. For ensemble methods, feature standardization is not recommended, and thus the unstandardized 26 metrics were used as input. The fitrensemble function was used with Bayesian optimization of hyperparameters including the method (Bag or LSBoost), number of learning cycles, and the learning rate. In both models, chronological age was the response vector, and parameter optimization was determined with the minimum 5-fold cross-validation loss. Feature selection was not part of cross-validation. Graph metrics were then ranked separately by order of SVM weights and ensemble model importance (i.e., highest load corresponding to the most important). Importance in the ensemble model was determined using the predictorImportance function, which is equal to the sum of changes in mean squared error due to splits on every predictor, divided by the number of branch nodes. The average rank from both models was then used to determine the overall importance of each metric.

In a second step, we aimed at creating an accurate model requiring the fewest number of features possible. We used training data to generate a neural net model and assessed its accuracy on the validation set. More specifically, the neural network was optimized by (i) generating different models using the training set, each model varying in number of features used as input and network complexity, and (ii) applying each model to the validation set (independent dataset/site) to evaluate which one provided the better generalizability (i.e., avoid overfitting and give the better prediction on an independent set). Graph metrics in both training and validation sets were standardized by subtracting the training group mean and dividing by the training group standard deviation. Network models had 5–25 input features in increments of 5, entered according to their importance, as determined previously (see above). A null model was also tested by applying the same feature increment procedure but entering the graph metrics in a random order. Each graph metric was only entered once as input for each neural network architecture tested, and the inputs were kept constant across the model's iterations; features of more complex models always included the features of the simpler ones. Architecture of the network was also tested with various number of hidden layers (1 or 2) and number of units in the hidden layers (2, 5, 7, or 10). Age was modeled on the training data using the fitnet function with Bayesian regularization backpropagation. Model accuracy was ultimately determined by the root mean squared error (rmse) between actual and predicted age on the validation data, with lower rmse reflecting higher accuracy. Because neural network units are initialized with random values, the rmse changed slightly each time model error was measured. Thus, the best model was determined by the lowest rmse, averaged over three iterations. Once the most accurate model was determined, it was applied on unseen data (test set).

**Additional measures in DIAN and PREVENT-AD samples (test set)**
*Genetics*. DIAN genotyping was performed by the DIAN Genetics Core at Washington University[22]. The presence or absence of ADAD mutation was determined using PCR-based amplification of the appropriate exon followed by Sanger sequencing. APOE genotype was determined using an ABI predesigned real-time Taqman assay (C___3084793_20 and C____904973_10 for rs429358 and rs7412 variants, respectively).

APOE genotype in PREVENT-AD was determined using the PyroMark Q96 pyrosequencer (Qiagen, Toronto, Canada) and the following primers: rs429358_amplification_forward 5′-ACGGCTGTCCAAGGAGCTG-3′, rs429358_amplification_reverse_biotinylated 5′-CACCTCGCCGCGGTACTG-3′, rs429358_sequencing 5′-CGGACATGGAGGACG-3′, rs7412_amplification_forward 5′-CTCCGCGATGCCGATGAC-3′,

rs7412_amplification_reverse_biotinylated 5′-CCCCGGCCTGGTACACTG-3′ and rs7412_sequencing 5′-CGATGACCTGCAGAAG-3′.

The full list of primers is provided in Supplementary Tables 3 and 4.

*PET acquisition and processing.* In DIAN, 28 mutation non-carriers and 117 mutation carriers from the test set had an Aβ-PET scans available at baseline. Aβ-PET scans were acquired in different centers, following ADNI protocol[86]. Briefly, participants were injected intravenously with 8 mCi to 18 mCi of $^{11}$C-PIB. Part of the participants underwent a full dynamic acquisition of 70 min, starting at the time of injection. The remaining part of the sample underwent a 30-min scan after a rest period of 40 min. A standard brain transmission scan (or computed tomography [CT] transmission scan for PET/CT scanners) was obtained for attenuation correction. Aβ-PET data was motion corrected and registered to their MRI[87]. Standardized uptake value ratio (SUVR) were calculated using the cerebellar gray matter as a reference and a global measure of Aβ burden was calculated by averaging SUVRs from the prefrontal cortex, temporal lobe, gyrus rectus, and precuneus of the Desikan-Killiany atlas[88]. A threshold of 1.31 was used to determine Aβ-positivity[87].

In the PREVENT-AD cohort, Aβ-PET scans were performed at the MNI (Montréal, Canada) on a Siemens HRRT. Sixty-four individuals from the test set underwent this examination, at a mean of 10.30 ± 5.63 months from their closest MRI session and 43.10 ± 17.95 months after their baseline session. A 30-min acquisition scan started 40 min after intravenous injection of ~5.4 mCi of $^{18}$F-NAV4694. Transmission scans were acquired for attenuation correction. Data were processed using a standard pipeline (see ref. [38] and https://github.com/villeneuvelab/vlpp for details). A global index of neocortical Aβ burden was derived by extracting, in native space, the mean standardized uptake value ratio (SUVR) of the frontal, temporal, parietal, and posterior cingulate cortex of the Desikan-Killiany atlas[88], using the cerebellum grey matter as reference region. A threshold for positivity was determined using Gaussian Mixture modeling[38] and scans with global neocortical Aβ burden ≥1.39 were considered positive.

*Estimated years to onset.* Estimated years from expected symptom onset (EYO) was calculated in each cohort taking the parental age at onset as a reference (see Supplementary Methods for details).

**Statistical analyses on the predicted age difference (test set).** To analyze the specificities of brain aging in the context of pre-clinical AD, we calculated the predicted age difference for DIAN and PREVENT-AD participants in the test set, as previously detailed[3], by subtracting the actual chronological age from the predicted brain age (output from the model). We were particularly interested in the influence of the genes involved in AD, which are either responsible for ADAD or increase the risk of sAD. We compared, in the test set, the predicted age difference (i.e., PAD) between (1) mutation non-carriers and mutation carriers from DIAN, and (2) *APOE*4 carriers vs non-carriers in the PREVENT-AD. We were also interested to further understand the influence of Aβ accumulation on functional brain aging in asymptomatic individuals. To do so we assessed the effect of Aβ deposition, measured by PET, on the PAD in both the DIAN and PREVENT-AD cohorts, both by comparing Aβ-positive and Aβ-negative individuals (dichotomous variable) and by assessing the correlation between PAD and Aβ load (continuous variable). All analyses were controlled for chronological age[14].

Exploratory analyses were conducted to assess the correlation between the PAD and estimated years to onset (EYO). Finally, we validated that our model was capturing advanced brain aging in sAD patients with cognitive impairment by comparing our cognitively unimpaired ADNI participants (15 *APOE*4 carriers from the test set) to a subset of 100 ADNI participants with MCI or dementia using a general linear model (one-tailed test), controlling for chronological age. Considering the small sample size in the control group, analyses were also replicated using nonparametric test (Mann–Whitney).

Analyses were conducted using Statistical Package for the Social Sciences (SPSS), and statistical significance was set at $p < 0.05$.

**Reporting summary.** Further information on research design is available in the Nature Research Reporting Summary linked to this article.

## Data availability

All data used in the present study are either publicly available (PREVENT-AD MRIs and demographics: https://openpreventad.loris.ca/; CamCAN: http://www.cam-can.org/; FCP-Cambridge and ICBM: http://fcon_1000.projects.nitrc.org/fcpClassic/FcpTable.html) or can be shared upon reasonable request and approval by the study scientific committees and/or institutional review boards (DIAN: dian.wustl.edu/our-research/observational-study; PREVENT-AD additional measurements, including PET: https://registeredpreventad.loris.ca/; ADNI: adni.loni.usc.edu). Sensitive (e.g., genetic, clinical) data are protected and are not provided due to data privacy laws. Source data are provided with this paper.

## Code availability

Custom codes of the neural networks developed in this study are provided at https://github.com/villeneuvelab/projects/tree/master/Gonneaud_2021_BrainAgeModel.

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

## Acknowledgements

The authors would like to thank the members of the Villeneuve Lab, J. Tremblay-Mercier, A. Labonté, D. Dea, C. Madjar, and all the PREVENT-AD center for participants' recruitment, data acquisition, and data management (a complete listing of PREVENT-AD Research Group can be found in the PREVENT-AD database: https://preventad.loris.ca/acknowledgements/acknowledgements.php?date=[2019-07-30]); the members of the Brain Imaging Center of the Douglas Mental Health Institute for MRI acquisitions; the member of the Cyclotron and PET Units of the Montreal Neurological Institute for PET tracer production and acquisitions; K. Paumier, R. Hornbeck, P. Wang, S. Flores, B. Esposito, and A. Renton for their help in DIAN data access and for providing DIAN procedure information, as well as all the centers involved in DIAN data acquisitions. A complete listing of the DIAN Study Group can be found in the Supplementary Notes. This manuscript has been reviewed by DIAN Study investigators for scientific content and consistency of data interpretation with previous DIAN Study publications. We acknowledge the altruism of the participants and their families and contributions of the DIAN research and support staff at each of the participating sites for their contributions to this study. Data used in preparation of this article were obtained from the Alzheimer's Disease Neuroimaging Initiative (ADNI) database (adni.loni.usc.edu). As such, the investigators within the ADNI contributed to the design and implementation of ADNI and/or provided data but did not participate in analysis or writing of this report. A complete listing of ADNI investigators can be found at: http://adni.loni.usc.edu/wp-content/uploads/how_to_apply/ADNI_Acknowledgement_List.pdf. Finally, we would like to thank all the participants and their family for their invaluable help. This work was supported by two Canada Research Chairs (S.V. and J.B.), a Canadian Institutes of Health Research project grant PJT-148963 (S.V.), a Canada Fund for Innovation (S.V.), an Alzheimer's Association Research Grant NIRG-397028 (S.V.), the Lemaire foundation (J.P. and S.V.), the J.L. Levesque Foundation (J.P.), a joint Alzheimer Society of Canada and a Brain Canada Research grant NIG-17-08 (S.V.), a StoP-AD fellowship (J.G.), a Quebec Bio-Imaging Network scholarship (J.G.), a joint FRQ-S and Alzheimer Society of Canada scholarship (A.P.B.). The PREVENT-AD was funded by a $13.5 million, 7-year public- private partnership using funds provided by McGill University, the Fonds de Recherche du Québec—Santé (FRQ-S), an unrestricted research grant from Pfizer Canada, the Levesque Foundation, the Douglas Hospital Research Centre and Foundation, the Government of Canada, the Canada Fund for Innovation and Genome Quebec Innovation Center (J.B. and J.P.). Data collection and sharing for this project was also supported by: (1) Data collection and sharing for this project was supported by The Dominantly Inherited Alzheimer's Network (DIAN, U19AG032438) funded by the National Institute on Aging (NIA), the German Center for Neurodegenerative Diseases (DZNE), Raul Carrea Institute for Neurological Research (FLENI), Partial support by the Research and Development Grants for Dementia from Japan Agency for Medical Research and Development, AMED, and the Korea Health Technology R&D Project through the Korea Health Industry Development Institute (KHIDI). (2) The Alzheimer's Disease Neuroimaging Initiative (ADNI) (National Institutes of Health Grant U01 AG024904) and DOD ADNI (Department of Defense award number W81XWH-12-2-0012). ADNI is funded by the National Institute on Aging, the National Institute of Biomedical Imaging and Bioengineering, and through generous contributions from the following: AbbVie, Alzheimer's Association; Alzheimer's Drug Discovery Foundation; Araclon Biotech; BioClinica, Inc.; Biogen; Bristol-Myers Squibb Company; CereSpir, Inc.; Cogstate; Eisai Inc.; Elan Pharmaceuticals, Inc.; Eli Lilly and Company; EuroImmun; F. Hoffmann-La Roche Ltd and its affiliated company Genentech, Inc.; Fujirebio; GE Healthcare; IXICO Ltd.; Janssen Alzheimer Immunotherapy Research & Development, LLC.; Johnson & Johnson Pharmaceutical Research & Development LLC.; Lumosity; Lundbeck; Merck & Co., Inc.; Meso Scale Diagnostics, LLC.; NeuroRx Research; Neurotrack Technologies; Novartis Pharmaceuticals Corporation; Pfizer Inc.; Piramal Imaging; Servier; Takeda Pharmaceutical Company; and Transition Therapeutics. The Canadian Institutes of Health Research is providing funds to support ADNI clinical sites in Canada. Private sector contributions are facilitated by the Foundation for the National Institutes of Health (www.fnih.org). The grantee organization is the Northern California Institute for Research and Education, and the study is coordinated by the Alzheimer's Therapeutic Research Institute at the University of Southern California. ADNI data are disseminated by the Laboratory for Neuro Imaging at the University of Southern California.

## Author contributions

Julie Gonneaud: study concept and design, data acquisition and processing, statistical analysis, interpretation of the results, writing the manuscript. Alex T. Baria: study concept and design, data acquisition and processing, neural net development, interpretation of the results, revising the manuscript for intellectual content. Alexa Pichet Binette: data acquisition, data processing, revising the manuscript for intellectual content. Brian A. Gordon: study concept and design, revising the manuscript for intellectual content. Jasmeer P. Chhatwal: study concept and design, revising the manuscript for intellectual content. Carlos Cruchaga: study concept and design, revising the manuscript for intellectual content. Mathias Jucker: study concept and design, revising the manuscript for intellectual content. Johannes Levin: study concept and design, revising the manuscript for intellectual content. Stephen Salloway: study concept and design, revising the manuscript for intellectual content. Martin Farlow: study concept and design, revising the manuscript for intellectual content. Serge Gauthier: study concept and design, revising the manuscript for intellectual content. Tammie L.S. Benzinger: study concept and design, revising the manuscript for intellectual content. John C. Morris: study concept and design, revising the manuscript for intellectual content. Randall J. Bateman: study concept and design, revising the manuscript for intellectual content. John C.S. Breitner: study concept and design, revising the manuscript for intellectual content. Judes Poirier: study concept and design, revising the manuscript for intellectual content. Etienne Vachon-Presseau: study concept and design, interpretation of the results, revising the manuscript for intellectual content. Sylvia Villeneuve: study concept and design, interpretation of the results, revising the manuscript for intellectual content. ADNI, DIAN, and PREVENT-AD provided the data used for this study.

## Competing interests

The authors declare no competing interests.

## Additional information

# Alzheimer's Disease Neuroimaging Initiative (ADNI)

Michael Weiner[12], Howard J. Rosen[12], Bruce L. Miller[12], Paul Aisen[13], Ronald G. Thomas[13], Michael Donohue[13], Sarah Walter[13], Devon Gessert[13], Tamie Sather[13], Gus Jiminez[13], Ronald Petersen[14], Clifford R. Jack Jr.[14],

Matthew Bernstein[14], Bret Borowski[14], Jeff Gunter[14], Matt Senjem[14], Prashanthi Vemuri[14], David Jones[14], Kejal Kantarci[14], Chad Ward[14], Sara S. Mason[14], Colleen S. Albers[14], David Knopman[14], Kris Johnson[14], William Jagust[15], Susan Landau[15], John Q. Trojanowki[16], Arthur W. Toga[17], Karen Crawford[17], Scott Neu[17], Laurel Beckett[18], Danielle Harvey[18], Charles DeCarli[18], Robert C. Green[19], Andrew J. Saykin[20], Tatiana M. Foroud[20], Li Shen[20], Faber Kelley[20], Sungeun Kim[20], Kwangsik Nho[20], Martin R. Farlow[8,20], Ann Marie Hake[20], Brandy R. Matthews[20], Scott Herring[20], Cynthia Hunt[20], Nigel J. Cairns[21], Erin Householder[21], Lisa Taylor Reinwald[21], Leslie M. Shaw[22], Steven E. Arnold[22], Jason H. Karlawish[22], David Wolk[22], Enchi Liu[23], Tom Montine[24], Nick Fox[25], Paul Thompson[26], Norbert Schuff[27], Robert A. Koeppe[28], Judith L. Heidebrink[28], Joanne L. Lord[28], Norm Foster[29], Eric M. Reiman[30], Kewei Chen[30], Adam Fleisher[30], Pierre Tariot[30], Stephanie Reeder[30], Chet Mathis[31], Oscar L. Lopez[31], MaryAnn Oakley[31], Donna M. Simpson[31], Virginia Lee[32], Magdalena Korecka[32], Michal Figurski[32], Steven Potkin[33], Zaven Kachaturian[34], Richard Frank[35], Peter J. Snyder[36], Susan Molchan[37], Jeffrey Kaye[38], Joseph Quinn[38], Betty Lind[38], Raina Carter[38], Sara Dolen[38], Lon S. Schneider[39], Sonia Pawluczyk[39], Mauricio Beccera[39], Liberty Teodoro[39], Bryan M. Spann[39], James Brewer[40], Helen Vanderswag[40], Rachelle S. Doody[41], Javier Villanueva Meyer[41], Munir Chowdhury[41], Susan Rountree[41], Mimi Dang[41], Yaakov Stern[42], Lawrence S. Honig[42], Karen L. Bell[42], John Morris[3,21,43], Beau Ances[43], Maria Carroll[43], Sue Leon[43], Mark A. Mintun[43], Stacy Schneider[43], Angela OliverNG[44], Randall Griffith[44], David Clark[44], David Geldmacher[44], John Brockington[44], Erik Roberson[44], Hillel Grossman[45], Effie Mitsis[45], Leyla deToledo-Morrell[46], Raj C. Shah[46], Ranjan Duara[47], Daniel Varon[47], Maria T. Greig[47], Peggy Roberts[47], Marilyn Albert[48], Chiadi Onyike[48], Daniel D'Agostino II[48], Stephanie Kielb[48], James E. Galvin[49], Dana M. Pogorelec[49], Brittany Cerbone[49], Christina A. Michel[49], Henry Rusinek[49], Mony J. de Leon[49], Lidia Glodzik[49], Susan De Santi[49], P. Murali Doraiswamy[50], Jeffrey R. Petrella[50], Terence Z. Wong[50], Charles D. Smith[51], Greg Jicha[51], Peter Hardy[51], Partha Sinha[51], Elizabeth Oates[51], Gary Conrad[51], Anton P. Porsteinsson[52], Bonnie S. Goldstein[52], Kim Martin[52], Kelly M. Makino[52], M. Saleem Ismail[52], Connie Brand[52], Ruth A. Mulnard[53], Gaby Thai[53], Catherine Mc Adams Ortiz[53], Kyle Womack[54], Dana Mathews[54], Mary Quiceno[54], Ramon Diaz Arrastia[54], Richard King[54], Myron Weiner[54], Kristen Martin Cook[54], Michael DeVous[54], Allan I. Levey[55], James J. Lah[55], Janet S. Cellar[55], Jeffrey M. Burns[56], Heather S. Anderson[56], Russell H. Swerdlow[56], Liana Apostolova[57], Kathleen Tingus[57], Ellen Woo[57], Daniel H. S. Silverman[57], Po H. Lu[57], George Bartzokis[57], Neill R. Graff Radford[58], Francine ParfittH[58], Tracy Kendall[58], Heather Johnson[58], Christopher H. van Dyck[59], Richard E. Carson[59], Martha G. MacAvoy[59], Howard Chertkow[60], Howard Bergman[60], Chris Hosein[60], Sandra Black[61], Bojana Stefanovic[61], Curtis Caldwell[61], Ging Yuek Robin Hsiung[62], Howard Feldman[62], Benita Mudge[62], Michele Assaly Past[62], Andrew Kertesz[63], John Rogers[63], Dick Trost[63], Charles Bernick[64], Donna Munic[64], Diana Kerwin[65], Marek Marsel Mesulam[65], Kristine Lipowski[65], Chuang Kuo Wu[65], Nancy Johnson[65], Carl Sadowsky[66], Walter Martinez[66], Teresa Villena[66], Raymond Scott Turner[67], Kathleen Johnson[67], Brigid Reynolds[67], Reisa A. Sperling[4], Keith A. Johnson[4], Gad Marshall[4], Meghan Frey[4], Jerome Yesavage[68], Joy L. Taylor[68], Barton Lane[68], Allyson Rosen[68], Jared Tinklenberg[68], Marwan N. Sabbagh[69], Christine M. Belden[69], Sandra A. Jacobson[69], Sherye A. Sirrel[69], Neil Kowall[70], Ronald Killiany[70], Andrew E. Budson[70], Alexander Norbash[70], Patricia Lynn Johnson[70], Thomas O. Obisesan[71], Saba Wolday[71], Joanne Allard[71], Alan Lerner[72], Paula Ogrocki[72], Leon Hudson[72], Evan Fletcher[73], Owen Carmichael[73], John Olichney[73], Smita Kittur[74], Michael Borrie[75], T. Y. Lee[75], Rob Bartha[75], Sterling Johnson[76], Sanjay Asthana[76], Cynthia M. Carlsson[76], Steven G. Potkin[77], Adrian Preda[77], Dana Nguyen[77], Vernice Bates[78], Horacio Capote[78], Michelle Rainka[78], Douglas W. Scharre[79], Maria Kataki[79], Anahita Adeli[79], Earl A. Zimmerman[80], Dzintra Celmins[80], Alice D. Brown[80], Godfrey D. Pearlson[81],

Karen Blank[81], Karen Anderson[81], Robert B. Santulli[82], Tamar J. Kitzmiller[82], Eben S. Schwartz[82], Kaycee M. Sinks[83], Jeff D. Williamson[83], Pradeep Garg[83], Franklin Watkins[83], Brian R. Ott[84], Henry Querfurth[84], Geoffrey Tremont[84], Stephen Salloway[7], Paul Malloy[7], Stephen Correia[7], Jacobo Mintzer[85], Kenneth Spicer[85], David Bachman[85], Elizabether Finger[86], Stephen Pasternak[86], Irina Rachinsky[86], Dick Drost[86], Nunzio Pomara[87], Raymundo Hernando[87], Antero Sarrael[87], Susan K. Schultz[88], Laura L. Boles Ponto[88], Hyungsub Shim[88], Karen Elizabeth Smith[88], Norman Relkin[89], Gloria Chaing[89], Lisa Raudin[89], Amanda Smith[90], Kristin Fargher[90] & Balebail Ashok Raj[90]

[12]UC San Francisco, San Francisco, CA, USA. [13]UC San Diego, San Diego, CA, USA. [14]Mayo Clinic, Rochester, NY, USA. [15]UC Berkeley, Berkeley, CA, USA. [16]University of Pennsylvania, Pennsylvania, CA, USA. [17]USC, Los Angeles, CA, USA. [18]UC Davis, Davis, CA, USA. [19]Brigham and Women's Hospital, Harvard Medical School, Boston, MA, USA. [20]Indiana University, Bloomington, IN, USA. [21]Washington University St Louis, St Louis, MO, USA. [22]University of Pennsylvania, Philadelphia, PA, USA. [23]Janssen Alzheimer Immunotherapy, South San Francisco, CA, USA. [24]University of Washington, Seattle, WA, USA. [25]University College London, London, UK. [26]USC School of Medicine, Los Angeles, CA, USA. [27]UCSF MRI, San Francisco, CA, USA. [28]University of Michigan, Ann Arbor, MI, USA. [29]University of Utah, Salt Lake City, UT, USA. [30]Banner Alzheimer's Institute, Phoenix, AZ, USA. [31]University of Pittsburgh, Pittsburgh, PA, USA. [32]UPenn School of Medicine, Philadelphia, PA, USA. [33]UC Irvine, Newport Beach, CA, USA. [34]Khachaturian, Radebaugh & Associates Inc and Alzheimer's Association's Ronald and Nancy Reagan's Research Institute, Chicago, IL, USA. [35]General Electric, Boston, MA, USA. [36]Brown University, Providence, RI, USA. [37]National Institute on Aging/National Institutes of Health, Bethesda, MD, USA. [38]Oregon Health and Science University, Portland, OR, USA. [39]University of Southern California, Los Angeles, CA, USA. [40]University of California San Diego, San Diego, CA, USA. [41]Baylor College of Medicine, Houston, TX, USA. [42]Columbia University Medical Center, New York, NY, USA. [43]Washington University, St Louis, MO, USA. [44]University of Alabama Birmingham, Birmingham, MO, USA. [45]Mount Sinai School of Medicine, New York, NY, USA. [46]Rush University Medical Center, Chicago, IL, USA. [47]Wien Center, Vienna, Austria. [48]Johns Hopkins University, Baltimore, MD, USA. [49]New York University, New York, NY, USA. [50]Duke University Medical Center, Durham, NC, USA. [51]University of Kentucky, Lexington, NC, USA. [52]University of Rochester Medical Center, Rochester, NY, USA. [53]University of California, Irvine, CA, USA. [54]University of Texas Southwestern Medical School, Dallas, TX, USA. [55]Emory University, Atlanta, GA, USA. [56]University of Kansas, Medical Center, Lawrence, KS, USA. [57]University of California, Los Angeles, CA, USA. [58]Mayo Clinic, Jacksonville, FL, USA. [59]Yale University School of Medicine, New Haven, CT, USA. [60]McGill University, Montreal Jewish General Hospital, Montreal, WI, USA. [61]Sunnybrook Health Sciences, Toronto, ON, Canada. [62]UBC Clinic for AD & Related Disorders, Vancouver, BC, Canada. [63]Cognitive Neurology St Joseph's, Toronto, ON, Canada. [64]Cleveland Clinic Lou Ruvo Center for Brain Health, Las Vegas, NV, USA. [65]Northwestern University, Evanston, IL, USA. [66]Premiere Research Inst Palm Beach Neurology, West Palm Beach, FL, USA. [67]Georgetown University Medical Center, Washington, DC, USA. [68]Stanford University, Santa Clara County, CA, USA. [69]Banner Sun Health Research Institute, Sun City, AZ, USA. [70]Boston University, Boston, MA, USA. [71]Howard University, Washington, DC, USA. [72]Case Western Reserve University, Cleveland, OH, USA. [73]University of California, Davis Sacramento, CA, USA. [74]Neurological Care of CNY, New York, NY, USA. [75]Parkwood Hospital, Parkwood, CA, USA. [76]University of Wisconsin, Madison, WI, USA. [77]University of California, Irvine BIC, Irvine, CA, USA. [78]Dent Neurologic Institute, Amherst, MA, USA. [79]Ohio State University, Columbus, OH, USA. [80]Albany Medical College, Albany, NY, USA. [81]Hartford Hospital, Olin Neuropsychiatry Research Center, Hartford, CT, USA. [82]Dartmouth Hitchcock Medical Center, Albany, NY, USA. [83]Wake Forest University Health Sciences, Winston-Salem, NC, USA. [84]Rhode Island Hospital, Providence, RI, USA. [85]Medical University South Carolina, Charleston, SC, USA. [86]St Joseph's Health Care, Toronto, ON, Canada. [87]Nathan Kline Institute, Orangeburg, SC, USA. [88]University of Iowa College of Medicine, Iowa City, IA, USA. [89]Cornell University, Ithaca, NY, USA. [90]University of South Florida, USF Health Byrd Alzheimer's Institute, Tampa, FL, USA.

## Dominantly Inherited Alzheimer Network (DIAN) Study Group

Ricardo Allegri[91], Randy Bateman[3], Jacob Bechara[92], Tammie Benzinger[3], Sarah Berman[31], Courtney Bodge[7], Susan Brandon[3], William (Bill) Brooks[92], Jill Buck[20], Virginia Buckles[3], Sochenda Chea[58], Jasmeer Chhatwal[4], Patricio Chrem[91], Helena Chui[39], Jake Cinco[25], Carlos Cruchaga[3], Tamara Donahue[3], Jane Douglas[25], Noelia Edigo[91], Nilufer Erekin-Taner[58], Anne Fagan[3], Marty Farlow[20], Colleen Fitzpatrick[4], Gigi Flynn[3], Nick Fox[25], Erin Franklin[3], Hisako Fujii[93], Cortaiga Gant[3], Samantha Gardener[94], Bernardino Ghetti[20], Alison Goate[95], Jill Goldman[96], Brian Gordon[3], Neill Graff-Radford[58], Julia Gray[3], Alexander Groves[3], Jason Hassenstab[3], Laura Hoechst-Swisher[3], David Holtzman[3], Russ Hornbeck[3], Siri Houeland DiBari[6], Takeshi Ikeuchi[97], Snezana Ikonomovic[31], Clifford Jack[14], Gina Jerome[3], Mathias Jucker[5], Celeste Karch[3], Kensaku Kasuga[97], Takeshi Kawarabayashi[98], William (Bill) Klunk[31], Robert Koeppe[28], Elke Kuder-Buletta[99], Christoph Laske[99], Jae-Hong Lee[100], Allan Levey[55], Johannes Levin[6], Ralph Martins[94], Neal Scott Mason[31], Colin Masters[101], Denise Maue-Dreyfus[3], Eric McDade[3], Hiroshi Mori[93], John Morris[3], Akem Nagamatsu[102], Katie Neimeyer[96], James Noble[96], Joanne Norton[3], Richard Perrin[3], Marc Raichle[3], Alan Renton[95], John Ringman[39], Jee Hoon Roh[100], Stephen Salloway[7], Hiroyuki Shimada[93], Wendy Sigurdson[3], Hamid Sohrabi[94], Paige Sparks[4], Kazushi Suzuki[102], Kevin Taddei[94], Peter Wang[3], Chengjie Xiong[3] & Xiong Xu[3]

[91]Fundación para la Lucha contra las Enfermedades Neurológicas de la Infancia (FLENI) Instituto de Investigaciones Neurológicas Raúl Correa, Buenos Aires, Argentina. [92]Neuroscience Research Australia, Sydney, NSW, Australia. [93]Osaka City University, Osaka, Japan. [94]Edith Cowan University, Perth, WA, Australia. [95]Icahn School of Medicine at Mount Sinai, New York, NY, USA. [96]Columbia University, New York, NY, USA. [97]Niigata University, Niigata, Japan. [98]Hirosaki University, Hirosaki, Japan. [99]German Center for Neurodegenerative Diseases (DZNE), Tübingen, Germany. [100]Asan Medical Center, Seoul, Republic of Korea. [101]University of Melbourne, Melbourne, VIC, Australia. [102]Tokyo University, Tokyo, Japan.

## Pre-symptomatic Evaluation of Experimental or Novel Treatments for Alzheimer's Disease (PREVENT-AD) Research Group

Pierre Bellec[103], Véronique Bohbot[1], John C. S. Breitner[1], Mallar Chakravarty[1], Louis Collins[2], Pierre Etienne[1], Alan Evans[2], Serge Gauthier[1,104], Rick Hoge[2], Yasser Ituria-Medina[2], Vasavan Nair[1,104], Jamie Near[1], Judes Poirier [1], Natasha Rajah[1], Pedro Rosa-Neto[1,104], Christine Tardif[1], Jennifer Tremblay-Mercier[1], Etienne Vachon-Presseau[1,9,10,11] & Sylvia Villeneuve [1,2,105]✉

[103]Centre De Recherche De L'Institut Universitaire De Gériatrie De Montreal, Université de Montréal, Montreal, QC, Canada. [104]McGill University Research Centre For Studies In Aging, Montreal, QC, Canada.

