## [Peer Review File · Nature Communications]

Accelerated functional brain aging in pre-clinical familial Alzheimer's diseaseREVIEWER COMMENTS

Reviewer #1 (Remarks to the Author):

This study investigates functional brain changes in preclinical AD as indexed by a brain age prediction model based on graph theory metrics derived from rs-fMRI data. The authors first develop this model in a large dataset of pooled rs-fMRI data of cognitively healthy adult individuals from different cohorts that cover the adult lifespan. They then apply this model to predict brain age in cognitively normal individuals from an autosomal-dominant AD cohort (DIAN) and another cohort of individuals at increased risk for sporadic AD due to a family history of AD (PREVENT-AD). The difference between predicted brain age and chronological age is used as a measure of abnormal brain function and compared between genetic risk carriers and non-carriers (mutation in DIAN, APOE4 genotype in PREVENT-AD). In subsamples with available amyloid-PET data the association between global amyloid load/positivity with the predicted age difference is also investigated. The main finding of the study is that presymptomatic carriers of autosomal-dominant AD mutations show a higher predicted age difference than non-carriers, and that this is independent from amyloid pathology. In the PREVENT-AD cohort neither APOE4 genotype nor amyloid status/load was related to the predicted age difference.

The study question is quite novel and complements recent studies using brain age predictions as imaging biomarkers in AD, which have mostly focused on structural MRI data and clinically overt disease stages. A big strength of the study is the use of large scale imaging data across multiple cohorts spanning cognitively normal adult individuals across the whole age range and with varying risk factors for AD, including deterministic AD mutations and genetic risk factors for sporadic AD.

However, while interesting, in my opinion the study findings have to be considered of incremental nature rather than providing a fundamental advance to the field. Several previous studies have examined functional brain changes as measured by rs-fMRI in presymptomatic AD, including studies in presymptomatic mutation carriers as well as in APOE4- and/or amyloid-positive individuals. Predicted brain age is a relatively novel and interesting summary measure of abnormal brain aging that does not require any specific a priori hypothesis on disease-specific brain changes. However, this also means that, by definition, it is not specific for any particular disease. What is the unique value of brain age predictions over more standard rs-fMRI metrics (e.g. inter-regional connectivity or just the plain graph metrics directly) in the specific context of studying early functional brain changes in presymptomatic AD? In contrast to the current study's findings, both APOE4 genotype and amyloid status have been found to be associated with rs-fMRI changes in cognitively unimpaired individuals in several previous studies, which could actually indicate a relatively low sensitivity of the brain age prediction method for AD-specific functional brain changes.

Some more specific comments and questions on the methodology:

- Given that the training/validation datasets were based on cognitively normal individuals without further biomarker characterization it is not clear how well these datasets actually reflect "healthy/normal" aging, particularly in comparison to the cognitively normal PREVENT-AD cohort. Have the authors considered the possibility of excluding preclinical disease stages/at-risk groups (to the extent possible, e.g. APOE4- and/or amyloid-positive) from their training samples to try to get to a closer estimate of "healthy" brain aging. Such a model could then potentially also be more sensitive to early (preclinical) pathological changes.

- How was the ADNI data selected from the larger ADNI cohort? Is there any reason why ADNI data was just used for model training/validation, but not for studying the effects of APOE4 and amyloid-PET on predicted brain age?

- In previous publications the authors have used the years to estimated symptom onset as an estimate of presymptomatic disease progression in the PREVENT-AD cohort. Was this index associated with predicted brain age in the present study?

- It was not clear to me if, and how, the initial feature ranking and selection was included in the cross-validated model tuning. Also, by choosing the final model based on performance in the validation set doesn't this effectively bias the selected model to the specifics of this (relatively small and monocentric) dataset?
- What is the specific rationale for using global graph theory metrics in the machine learning model, compared to other commonly used rs-fMRI-derived metrics, such as e.g. simple inter-regional connectivity values?
- I also wonder whether a neural network architecture is really necessary for an input feature space that (maximally) includes 26 different (global) graph metrics. How do simpler statistical prediction techniques perform with this data? Also, one would expect some of the different metrics to be highly correlated, and it would be informative to see the covariance structure of the input features (maybe as supplementary material).
- The authors discuss the possibility of site effects in their analyses. However, model performance in the test data is only shown for the pooled dataset across several different cohorts. How did the model perform in the individual datasets separately, and especially in the healthy/non-risk portions of these datasets?
- In the results section it would also be helpful to report statistics for the deviance of the predicated brain age differences from zero, in addition to the (genetic and amyloid) group comparisons of this metric.
- While rs-fMRI may theoretically be more sensitive for detecting subtle functional brain changes, it is also known to be considerable more noisy than structural MRI and I think it should be mentioned in the discussion section that the actual advantage of rs-fMRI over structural MRI in the study of early brain changes in AD is far from being clear. In this context it would also be very helpful to know how the performance (R2, mean error,...) of the rs-fMRI/graph theory-based model compares to previously reported brain age prediction models based on structural MRI data (or different rs-fMRI metrics). I generally missed the mentioning of two recent seminal studies in the field of imaging-based brain age predictions and their relevance for neurodegenerative disease: (Kaufmann et al., 2019; Wang et al., 2019)
- The paper is generally well written and easy to follow, but there remain several typos in the text.

Kaufmann, T., van der Meer, D., Doan, N. T., Schwarz, E., Lund, M. J., Agartz, I., . . . Westlye, L. T. (2019). Common brain disorders are associated with heritable patterns of apparent aging of the brain. *Nat Neurosci*, 22(10), 1617-1623. doi:10.1038/s41593-019-0471-7

Wang, J., Knol, M. J., Tiulpin, A., Dubost, F., de Bruijne, M., Vernooij, M. W., . . . Roshchupkin, G. V. (2019). Gray Matter Age Prediction as a Biomarker for Risk of Dementia. *Proc Natl Acad Sci U S A*, 116(42), 21213-21218. doi:10.1073/pnas.1902376116

Reviewer #2 (Remarks to the Author):

Reviewer comments

This study reports the association of brain function connectivity measures based on resting-state functional MRI with age across a large dataset spanning 18-29

Strong aspects of this study are: the use of a large dataset, the use of a validation data set for model tuning, and an independent test dataset spanning all centers and ages for testing.

Questions/remarks:

Introduction/discussion:

-Trying to disentangle ageing effects from those caused by amyloid in early preclinical stages is also an important point to study, which deserves its own focus and this study seems well poised to study such an angle. However, the framing in this study of AD as a problem of accelerated ageing is in my view unhelpful to study this disabling disease, because aging on its own does not mean anything concrete, whereas in the case of AD there is a clear cause (i.e., amyloid aggregation) for neurodegeneration. Furthermore, no associations are found with the presence of amyloid aggregation, which suggests that the relationship between age and functional connectivity measures does not depend on amyloid, and thus possibly indicates a non-AD process. Of note, a previous study that also investigated ADNI resting-state functional connectivity did find a relationship of functional connectivity and amyloid load (see: Palmqvist S, Schöll M, Strandberg O, et al. Earliest accumulation of β -amyloid occurs within the default-mode network and concurrently affects brain connectivity. *Nature Communications*. 2017;8(1):608. doi:10.1038/s41467-017-01150-x), how could the lack of such a relationship in the present study be explained?

-Another issue contributing to the concept of brain-aging as studied presently, is that whether an individual is showing 'accelerated' or 'slow' aging is completely dependent on the group mean: In another group a person may be placed on either side of such a group mean and switch from accelerated to slow ageing.

-Furthermore, no assessment and relationships with cognitive data are presented, so it remains unclear whether these results contain any information that is relevant for cognitive functioning.

-The predicted ages have a very broad range, e.g., almost a third of 20 year olds have a predicted brain age of 40-60 years? What does that mean (biologically and cognitively)?

Methods:

-it is not so clear how differences between centers were dealt with, and how this may have influenced the results. For example, figure 2 shows a red blob for the PREVENT-AD study that shows a much less variability in predicted age, which is systematically higher than all other data points considered. It cannot be excluded that this is a center, rather than a family history effect.

-The use of independent training, validation and test sets is a strong aspect of this research.

However, selection criteria for sampling individuals for the training, validation and test datasets are unclear. For example:

- Why was about 80% of DIAN and CamCAN selected for training, and only 12% from PREVENT-AD? Why not balance the centers at least for training?

On what basis was the validation data set created? Why was data from 1 center only used, and why was this so small compared to the test data set?

-Why not make use of the notion that there are multiple centers and investigate dependencies of classification on center/scanner? Were there any systematic differences between centers in graph metrics used?

-What is the rationale for the selection of these 26 network measures, and why was the most simple measure (i.e., degree) not included, which influences many of these higher level metrics?

-Networks were thresholded to include only 5% of possible connections, did this result in disconnected networks? If so how was this dealt with?

-Please define in text 'resilience metric'.

-Graph metrics can be strongly correlated to each other, how does the neural net deal with multicollinearity issues? In multiple training rounds, what was the proportion of times that the same feature was selected for the selected model of 10 features with 5 units and 2 hidden layers? Please provide this info in a supplemental table

Other issues:

- Could the CDR or MMSE be added to the description of table 2 and possibly table 3 to indicate whether possibly individuals may have been included with mild cognitive impairment? P22 states - "In

order to reduce the number of inputs to the model and avoid circularity bias, we searched for the graph metrics that were most reliably predictive of age”: This sentence reads as if first a filtering step was performed to select features based on their dependency on the outcome, which variable is a circular approach and creates a bias. However, the text after that suggest that feature selection was part of the cross-validation step, in which features and model training are performed on *independent* data. Please clarify.

Reviewer #3 (Remarks to the Author):

This manuscript aims at developing a model able to predict brain aging from resting state functional connectivity. When applying the model to the AD subjects, they found that the pre-symptomatic phase of ADAD is characterized by accelerated functional brain aging. This is not surprise to us. These findings are further precede, detectable fibrillar A β deposition. The study uses a large sample size of healthy control (n=773) participants and the manuscript is well written. I congratulate the authors on these good results. While I think the methods and results are novel and interesting, I have several suggestions that could strengthen the results of the manuscript.

1. When I saw the MAE of the results, especially in the training dataset, it is a little bit large in comparison with previous studies. One possible reason is that the training data come from different sites, hence the MAE is a little large for these unnormalized datasets. It is really difficult for me to accept the post serise result based on this model with large MAE, for the differences between NC and AD might be caused the not right model.
2. In the method section, did the authors have tried any harmonization protocol, which might allows the analyses of data collected from different sites and therefore increases the sample size substantially. it would be better if the authors can consider how to harmonize the data from different sites? Or maybe you can consider to train the model by CAMCAN data for it is the largest with N>400. In that way, the authors can consider to validate the results in the other datasets, then investigate the difference between the MCI, AD and NC in ADNI and ADAD dataset
3. If I am right, I think ADNI has more than 100 subjects, why only 44 subjects were included in the present study. In that way, the results in figure 3 will be enriched by the data from ADNI. Also the results can be enriched by the longitude MCI subects in ADNI. I strongly suggest the authors can consider to add this part with more well fitted model.
4. The figures look great and are informative, but the text in figure 1 are too small to be legible and need to be increased.

Some papers for age prediction can be found:

<https://www.frontiersin.org/articles/10.3389/fnhum.2015.00418/full>

<https://science.sciencemag.org/content/329/5997/1358>

<https://pubmed.ncbi.nlm.nih.gov/30079125/>

Submission of the revised manuscript entitled “*Functional brain age prediction suggests accelerated aging in preclinical familial Alzheimer’s disease, irrespective of fibrillar amyloid-beta pathology*”, by Gonneaud et al., to be considered for publication in Nature Communications.

We would like to thank the reviewers for their insightful comments and suggestions. Below, we address the reviewers’ comments point-by-point.

Responses to reviews

Reviewer #1 (Remarks to the Author):

This study investigates functional brain changes in preclinical AD as indexed by a brain age prediction model based on graph theory metrics derived from rs-fMRI data. The authors first develop this model in a large dataset of pooled rs-fMRI data of cognitively healthy adult individuals from different cohorts that cover the adult lifespan. They then apply this model to predict brain age in cognitively normal individuals from an autosomal-dominant AD cohort (DIAN) and another cohort of individuals at increased risk for sporadic AD due to a family history of AD (PREVENT-AD). The difference between predicted brain age and chronological age is used as a measure of abnormal brain function and compared between genetic risk carriers and non-carriers (mutation in DIAN, APOE4 genotype in PREVENT-AD). In subsamples with available amyloid-PET data the association between global amyloid load/positivity with the predicted age difference is also investigated. The main finding of the study is that presymptomatic carriers of autosomal-dominant AD mutations show a higher predicted age difference than non-carriers, and that this is independent from amyloid pathology. In the PREVENT-AD cohort neither APOE4 genotype nor amyloid status/load was related to the predicted age difference.

The study question is quite novel and complements recent studies using brain age predictions as imaging biomarkers in AD, which have mostly focused on structural MRI data and clinically overt disease stages. A big strength of the study is the use of large scale imaging data across multiple cohorts spanning cognitively normal adult individuals across the whole age range and with varying risk factors for AD, including deterministic AD mutations and genetic risk factors for sporadic AD.

(Rev.1, Q. 1) However, while interesting, in my opinion the study findings have to be considered of incremental nature rather than providing a fundamental advance to the field. Several previous studies have examined functional brain changes as measured by rs-fMRI in presymptomatic AD, including studies in presymptomatic mutation carriers as well as in APOE4- and/or amyloid-positive individuals. Predicted brain age is a relatively novel and interesting summary measure of abnormal brain aging that does not require any specific a priori hypothesis on disease-specific brain changes. However, this also means that, by definition, it is not specific for any particular disease. What is the unique value of brain age

predictions over more standard rs-fMRI metrics (e.g. inter-regional connectivity or just the plain graph metrics directly) in the specific context of studying early functional brain changes in presymptomatic AD? In contrast to the current study's findings, both APOE4 genotype and amyloid status have been found to be associated with rs-fMRI changes in cognitively unimpaired individuals in several previous studies, which could actually indicate a relatively low sensitivity of the brain age prediction method for AD-specific functional brain changes.

Response: We agree that many studies evaluated the association between functional properties of the brain and Alzheimer's disease markers/predictors and that our study is of incremental nature for the field. Previous studies mainly focused on identifying the specific functional changes associated with amyloid burden or *APOE4* carriage in cognitively unimpaired individuals, most of the time by assessing group differences on a wide range of rs-fMRI measurements. They tend to show that amyloid burden and *APOE4* status are associated with rs-fMRI changes (e.g. DMN and salience networks hypoactivation or hyperactivation, changes in visual network connectivity, graph metrics value reductions; Schultz et al., *JNeurosci*, 2017; Brier et al, *Neurobiol Aging*, 2014; McKenna, *Brain Imaging Behav*, 2016; Quevenco, *Frontiers Aging Neurosci*, 2020; Drzezga et al., 2011; Elman et al., 2016; Myers et al., 2014; etc.). On the other hand, to our knowledge only a handful of studies evaluated rs-fMRI changes in autosomal dominant Alzheimer's disease (ADAD; Chhatwal et al., *Neurology*, 2013; Thomas et al., *JAMA Neurol*, 2014; Franzmeier et al., *Brain*, 2018; Chhatwal et al., *Brain*, 2018), and all considered the full ADAD spectrum (from asymptomatic to symptomatic carriers), which does not address the specific changes associated with preclinical ADAD. Only one of these studies (Chhatwal et al., *Neurology*, 2013), focusing on the DMN connectivity, reported the comparison between asymptomatic carriers and noncarriers (showing group differences in precuneus/posterior cingulate and right lateral parietal cortex connectivity, but not in the other sections of the DMN). Considering that ADAD is the only population in whom future conversion can be predicted with certainty (*APOE4* carriers or amyloid positive individuals' evolution remains uncertain), better characterization of brain changes in asymptomatic individuals with ADAD is particularly relevant for our understanding of preclinical AD.

Importantly, the goal of our study was not to test for group differences between individuals with and without preclinical AD, but to assess if amyloid or AD genetic risk are associated with more advanced brain aging (or biological aging). To do so we 1) created a model of brain age based on brain functional characteristics and 2) tested if cognitively normal individuals with amyloid or genetic predisposition to AD have advanced brain aging by computing the difference between their chronological and predicted age (PAD, predicted age difference). As pointed out by the reviewer and others (see Kaufmann et al., *Nature Neuroscience*, 2019), PAD is not specific to AD. That, however, does not mean it is not sensitive to AD, that it is not of interest, or that it represents a limitation of the current study. To draw a parallel, hippocampal atrophy is not specific to AD (Geuze et al., *Molecular Psychiatry*, 2005; Small et al., *Nature Reviews Neuroscience*, 2011) while being one of the most studied biomarkers of AD. Default mode network changes would be another example of a non-specific, but sensitive marker of AD.

Of note, the absence of robust associations between PAD and amyloid pathology does not question either the sensitivity of the brain age measure, nor the previous associations found using other approaches. This apparent inconsistency is merely due to the fact that the measures are of different nature (*i.e.* strength of the synchrony between brain regions' vs deviation from normal aging trajectory based on connectivity pattern) and therefore will reflect different processes. In addition to the differences in constructs, approaches are completely different as, while previous studies were specifically designed to find the connectivity metrics associated with amyloid or *APOE4*, we extracted a global metric of biological aging and explored whether or not it was influenced by amyloid and genetics (including *APOE4*). The existence of an effect of the ADAD mutation on functional brain age is, *per se*, the strongest evidence that functional brain age is sensitive to preclinical AD. The fact that we don't see strong associations with amyloid and *APOE4* suggest that brain age is not affected by these factors, which seems consistent with two previous studies. A previous study using grey matter to predict brain age in cognitively unimpaired older adults and AD patients showed no association between brain age and *APOE* in cognitively unimpaired older adults (Löwe et al., *PlosOne*, 2016). The same study showed that brain age was able to distinguish MCI who remained stable over time from those who progressed to dementia, confirming, despite the absence of association with *APOE4* in cognitively unimpaired individuals, that brain aging is affected in AD. Another study, using metabolic imaging to evaluate brain age (Goyal et al., *PNAS*, 2017), was able to develop a model of metabolic brain aging, but found no evidence of an association between this proxy of brain age and amyloid burden.

In addition, following the review and in order to assess the robustness of our main results, we added chronological age as a covariate in our main analyses (see also Kaufmann et al., *Nature Neuroscience*, 2019 or Wang et al., *PNAS*, 2019, for a similar procedure). Our main results remained similar, showing higher PAD in DIAN mutation carriers compared to non-carriers, and no influence of amyloid burden load within mutation carriers. However, the difference between DIAN non-carriers and amyloid-negative DIAN carriers became marginal (*because the amyloid-negative DIAN carriers were younger than the DIAN non-carriers and the amyloid-negative DIAN carriers*). Results in PREVENT-AD remained unchanged. Please see pages 10-12 of the manuscript for details. The results in DIAN amyloid-negative carriers led us to be more cautious in our interpretation as they suggest that some subtle associations might exist between advanced brain aging and amyloid pathology. The sentence "independently from amyloid accumulation" as now been removed from the title, abstract, and the discussion has been revised to be more moderate in the interpretation (see pages 15-16).

Overall, we believe our study is timely and offers new insights on biological aging in the context of preclinical AD. This has been clarified in the manuscript.

"Predicted brain age could be a particularly relevant biomarker in the context of aging and neurodegenerative diseases,² since it allows to identify who deviates from the "normal" aging trajectories and which factors influence these deviations. Based on neuroimaging data, brain age has thus the potential to provide insight on biological aging, to complement more usual measurements of brain integrity. Previous studies found that brain age is influenced by lifestyle factors¹¹⁻¹⁵ and other conditions, including AD dementia.¹⁶⁻¹⁸" Page 3

“Our main objective is to test if the disease influences brain aging trajectories early on, prior to dementia and if so, if it is associated with A β pathology.” Page 4

“Functional brain changes have been suggested to be a sensitive early marker of brain changes in various conditions, including AD.³⁴⁻³⁸ Studies on asymptomatic individuals notably suggest early changes in rs-fMRI in the preclinical phase of AD, showing changes in association with APOE4 carriage and A β accumulation.^{36,39-45} Rs-fMRI changes in preclinical ADAD remains, on the other hand, to be determined as most studies only considered the AD spectrum as a whole.⁴⁶⁻⁴⁹” Pages 4-5

“Studies employing rs-fMRI in this population are, nevertheless, relatively rare and all consider the whole ADAD spectrum,⁴⁶⁻⁴⁹ making it difficult to draw any conclusions on functional changes specific to the preclinical phase of ADAD. One of these studies, however, compared asymptomatic carriers and non-carriers, suggesting reduction in DMN functional connectivity in asymptomatic ADAD carriers.⁴⁷” Page 15

“When investigating the characteristics of PAD in individuals with a family history of sAD (PREVENT-AD), we did not find a difference between APOE4 carriers and non-carriers nor associations with A β burden (for similar results with structural and metabolic brain age, see^{16,65}). These findings are not necessarily in contradiction with the literature suggesting associations between APOE4 status, A β and rs-fMRI,^{36,39-45} but see 74,75 for contradictory results they just suggest that we are capturing different constructs. While the previous studies tested the direct effect of these two factors on rs-fMRI metrics, we are testing the associations between these AD risk factors and a proxy of biological aging derived from rs-fMRI. Both approaches provide complementary information with regard to functional brain changes in preclinical AD and brain aging.” Page 16

Some more specific comments and questions on the methodology:

- **(Rev.1, Q. 2)** Given that the training/validation datasets were based on cognitively normal individuals without further biomarker characterization it is not clear how well these datasets actually reflect “healthy/normal” aging, particularly in comparison to the cognitively normal PREVENT-AD cohort. Have the authors considered the possibility of excluding preclinical disease stages/at-risk groups (to the extent possible, e.g. APOE4-and/or amyloid-positive) from their training samples to try to get to a closer estimate of “healthy” brain aging. Such a model could then potentially also be more sensitive to early (preclinical) pathological changes.

Response: The reviewer raised an important point. We have, whenever possible, selected the training set data based on the absence of AD risk factors. Therefore, only DIAN mutation non-carriers were included in the training set and we limited the inclusion of PREVENT-AD participants to a small subset of APOE4 non-carriers. Once the model was tested (*i.e.* it was thus not possible to further tune it without transgressing our methodological plan (testing our model

only once in the left-out subjects)), a revision of the genotyping showed that two of the PREVENT-AD included in the training set were *APOE4* carriers. ADNI participants included in the training set were also selected based on their *APOE4* status, and only non-carriers were included in the training set.

Unfortunately, *APOE4* (or amyloid) status was not available in the CamCAN and ICBM studies. Thus, we had no possibility to ensure that none of the individuals in the training set had no genetic risk for AD or did not exhibit some levels of amyloid pathology. Nevertheless, based on previous estimates of *APOE4* and amyloid positivity prevalence, we expect no more than 20 to 30% of the older individuals to be *APOE4* carrier and/or amyloid positive. Thus, while we cannot rule out the inclusion of some individuals in the preclinical phase/at risk of AD in our training set, their influence should be minimal, with a lower prevalence compared to what has been used in previous brain age models; we are not aware of previous studies that have removed *APOE4* carriers and/or individuals with amyloid pathology when training their model.

The methods have been clarified regarding the dataset division and the potential inclusion of some individuals in the preclinical AD in the training set has been further acknowledged in the limitations section of the paper.

*“The dataset was divided so that 1) the training and validation sets were representative of healthy aging, 2) the training set benefited from a large amount of data, from different centers, in order to increase both model accuracy and generalizability to new datasets, and 3) all individuals with a known genetic risk of AD were included in the test set only, in particular DIAN mutation carriers (i.e. in the pre-symptomatic phase of ADAD) and PREVENT-AD participants (i.e. asymptomatic older adults at increased risk of AD due to family history), in whom the model’s prediction was tested to characterize brain age changes in preclinical AD. As a result, cognitively unimpaired individuals from the different cohorts were assigned randomly to the training set, except when their genetic status was available (DIAN, PREVENT-AD and ADNI), in which case only individuals with no genetic predisposition for AD were included in the training set; the remaining data (including individuals at increased risk of AD) was assigned to the test set. More specifically, mutation non-carriers from DIAN (~80% of DIAN non-carriers, randomly selected) were assigned to the training set, along with ADNI *APOE4* non-carriers, individuals from FCP-Cambridge, and ~80% of CamCAN participants (randomly selected). While the PREVENT-AD cohort has an increased risk of sAD, a few individuals from this cohort (~10%) were assigned to the training set to expose the model to this site’s characteristics; these individuals were randomly selected from the subsample of *APOE4* non-carriers (with the exception of two *APOE4* carriers that were accidentally included in the training set). ICBM was used as an independent sample of healthy individuals, which despite being of modest size covers the entire adulthood, to assess the generalizability of the brain age model to other datasets and reduce overfitting bias (validation set). Finally, the test set included our population of interest (DIAN mutation carriers, most PREVENT-AD participants) and the remaining asymptomatic individuals from the other cohorts (DIAN mutation non-carriers, CamCAN and ADNI *APOE4*-carrier participants). Importantly, each individual was*

included in only one of the three sets such that no overlap exists between the training, validation and test sets” Paged 6-7 and 21-22.

- (Rev.1, Q. 3) How was the ADNI data selected from the larger ADNI cohort? Is there any reason why ADNI data was just used for model training/validation, but not for studying the effects of APOE4 and amyloid-PET on predicted brain age?

Response: To assess brain aging in preclinical AD, we were interested in individuals either known to be in the preclinical phase of AD (ADAD mutation carriers) or at increased risk of AD due to a family history of sporadic AD. Therefore, we chose to focus on DIAN and PREVENT-AD. In the process of building a model of healthy brain aging, we aimed at including a large sample of rs-fMRI data covering the entire adulthood (18 to 90+), taking advantage of various datasets to increase its generalizability (see also response Rev. 3 comment 3). ADNI was included for this purpose. By limiting the inclusion of individuals at risk of AD in the training set (see previous comment), all ADNI *APOE4* noncarriers were assigned to the training set, leaving only *APOE4* carriers in the test set.

We included in the current study participants from ADNI2 which resulted in 155 participants with at least one rs-fMRI scan from which only 49 were cognitively normal. Amongst these 49 individuals, the scans of 5 participants did not pass quality control and were excluded from the analyses (see Methods section for details). As a result, only 44 scans were available for this study.

Considering both the primary purpose of including ADNI in this study (increase training set size and variability) and the presence of *APOE4* carriers only in the few participants remaining in test set ADNI sample (n=15), we were unable to further assess the influence of amyloid or *APOE4* in the ADNI sample.

ADNI selection criteria were clarified in the manuscript.

“We gathered rs-fMRI data collected in 1624 cognitively unimpaired participants from 18 to 94 years old, provided by the Dominantly Inherited Alzheimer Network (DIAN), Pre-symptomatic Evaluation of Experimental or Novel Treatments for Alzheimer’s Disease (PREVENT-AD), Cambridge Centre for Ageing and Neuroscience (CamCAN), 1000-Functional Connectomes Project – Cambridge site (FCP-Cambridge), Alzheimer’s Disease Neuroimaging Initiative (ADNI) and International Consortium for Brain Mapping (ICBM) cohorts, to build a “brain age” predictive model. Considering our focus on the preclinical phase of AD, individuals with mild cognitive impairment (MCI) or AD dementia were excluded from the present study.

[...] More specifically, mutation non-carriers from DIAN (~80% of DIAN non-carriers, randomly selected) were assigned to the training set, along with ADNI APOE4 non-carriers, individuals from FCP-Cambridge, and ~80% of CamCAN participants (randomly selected). [...] Finally, the test set included our population of interest (DIAN mutation carriers, most PREVENT-AD participants) and the

remaining asymptomatic individuals from the other cohorts (DIAN mutation non-carriers, CamCAN and ADNI APOE4-carrier participants).” Pages 5-6

“The primary goal of ADNI has been to test whether serial MRI, PET, other biological markers, and clinical and neuropsychological assessment can be combined to measure the progression of MCI and early AD. Considering the focus on preclinical AD, MCI and demented patients were not eligible for the present study.” Page 20

- (Rev.1, Q. 4) In previous publications the authors have used the years to estimated symptom onset as an estimate of presymptomatic disease progression in the PREVENT-AD cohort. Was this index associated with predicted brain age in the present study?

Response: Estimated years to symptom onset (EYO) is often used to track disease progression in ADAD. We previously showed that this index is associated with increased amyloid burden and AD risk in individuals with a parental history of sporadic AD, even when controlling for the chronological age of the individuals (Villeneuve et al., *JAMA Neurol*, 2018; Gonneaud et al., *Neurology*, 2020; Vogel et al., *Brain*, 2018). Please note that the associations between EYO and amyloid burden or other brain metrics are relatively weak in the PREVENT-AD cohort and the absence of associations between EYO and PAD would not necessarily discredit the results of the present study. Even in ADAD, the association between EYO and other biomarkers are not expected to be striking since the participants are in the preclinical phase of the disease - the associations are usually found when taking advantage of the whole ADAD spectrum - and EYO is only an estimate of disease progression. Thus, it is reasonable to hypothesize that brain features tracking EYO may not be the same as those predicting chronological age. Nevertheless, this is an interesting point and as suggested by the reviewer, we assessed the association between EYO and the predicted brain age. We additionally explored the association between EYO and the predicted age difference (PAD) in DIAN and PREVENT-AD.

A trend was found between EYO and predicted brain age in the PREVENT-AD ($r=.13$, $p=.05$) such that individuals that had older predicted ages tended to also be closer to their expected age of onset. This result was expected since older participants are usually closer to the EYO. No such association was found in mutation carriers or non-mutation carriers in DIAN ($r=.03$, $p=.71$ and $r=.07$, $p=.73$, respectively).

Looking at the PAD, our brain marker of interest, we found a weak, but significant association with EYO in PREVENT-AD ($r=-.13$; $p=.05$). While it goes in the expected direction and shows a similar effect size, the same association did not reach significance in DIAN carriers ($r=.13$, $p=.16$) and noncarriers ($r=.23$, $p=.24$). Analyses were controlled for chronological age (which is what we did in our previous publications).

These results should be interpreted with caution: the association found in PREVENT-AD is extremely modest and of a strength similar to the one observed in DIAN mutation carriers (that did not reach significance).

These a priori unsatisfactory results (that are further corroborated with an absence of association with cognition, see response to Rev.2, Q. 3) might have different interpretations:

- 1- Brain aging is affected early (as shown by the effect of mutation status in DIAN), but might not further accelerate with the disease progression. Thus, our results could reflect an “advanced aging” in the preclinical phase of AD, more than an “accelerated aging”; as defined previously (see the review of Franke & Gaser, *Frontiers in Neurology*, 2019). The distinction between advanced versus accelerated brain aging is depicted on the figure on the right (Figure 1 - response), where advanced brain aging is represented by the orange slope, while accelerated brain aging is represented in red.

Figure 1 - response. From Franke & Gaser, *Frontiers in Neurology*, 2019

- 2- Alternatively, one possibility is that individuals showing both a large amount of amyloid and/or being close to their estimated years of onset, in addition to a substantially increased brain age, already developed symptoms and thus, were excluded from this study. Therefore, maybe only those with an increased risk of AD but who were able to maintain a certain brain age were able to remain cognitively unimpaired, biasing somehow the association in those closer to their expected age of onset (resilience mechanism,).

We now report analyses on the association with EYO, controlling for age, in the Supplementary and refer to the results in the manuscript. Also, “accelerated” aging has been replaced by “advanced” aging when appropriate, and the possible impact of the study design on the weak/lack of association has been mentioned.

“Finally, we explored the association between PAD and other measures of disease progression, such as the estimated years to symptom onset or cognition (see Supplementary for details). Briefly, estimated years from expected symptom onset (EYO) has been widely used as an estimate of disease progression in DIAN^{21,51} and has been associated with amyloid pathology in individuals with a parental history of sporadic AD.⁵²⁻⁵⁴ This index, calculated as the difference between parental age at symptom onset and participant’s chronological age represent an estimation of their proximity to symptom onset. A weak but positive association was found between EYO and PAD in the PREVENT-AD ($r=.13$, $p=.05$), such that individuals that had higher PAD tended to also be closer to their expected age of onset. No such association was found in DIAN mutation carriers, or non-carriers. No association was found between PAD and cognition, in either the PREVENT-AD or DIAN cohort.” Page12

“While we evidenced an increased brain age in ADAD pre-symptomatic mutation carriers, the absence of robust associations between brain age and EYO or cognition might suggest that functional brain age is affected early on in the disease process,

but does not further accelerate with disease progression. Therefore, preclinical AD could be characterized by an “advanced aging”, more than an “accelerated aging”. Alternatively, another explanation of this a priori negative result could be that individuals showing both a large amount of amyloid and/or being close to their EYO, in addition to a substantially increased brain age, already developed symptoms and thus, were excluded from this study. Therefore, maybe only those with an increased risk of AD but who were able to maintain a certain brain age were able to remain cognitively unimpaired. While the association between EYO and PAD in PREVENT-AD is encouraging, suggesting that individuals closer to their expected age of symptom onset might have a more advanced brain aging, the relatively modest size of this effect, that is isolated (i.e., not replicated in DIAN or using cognition), prevent us from making any strong conclusion. Overall, the interpretation of these specific findings remains uncertain and highly hypothetical. Further studies, specifically designed to address this question are needed to interpret the interplay between functional brain aging and indices of disease progression.” Pages 17-18

“Estimated years to onset and cognitive assessments
Estimated years from expected symptom onset (EYO) was calculated in each cohort taking the parental age at onset as a reference (see Supplementary for details).
Extensive cognitive evaluation was performed both in DIAN and PREVENT-AD, allowing notably to evaluate global functioning and episodic memory (immediate and delayed recall; see Supplementary for details).” Page 31

“Exploratory analyses were finally conducted to assess the correlation between the PAD and estimated years to onset and cognition. The analyses on cognition were further controlled for education.” Page 31

- **(Rev.1, Q. 5) It was not clear to me if, and how, the initial feature ranking and selection was included in the cross-validated model tuning.**

Response: We thank the reviewer for pointing out this confusing point. Initial feature ranking was not used to cross-validate the neural network model. It was only an initial step performed within the train set to determine the order of the features selected and entered for training the subsequent neural networks (the neural net was trained with between 5 and 25 features and these features were ordered and entered based on their ranking). For example, for a network with only 5 inputs, we aimed to use the 5 features that were most likely to be strong indicators of age. These 5 features were determined by taking the conjunction from the initial SVM and tree ensemble feature ranking (see Figure 1 and Methods for details). We added an additional sentence to the methods and results sections to clarify this process:

“Feature rank determined which metrics would be inputs to the subsequent neural network models. The rationale of feature ranking was to select the metrics with the highest predictive values to reduce the number of features in the final model” Page 8

Also, by choosing the final model based on performance in the validation set doesn't this effectively bias the selected model to the specifics of this (relatively small and monocentric) dataset?

The validation set was used to select the architecture of the final model, but the models themselves were developed on the training data. We agree with the reviewer that using a larger, and potentially multicenter, lifespan validation set would have been optimal. Unfortunately, at the time of the study, such a dataset was not available to us. It is also important to mention that, while imperfect, the use of an external validation set constitutes an improvement compared to what is usually done in the field: it is an extra step toward model generalizability. Of note, performance on the training set is in a reasonable range, especially when considering that the highest performing architectures are likely to present overfitting (see Figure 1B of the manuscript). Furthermore, when assessing the model performance on the test set, *that has never been used to build or tune the model*, performance remains relatively high, suggesting good model generalizability to new data despite the small validation set.

We further discussed this limitation in the revised version of the manuscript.

"The neural network was optimized by i) generating different models using the training set and ii) evaluating which one provided the best generalizability to the validation set (i.e., avoid overfitting and give better prediction on an independent set). Overfitting happens when a model fits too well the data and noise of a specific dataset, making generalizability of the model to other data unlikely. The validation set helped us to determine the best trade-off between optimizing the age prediction in the training set and enabling a good generalizability." Page 8

"While being of modest size, this validation set represents a completely independent dataset that covers the entire adulthood. Of note, this external validation step is a major strength of this work. While we cannot rule out the possibility that a larger and multicenter validation cohort might have led to the selection of a slightly different network architecture, the fact that the rmse was close across the different sets, including the test set, suggests that the validation step outcome was appropriate." Page 13

- (Rev.1, Q. 6) What is the specific rationale for using global graph theory metrics in the machine learning model, compared to other commonly used rs-fMRI-derived metrics, such as e.g. simple inter-regional connectivity values?

Response: One main advantage of this approach was to reduce the number of inputs into our final model. In our initial analyses (not mentioned in the manuscript), we modeled brain age with inter-regional connectivity values with relative success, but they were difficult to interpret in that connections could be chosen randomly with little influence on performance. The advantage of graph metrics is that every connection is defined by other connections, which could lead to easier interrogation. We added a few sentences to the Discussion explaining our rationale:

“Of note, models based on using individual functional connections as model inputs are also possible. Such models, however, have been shown to require multiple dozens⁶⁴ or hundreds¹⁰ of functional connections, whose inter-relationships are not defined. If we aim to understand the network characteristics related to brain age, the advantage of a model based on graph metrics is that every connection is defined in relation to all others.” Page 14

- **(Rev.1, Q. 7)** I also wonder whether a neural network architecture is really necessary for an input feature space that (maximally) includes 26 different (global) graph metrics. How do simpler statistical prediction techniques perform with this data? Also, one would expect some of the different metrics to be highly correlated, and it would be informative to see the covariance structure of the input features (maybe as supplementary material).

Response: We chose the neural network because it outperformed simpler models. More particularly, when looking at the performance of SVM and tree ensemble, error was found to be higher with these methods (rmse = 16.45 and 16.08, respectively, compared to 13.89 for the neural net selected, or lower for more complex models, see Figure 1 for details).

We have now added these additional results covering the performance of SVM and tree ensemble models:

“The root mean squared error (rmse) of chronological to predicted age for SVM and the tree ensemble were 16.45 and 16.08, respectively.” Page 7

Multicollinearity of variables certainly can be an issue in determining variable importance. However, our feature ranking strategy for determining inputs into our neural network exhibited higher performance than choosing features randomly, suggesting our feature ranks were not heavily influenced by multicollinearity. Additionally, feature importance for both SVM and the tree ensemble were mostly in agreement, and it’s unlikely that multicollinearity would have had equal influence on these two very different algorithms. Nonetheless, we have added a paragraph in the discussion section covering the issue of multicollinearity:

“We think our approach to model feature selection is a step in the right direction towards interpretability, but multicollinearity in graph metrics may have influenced our initial analyses of feature importance with SVM and tree ensembles. We are encouraged however, that the 10 graph metrics deemed most important by these algorithms provided much lower error in our final neural network model, compared to when they were chosen randomly. This suggests that these 10 metrics are important for determining brain age and are viable targets for further study.” Page 14

The covariance structure of the input features was additionally added as supplementary material.

“For reference, the covariance matrix of the 10 selected graph metrics is presented as supplementary material (eFigure 1).” Page 9

eFigure 1: Correlation between the 10 graph metrics used as input in the neural network

- (Rev.1, Q. 8) The authors discuss the possibility of site effects in their analyses. However, model performance in the test data is only shown for the pooled dataset across several different cohorts. How did the model perform in the individual datasets separately, and especially in the healthy/non-risk portions of these datasets?

Response: In the lifespan CamCAN cohort [single site dataset; n=95], the model’s prediction was similar to what was found in the validation set ($R^2=.26$; $p<.001$; $RMSE=16.70$; $MAE=14.32$; see also Figure R1 below). The statistics for the CamCAN subgroup are now reported in the text.

“The model was also able to predict chronological age from functional brain properties in the test set ($R^2=.36$, $p < .0001$; $rmse=13.24$; $mae = 11.58$; Figure 2C). Notably, the same was true when restricting the analyses to the CamCAN cohort, considered as a lifespan dataset representative of healthy aging ($R^2=.26$; $p<.001$; $rmse=16.70$; $mae=14.32$).” page 9-10

Figure R1. Model's prediction in the CamCAN (test set)

The model performance for the other sub-cohorts are presented below, but not included in the manuscript. In fact, interpreting the prediction for the other sub cohorts independently is more problematic as, unlike the CamCAN, 1) they have restricted age range and, more importantly, 2) almost all include individuals selected for their risk of AD (DIAN mutation carriers being in the pre-symptomatic phase of AD, PREVENT-AD being at increased risk of AD due to a family history of AD and ADNI participants being all *APOE4* carriers). See also responses 5 and 8 to Reviewer 2 for additional consideration of cohort effects.

Model performance for each (sub-)cohort of the test set:

ADNI *APOE4* carriers (multiple sites): $R^2 = .14$; $p = .18$; RMSE = 6.47; MAE = 16.75

PREVENT-AD (single site): $R^2 = 0.001$; $p = .65$; RMSE = 5.37; MAE = 7.66

DIAN mutation carriers (multiple sites): $R^2 = .04$; $p = .32$; RMSE = 11.56; MAE = 15.58

DIAN noncarriers (multiple sites): $R^2 = .01$; $p = .29$; RMSE = 9.66; MAE = 17.16

- **(Rev.1, Q. 9)** In the results section it would also be helpful to report statistics for the deviance of the predicated brain age differences from zero, in addition to the (genetic and amyloid) group comparisons of this metric.

Response: We thank the reviewer for this suggestion. Of note, our model's slope is not equal to 1, meaning that the model's prediction deviates slightly from the chronological age (*i.e.* in the training performance, younger ages tend to be overestimated by the models while older ages are predicted to be younger than expected); therefore, a predicted brain age difference of zero does not (necessarily) reflect a "normal brain age" prediction. The potential existence of site effects additionally suggests that such comparison needs to be considered with caution.

Analyses indicate that both in PREVENT-AD and in DIAN mutation carriers, the predicted brain age difference was statistically higher than zero ($t_{255} = 11.52$; $p < .001$ and $t_{124} = 5.19$; $p < .004$; respectively), while it did not differ from zero in the DIAN non-carriers ($t_{28} = -1.01$; $p = .32$).

In order to ensure that the model's slope did not bias this result, we replicated these analyses by calculating the residuals instead of the PAD (taking the training set regression line as a

reference), and compared them to zero. Results were relatively consistent, showing that both in PREVENT-AD and in DIAN mutation carriers, residuals were statistically higher than zero ($t_{255} = 40.82$; $p < .001$ and $t_{124} = 2.90$; $p = .004$; respectively), while they tend to be lower than zero in DIAN non-carriers ($t_{28} = -1.97$; $p = .06$).

If these results are in the expected direction, we believe that adding them to the manuscript would be source of overinterpretation. Since we strongly encourage against interpreting the PAD value itself, we prefer not to include these results.

We now report the statistics for the deviance from zero and recommend using this index only to compare brain aging different within the same cohort.

“PAD deviation from zero should not be interpreted in isolation due to the potential existence of site/cohort effects; thus, we will only interpret group comparisons within cohorts.” page 10

“As a result, the model output needs to be interpreted with caution and should not be interpreted on its own as reflecting some biological or cognitive state. Rather, the model output should be used to compare brain age characteristics between groups from the same site, as detailed below.” Page 15

- **(Rev.1, Q. 10)** While rs-fMRI may theoretically be more sensitive for detecting subtle functional brain changes, it is also known to be considerable more noisy than structural MRI and I think it should be mentioned in the discussion section that the actual advantage of rs-fMRI over structural MRI in the study of early brain changes in AD is far from being clear. In this context it would also be very helpful to know how the performance (R^2 , mean error,...) of the rs-fMRI/graph theory-based model compares to previously reported brain age prediction models based on structural MRI data (or different rs-fMRI metrics). I generally missed the mentioning of two recent seminal studies in the field of imaging-based brain age predictions and their relevance for neurodegenerative disease: (Kaufmann et al., 2019; Wang et al., 2019)

Response: We agree with the reviewer that rs-fMRI is noisier, or at least more variable, than structural MRI. Nevertheless, as pointed out by many studies, rs-fMRI has an incremental value and can allow studying/targeting processes that cannot be measured through structural MRI techniques. We also think that some previous works nicely demonstrate the value of rs-fMRI, or even its early sensitivity to the AD process (see for example the works of Iturria-Medina et al., *Nature Communications*, 2016 or Jones et al., *Brain*, 2015). Rs-fMRI functional connectivity has been also often used to predict AD pathology propagation, including amyloid (Buckner et al., *J. Neurosci*, 2009; Seeley et al., *Neuron*, 2009), suggesting that rs-fMRI changes might precede AD pathology. Thus, we believe that it is important to further characterize rs-fMRI-based brain age, as a complement to previous studies on structural brain aging.

Regarding the model performance, the reviewer is correct that our model has a lower accuracy compared to single site models using structural data (see a non-exhaustive list in the Table below, including notably seminal works on Brain Age). Rs-fMRI has been suggested to provide

lower accuracy when compared to structural MRI (see Dadi et al., *bioRxiv*, 2020; but see Liem et al., *NeuroImage*, 2017 for divergent results) and the use of multiple datasets is prone to increase in prediction error (Liem et al., *NeuroImage*, 2017; Orban et al., *Schizophrenia Research*, 2018). While using one single dataset increases model performance, it has the major disadvantage of lacking generalizability. By using multiple cohorts and an external validation set to define the best predictive model, we reduced the precision of our prediction, but avoided overfitting and improved generalizability, which is the main limitation of most published predictive models. Finally, recent findings suggested that moderately fitting models of brain age could provide a better differentiation of clinical groups compared to tightly-fitting models (Bashyam et al., *Brain*, 2020); suggesting that the lower performance of our model does not necessarily imply a lower ability/sensitivity to detect brain age changes in clinical populations.

This has been further discussed in the manuscript.

“Functional brain changes have been suggested to be a sensitive early marker of brain changes in various conditions, including AD.³⁴⁻³⁸ Studies on asymptomatic individuals notably suggest early changes in rs-fMRI in the preclinical phase of AD, showing changes in association with APOE4 carriage and A β accumulation.^{36,39-45} Rs-fMRI changes in preclinical ADAD remains, on the other hand, to be determined as most studies only considered the AD spectrum as a whole.⁴⁶⁻⁴⁹” Pages 4-5

“Nevertheless, it is important to acknowledge that the error provided by our model is higher than in previous studies.^{3,7,16,32,33,65} This might be due to both the use of rs-fMRI data, which is probably noisier and experiences more dynamic changes than structural data, and the inclusion of different datasets to build our model.^{7,66,67} While multisite models are known to have larger predictive errors, they have the advantage of being more generalizable to unseen data (to reduce the overfitting of the models). As a result, the model output needs to be interpreted with caution and should not be interpreted on its own as reflecting some biological or cognitive state. Rather, the model output should be used to compare brain age characteristics between groups from the same site, as detailed below.” Pages 14-15

Table – Response: Accuracy of Brain Age models

Study	Modality	Site	Age-range	Reported measure(s) of model performance			
				R	R ²	RMSE	MAE
Present manuscript							
Training set	rs-fMRI	Multiple	18-94		.53	14.01	11.00
Validation set		Single	19-79		.49	13.84	11.90
Test set		Multiple	18-90		.36	13.24	11.58
Kaufmann et al., Nature Neurosci , 2019	Struct. MRI	Multiple	3–89	.93(women) .94 (men)	-	-	-
Wang et al., PNAS , 2019	Struct MRI	Single	~46-96	.85	-	-	4.45
Franke et al., NeuroImage , 2010	Struct MRI	Multiple	20-86	.92	-	6.28	4.98
Cole et al., NeuroImage , 2017	Struct. MRI	Multiple	10-90	.57-.96 (range)	.32-.92 (range)	6.31-15.10 (range)	4.16-11.8 (range)
Liem et al., NeuroImage , 2017	rs-fMRI	Single + replication	19-82	-	.75-.80 (range)	-	5.25-5.99 (range)
	Struct. MRI			-	.62-.83 (range)	-	4.83-7.29 (range)
	Multimodal			-	.87	-	4.29
Goyal et al., PNAS , 2019	Metabolic PET (glucose, oxygen consumption, cerebral blood flow)	Multisite (2 cohorts with homogenised parameters)	20-82	.89	-	-	~5.4
Amen et al., JAD , 2018	SPECT	Multiple	0-105	-	.73	-	-
Dosenbach et al., Science , 2010	rs-fMRI	Single	6-35	-	.55	-	-
Lee et al., Proc IEEE Int Symp Biomed Imaging , 2018	rs-fMRI	Single	8-22	.36-.61 (range)			2.15-3.02 (range)

- (**Rev.1, Q. 11**) *The paper is generally well written and easy to follow, but there remain several typos in the text.;*

Response: We thank the reviewer for reporting such an issue. We paid attention to typos in the revised version of the manuscript.

*Kaufmann, T., van der Meer, D., Doan, N. T., Schwarz, E., Lund, M. J., Agartz, I., . . . Westlye, L. T. (2019). Common brain disorders are associated with heritable patterns of apparent aging of the brain. *Nat Neurosci*, 22(10), 1617-1623. doi:10.1038/s41593-019-0471-7*
*Wang, J., Knol, M. J., Tiulpin, A., Dubost, F., de Bruijne, M., Vernooij, M. W., . . . Roshchupkin, G. V. (2019). Gray Matter Age Prediction as a Biomarker for Risk of Dementia. *Proc Natl Acad Sci U S A*, 116(42), 21213-21218. doi:10.1073/pnas.1902376116*

References mentioned by the reviewers have been included in the manuscript.

Reviewer #2 (Remarks to the Author):

This study reports the association of brain function connectivity measures based on resting-state functional MRI with age across a large dataset spanning 18-29
Strong aspects of this study are: the use of a large dataset, the use of a validation data set for model tuning, and an independent test dataset spanning all centers and ages for testing.

Questions/remarks:

Introduction/discussion:

- (Rev.2, Q. 1) Trying to disentangle ageing effects from those caused by amyloid in early preclinical stages is also an important point to study, which deserves its own focus and this study seems well poised to study such an angle. However, the framing in this study of AD as a problem of accelerated ageing is in my view unhelpful to study this disabling disease, because aging on its own does not mean anything concrete, whereas in the case of AD there is a clear cause (i.e., amyloid aggregation) for neurodegeneration.

Furthermore, no associations are found with the presence of amyloid aggregation, which suggests that the relationship between age and functional connectivity measures does not depend on amyloid, and thus possibly indicates a non-AD process.

Of note, a previous study that also investigated ADNI resting-state functional connectivity did find a relationship of functional connectivity and amyloid load (see: Palmqvist S, Schöll M, Strandberg O, et al. Earliest accumulation of β -amyloid occurs within the default-mode network and concurrently affects brain connectivity. *Nature Communications*. 2017;8(1):608. doi:10.1038/s41467-017-01150-x), how could the lack of such a relationship in the present study be explained?

Response: We would like to apologize if our manuscript was misleading as our objective was not to reduce preclinical AD to a problem of accelerated aging. In addition, we do not interpret the lack of association with amyloid as the reflection of a non-AD process. While amyloid is a requirement for AD, amyloid-independent changes might still exist. There has been a whole research area suggesting the existence of early AD changes, independent from amyloid accumulation, or at least from detectable levels of amyloid pathology. Notably, the main model of AD cascade (Jack et al., 2010) has been updated in 2013 (Jack et al., 2013) to account for these changes appearing before amyloid increases, but at a subthreshold level. Higher neuronal activity has been found to promote amyloid secretions in transgenic mice (Bero et al., *Nature Neuro*, 2011). Our interpretation of the results is that individuals carrying a genetic mutation of AD have advanced brain aging that can be captured by functional brain features. Advanced brain aging might occur as early as (or even precede) amyloid accumulation, and when amyloid starts to accumulate, other pathological processes that are not captured by our brain age model are triggered. In our opinion, these give new insights on early brain dysfunction in individuals with ADAD, and supports the increasing evidence that functional brain abnormalities may precede amyloid accumulation (see Jagust & Mormino, *Trends in Cognitive Science*, 2011).

We revised the manuscript to clarify our objective and the complementarity of our results to previous approaches. See also Response 1 to Reviewer 1.

“Predicted brain age could be a particularly relevant biomarker in the context of aging and neurodegenerative diseases,² since it allows to identify who deviates from the “normal” aging trajectories and which factors influence these deviations. Based on neuroimaging data, brain age has thus the potential to provide insight on biological aging, to complement more usual measurements of brain integrity. Previous studies found that brain age is influenced by lifestyle factors^{11–15} and other conditions, including AD dementia.^{16–18}” Page 3

“Our objective is to test if the disease influences brain aging trajectories early on, prior to dementia and if so, if it is associated with A β pathology.” Page 4

Importantly, the existence of an effect of the ADAD mutation on functional brain age is, *per se*, the strongest evidence that functional brain age is affected in the preclinical phase of AD. In fact, while the presence of amyloid is one of the core features of AD, ADAD mutation is, to date, the only certain predictor of AD progression. As discussed previously (see also responses to Reviewer 1), the absence of robust evidence for an association between brain age and amyloid burden, or *APOE4* status, does not contradict previous studies. These previous studies were designed to detect the rs-fMRI changes associated with amyloid or *APOE4* (*i.e.* identify a “functional-signature” of amyloid-positivity or *APOE4* genotype). Here we adopted the inverse approach, developing a marker of biological aging and assessing whether it was associated with different indicators of preclinical AD, including ADAD mutation, amyloid and *APOE4*. Our results suggest that brain age is sensitive to ADAD mutation, but might be less sensitive to—or even independent from—*APOE4* and amyloid. Consistently, a previous study using grey matter to predict brain age across the AD spectrum (Löwe et al., *PlosOne*, 2016) showed that, while brain age was able to distinguish MCI who will remain stable over time from those who will progress (*i.e.* brain age was predictive of AD progression), it was not different between cognitively unimpaired *APOE4* carriers and non-carriers (*i.e.* brain age was not influenced by *APOE4* genotype). On the other hand, another study using metabolic imaging (Goyal et al., *PNAS*, 2017) was able to develop a model of metabolic brain aging, but failed to identify an association between this proxy of brain age and amyloid burden. More studies are needed to further understand the (lack of) association between brain age and markers such as amyloid burden and *APOE4*.

Considering the specific study of Palmqvist et al. (*Nature Communications*, 2017), the main reason of divergent results resides in the fact that we capture different mechanisms and use different approaches (see above), even if both studies include rs-fMRI. In addition, Palmqvist and collaborators used markers of amyloid burden that could be more sensitive to earlier stages of amyloidosis (notably CSF amyloid or voxel-wise amyloid-PET). We cannot rule out that some forms of amyloid pathology, that we were unable to capture, could be associated with our proxy of brain aging. To acknowledge for this pitfall, we paid attention to specify in the manuscript that our results applied to fibrillar amyloid.

Of importance, we now qualify our interpretation of the results in terms of amyloid-independent phenomenon and are more cautious on that point (see Reviewer 1 – Q1 for details).

In addition, we now more clearly discuss our results in relation to previous studies associating rs-fMRI and amyloid pathology.

“Functional brain changes have been suggested to be a sensitive early marker of brain changes in various conditions, including AD.³⁴⁻³⁸ Studies on asymptomatic individuals notably suggest early changes in rs-fMRI in the preclinical phase of AD, showing changes in association with APOE4 carriage and A β accumulation.^{36,39-45} Rs-fMRI changes in preclinical ADAD remains, on the other hand, to be determined as most studies only considered the AD spectrum as a whole.⁴⁶⁻⁴⁹” Pages 4-5

“Studies employing rs-fMRI in this population are, nevertheless, relatively rare and all consider the whole ADAD spectrum,⁴⁶⁻⁴⁹ making it difficult to draw any conclusions on functional changes specific to the preclinical phase of ADAD. One of these studies, however, compared asymptomatic carriers and non-carriers, suggesting reduction in DMN functional connectivity in asymptomatic ADAD carriers.⁴⁷” Page 15

“This suggests that the pre-symptomatic phase of AD is accompanied by advanced brain aging. The absence of association between PAD and A β burden (either considering fibrillar A β load or A β -positivity) suggests that the advanced brain aging evidenced in carriers cannot be entirely driven by the accumulation of A β . Nevertheless, the fact that A β -negative mutation carriers only tend to differ from the A β -negative non-carriers prevents us from making a definite conclusion regarding the independence between advanced functional brain aging and A β burden in preclinical ADAD. While we cannot exclude the fact that some A β negative individuals would in fact be A β accumulators⁷⁰⁻⁷² or present other forms of A β that cannot be detected through PET, this could also suggest that mutated genes have life-long effects on the brain that are not fully dependent on A β accumulation. [...] Altogether, this suggests that AD genetic mutations can influence brain properties early in life and that functional brain changes are among the earliest brain changes occurring in ADAD, and might occur partly independently from A β accumulation.^{see also 55.}” Pages 15-16

- (Rev.2, Q.2) Another issue contributing to the concept of brain-aging as studied presently, is that whether an individual is showing ‘accelerated’ or ‘slow’ aging is completely dependent on the group mean: In another group a person may be placed on either side of such a group mean and switch from accelerated to slow ageing.

Response: Unfortunately, this issue is intrinsic to all statistical analyses as effects are always evidenced in comparison to a specific group, considered as the reference group. The only way to overcome this issue is to have reference groups as large and representative as possible.

In the present study we tried to reduce as much as possible the cohort bias by building our model of brain age with multiple cohorts and assessing its generalizability in an independent validation set before applying it to the test set. By doing so, we intended to make our reference less dependent on a specific group mean. While this analytic choice reduced bias toward one group/cohort’s mean, it probably also has its limitations (see notably Response 10 to Reviewer 1).

The precautions adopted to increase the generalizability of the model has been now clarified in the manuscript.

“The dataset was divided so that 1) the training and validation sets were representative of healthy aging, 2) the training set benefited from a large amount of data, from different centers, in order to increase both model accuracy and generalizability to new datasets, and 3) all individuals with a known genetic risk of AD were included in the test set only, in particular DIAN mutation carriers (i.e. in the pre-symptomatic phase of ADAD) and PREVENT-AD participants (i.e. asymptomatic older adults at increased risk of AD due to family history), in whom the model’s prediction was tested to characterize brain age changes in preclinical AD. As a result, cognitively unimpaired individuals from the different cohorts were assigned randomly to the training set, except when their genetic status was available (DIAN, PREVENT-AD and ADNI), in which case only individuals with no genetic predisposition for AD were included in the training set; the remaining data (including individuals at increased risk of AD) was assigned to the test set. More specifically, mutation non-carriers from DIAN (~80% of DIAN non-carriers, randomly selected) were assigned to the training set, along with ADNI APOE4 non-carriers, individuals from FCP-Cambridge, and ~80% of CamCAN participants (randomly selected). While the PREVENT-AD cohort has an increased risk of sAD, a few individuals from this cohort (~10%) were assigned to the training set to expose the model to this site’s characteristics; these individuals were randomly selected from the subsample of APOE4 non-carriers (with the exception of two APOE4 carriers that were accidentally included in the training set). ICBM was used as an independent sample of healthy individuals, which despite being of modest size covers the entire adulthood, to assess the generalizability of the brain age model to other datasets and reduce overfitting bias (validation set). Finally, the test set included our population of interest (DIAN mutation carriers, most PREVENT-AD participants) and the remaining asymptomatic individuals from the other cohorts (DIAN mutation non-carriers, CamCAN and ADNI APOE4-carrier participants). Importantly, each individual was included in only one of the three sets such that no overlap exists between the training, validation and test sets” Paged 6-7 and 21-22.

- **(Rev.2, Q. 3) Furthermore, no assessment and relationships with cognitive data are presented, so it remains unclear whether these results contain any information that is relevant for cognitive functioning.**

Response: Of importance, all participants in the present study were cognitively unimpaired, as the objective was to focus on preclinical AD. Thus, variability in cognitive performance is expected to be very low. Therefore, we expected cognition to be associated with brain age prediction only in later stages of the AD process, and did not assess the association with cognition in the initial version of the manuscript.

We now investigated the association between the predicted brain age difference and 1) global functioning (MMSE or MoCA) and 2) episodic memory (immediate and delayed performance), controlling for age and education.

In DIAN mutation carriers, the predicted brain age difference (PAD) was not associated with global cognition (MMSE; $r=.14$; $p=.12$) or episodic memory (immediate list or story recall: $r=-.13$, $p=.17$ and $r=.03$, $p=.73$; delayed list or story recall: $r=.001$; $p=.99$, $r=-.04$, $p=.69$, respectively). The same results were found in DIAN non-carriers ($r=-.21$, $p=.32$; $r=.19$; $p=.36$, $r=-.09$, $p=.68$, $r=-.13$; $p=.54$, $r=-.15$, $p=.48$; for global cognition, immediate list recall, immediate story recall, delayed list recall, and delayed word recall, respectively)

In PREVENT-AD, the predicted brain age difference was not associated with global cognition (MoCA, $r=.09$, $p=.14$), immediate memory (immediate score from the RBANS, $r=.10$, $p=.10$), or delayed memory (delayed score from the RBANS, $r=.03$, $p=.65$).

The absence of association between PAD and cognition could be due to the abovementioned lack of variability in these population of cognitively intact individuals. Nevertheless, the absence of association also aligns with the lack of robust association between PAD and amyloid burden or estimated years to symptoms onset (EYO; see Response 4 to Reviewer 1), suggesting that the premature brain aging observed in DIAN mutation carriers could be an initial feature of the disease, which does not accompany disease progression (*i.e.* measured with increased amyloid, EYO or decrease in cognition). While this does not mean that an accelerated brain aging could not be captured later in the disease (*e.g.* with the appearance of cognitive decline), our current conclusion is that asymptomatic individuals with a genetic mutation responsible for ADAD show advanced brain age that is not strongly influenced by amyloid, disease progression (EYO) or cognition. Nevertheless, some studies evaluating structural brain age in MCI/AD patients suggest that brain aging might be more strongly related to the existence of an AD process (*i.e.* advanced brain age in MCI who will progress to AD compared to those who will not convert), than to cognition (*i.e.* no difference in brain age between MCI who will progress and AD patients; Franke & Gaser, *GeroPsych*, 2012; Löwe et al., *PlosOne*, 2016). These associations remain to be further studied to better understand how brain age relates to cognitive decline.

The cognitive analyses have been added in the Supplementary and further discussed in the main text:

“Finally, we explored the association between PAD and other measures of disease progression, such as the estimated years to symptom onset or cognition (see Supplementary for details). Briefly, estimated years from expected symptom onset (EYO) has been widely used as an estimate of disease progression in DIAN^{21,51} and has been associated with amyloid pathology in individuals with a parental history of sporadic AD.⁵²⁻⁵⁴ This index, calculated as the difference between parental age at symptom onset and participant’s chronological age represent an estimation of their proximity to symptom onset. A weak but positive association was found between EYO and PAD in the PREVENT-AD ($r=.13$, $p=.05$), such that individuals that had higher PAD tended to also be closer to their expected age of onset. No such association was found in DIAN mutation carriers, or non-carriers. No association was found between PAD and cognition, in either the PREVENT-AD or DIAN cohort.” Page12

“While we evidenced an increased brain age in ADAD pre-symptomatic mutation carriers, the absence of robust associations between brain age and EYO or cognition might suggest that functional brain age is affected early on in the disease process,

but does not further accelerate with disease progression. Therefore, preclinical AD could be characterized by an “advanced aging”, more than an “accelerated aging”. [...] While the association between EYO and PAD in PREVENT-AD is encouraging, suggesting that individuals closer to their expected age of symptom onset might have a more advanced brain aging, the relatively modest size of this effect, that is isolated (i.e., not replicated in DIAN or using cognition), prevent us from making any strong conclusion. Overall, the interpretation of these specific findings remains uncertain and highly hypothetical. Further studies, specifically designed to address this question are needed to interpret the interplay between functional brain aging and indices of disease progression.” Pages 17-18

“Estimated years to onset and cognitive assessments
Estimated years from expected symptom onset (EYO) was calculated in each cohort taking the parental age at onset as a reference (see Supplementary for details).
Extensive cognitive evaluation was performed both in DIAN and PREVENT-AD, allowing notably to evaluate global functioning and episodic memory (immediate and delayed recall; see Supplementary for details).” Page 31

“Exploratory analyses were finally conducted to assess the correlation between the PAD and estimated years to onset and cognition. The analyses on cognition were further controlled for education.” Page 31

- **(Rev.2, Q.4) The predicted ages have a very broad range, e.g., almost a third of 20 year old have a predicted brain age of 40-60 years? What does that mean (biologically and cognitively)?**

Response: There is indeed a large heterogeneity in brain predictions. We believe, however, that this heterogeneity is not specific to our prediction. Inter-individual heterogeneity is commonly found in brain aging studies and also, more largely in neuroimaging studies, even when looking at more “classic” measures such as grey matter volume (Pichet Binette et al., *Brain* 2020). Beyond the heterogeneity, the model’s prediction tends to be-shifted such that younger ages tend to be overestimated and older ages tend to be underestimated (see Figure 2). This is a common feature in brain age models, even though the degree of this shifting varies across studies. This phenomenon makes it particularly hazardous to try to give a biological meaning to these raw scores.

As a result, we paid attention not to draw conclusions on the predicted age difference *per se* (i.e. how it varies from zero), and only interpreted group differences in brain aging (See also Rev 1 – Q9). We also now covary all analyses for chronological age to prevent any bias in our analyses that may be due to changes in our model performance across age (see Kaufmann et al., *Nature Neuroscience*, 2019 or Wang et al., *PNAS*, 2019, for similar procedure).

The cognitive meaning of the index would be even harder to interpret, as cognition was not accounted for when building the index and all individuals were selected based on the absence of cognitive impairment.

We now state this more clearly in the manuscript.

“PAD deviation from zero should not be interpreted in isolation due to the potential existence of site/cohort effects; thus, we will only interpret group comparisons within cohorts. In addition, considering the tendency of the model to overestimate younger ages and under-estimate older ages, all subsequent analyses were controlled for chronological age.^{see 32 for similar procedure}.” Page 10

“As a result, the model output needs to be interpreted with caution and should not be interpreted on its own as reflecting some biological or cognitive state. Rather, the model output should be used to compare brain age characteristics between groups from the same site, as detailed below.” Page 15

Methods:

- **(Rev.2, Q.5)** it is not so clear how differences between centers were dealt with, and how this may have influenced the results. For example, figure 2 shows a red blob for the PREVENT-AD study that shows a much less variability in predicted age, which is systematically higher than all other data points considered. It cannot be excluded that this is a center, rather than a family history effect.

Response: Despite our efforts to construct a model that could be generalized (see responses to Reviewer 1), some site effects might still exist and, as already mentioned in the manuscript, it is likely to be the case in the PREVENT-AD dataset.

To increase the generalizability of our model to different centers, we included different cohorts/sites in the training set, and validated the model on an external dataset (different cohort and site). Previous studies showed that machine learning models trained on data collected from different sites drastically increased the prediction when predicting data from a new site, and this increase in prediction was proportional to the number of sites included in the model (see Orban et al., *Schizophrenia Research*, 2018). Thus, it is actually considered beneficial to use heterogenous data (*i.e.* multi-site data instead of single-site), to train fMRI-based classifiers.

We also ensured that all data were processed the exact same way for all subjects/sites to reduce inter-site disparities. Thus, all data went through the same pre-processing pipeline using NIAK (as detailed in the methods section) and followed previously published methods for mitigating site effects, such as normalizing and binarizing graphs. By doing so, we expected to reduce the variability caused by the different sites and limit site effects. It seems, however, that such effects might still persist. Accordingly, we advised against making any direct conclusion based on the predicted brain age difference between sites or cohorts. We recommend, instead, using this index to compare brain aging between individuals within the same cohort, or evaluate its association with other markers using data from the same study, as we did in the present paper.

Nevertheless, following the reviewer’s suggestion, we applied the ComBat harmonization method on the data (<https://github.com/Jfortin1/ComBatHarmonization/tree/master/Matlab>) to investigate how it could affect the results. ComBat uses empirical Bayes estimation to reduce differences between sites/batches (Fortin et al., *NeuroImage*, 2018; Yu et al., *HBM*, 2018; Pomponio et al., *NeuroImage*, 2020).

We first applied ComBat to the 26 graph metrics used to derive our brain age model by removing the site effects. We then applied the same feature selection procedure as in the main manuscript (*i.e.* ranking features based on the best age prediction using ensemble tree and SVM) using the harmonized graph metrics as input. The figure below (Figure 2 – response) shows the age prediction from SVM using the original metrics (panel A) vs. the harmonized metrics (panel B). We can clearly see that the harmonized metrics yield a worse age prediction. The reason may be because the site or cohort effects are inherently related to the different age composition of the cohorts (*e.g.* young adults only, lifespan dataset, older adults only), and thus harmonizing by sites blurred the age differences.

These analyses suggest that harmonization of the model inputs across cohorts would not be appropriate for the current study and we believe that the steps we used to mitigate site effects processing the data were sufficient and more appropriate. Importantly, as described above, evidence suggests that training, validating, and testing using heterogenous data collected from several sites actually drastically increases the generalizability of the predictions (see Orban et al., *Schizophrenia Research*, 2018).

Figure 2 - Response. Age prediction from support vector machine models using original graph metrics as input (A) and age prediction from support vector machine models using harmonized graph metrics from ComBat as input. (B)

The different steps adopted to mitigate site effects have been clarified in the manuscript.

“PAD deviation from zero should not be interpreted in isolation due to the potential existence of site/cohort effects; thus, we will only interpret group comparisons within cohorts..” Page 10

“To minimize the possibility of site effects, we included data from a variety of cohorts and sites, validated the model on a completely independent validation set (new site) and applied similar processing methods to all data in order minimize between-site effects.” Page 17

“In order to limit site effects, all functional images were processed in our laboratory (by APB) applying the exact same pipeline and processing steps..” Page 24

- **(Rev.2, Q.6)** The use of independent training, validation and test sets is a strong aspect of this research. However, selection criteria for sampling individuals for the training, validation and test datasets are unclear. For example:

- Why was about 80% of DIAN and CamCAN selected for training, and only 12% from PREVENT-AD? Why not balance the centers at least for training?

Response: We apologize if the sampling strategy was unclear. We clarified this in the revised version of the manuscript (see also response to Reviewer 1 – Q2 and Q3).

- Training sample was selected to represent healthy aging and, as much as possible, non-preclinical stage of AD. Thus, we only included DIAN non-carriers (*i.e.* individuals <65 years old with no genetic mutation responsible for ADAD). PREVENT-AD participants were all at increased risk of sporadic AD due to a family history of AD, as a result we did not want to include them in the training set. We nevertheless included a small proportion of these participants, randomly selected amongst *APOE4* noncarriers, in the training set so that specificity of this site could be accounted for in the model. At the end of the study (model trained, validated and applied to the test set), *APOE* genotypes were revised and 2 of the 36 participants were found to be *APOE4* carriers; but we expect their influence to be minimal (see response to Reviewer 1 – Q2). ADNI participants were selected to be *APOE4* non-carriers. FCP were all young cognitively unimpaired individuals so were included as training data. Finally, CamCAN being a lifespan dataset supposed to reflect healthy aging, in which we had no access on their risk of AD, individuals from this cohort were included randomly in the training set. With the objective of having a best prediction, we tried to include as many “healthy” participants as possible in the training set, explaining the large proportion (~80%) of DIAN noncarriers and CamCAN participants.
- The validation set was selected to be a lifespan dataset of healthy aging from a totally independent cohort and site, with the objective to assessing the generalizability of the model to completely distinct data.
- The test set included the remaining participants (a small portion of the DIAN noncarriers and CamCAN participants [~20% not included in the training set], ADNI *APOE4* carriers, the largest part of the PREVENT AD and all DIAN mutation carriers).

Overall, the sampling strategy was mainly driven by the idea of 1) including as many cognitively unimpaired individuals as possible (from different sites) in the training set and 2) keeping the individuals at risk of AD for the test set.

This rationale was further details in the methods section, see response to Rev.1, Q. 2 for specific details.

- **(Rev.2, Q. 7)** On what basis was the validation data set created? Why was data from 1 center only used, and why was this so small compared to the test data set?

Response: The designated validation set was used in addition to any k-fold cross-validation that was performed during training. We chose to use an external validation set from one site that was not included in the training data as an indicator of model performance on holdout data, to help determine the best model architecture for prediction generalizability (*i.e.*, better trade-off

between performance and overfitting) before applying it to the test set. We chose the ICBM set specifically for our extra validation set because all participants were healthy and the age range was one of the largest for any single site. The initial sample size was larger (n=86); unfortunately, only ~50% of them passed quality control, causing a reduction in the validation set sample size.

The rationale for the validation set selection has been now clarified (please also see response to Rev.1 - Q5 and Rev.2 - Q.6).

“ICBM was used as an independent sample of healthy individuals, which despite being of modest size covers the entire adulthood, to assess the generalizability of the brain age model to other datasets and reduce overfitting bias (validation set).” Page 7

“The neural network was optimized by i) generating different models using the training set and ii) evaluating which one provided the best generalizability to the validation set (i.e., avoid overfitting and give better prediction on an independent set). Overfitting happens when a model fits too well the data and noise of a specific dataset, making generalizability of the model to other data unlikely. The validation set helped us to determine the best trade-off between optimizing the age prediction in the training set and enabling a good generalizability.” Page 8

“While being of modest size, this validation set represents a completely independent dataset that covers the entire adulthood. Of note, this external validation step is a major strength of this work. While we cannot rule out the possibility that a larger and multicenter validation cohort might have led to the selection of a slightly different network architecture, the fact that the rmse was close across the different sets, including the test set, suggests that the validation step outcome was appropriate.” Page 13

- (Rev.2, Q. 8) Why not make use of the notion that there are multiple centers and investigate dependencies of classification on center/scanner? Where there any systematic differences between centers in graph metrics used?

Response: Rather than creating multiple models which are site-specific, our goal was to create the most generalizable model possible. Previous publications addressed this question and concluded that both increased sample size and inclusion of various sites to build models are mandatory for model precision and generalizability (see notably Liem et al., *NeuroImage*, 2017; Orban et al., *Schizophrenia Research*, 2018; Varoquaux, *NeuroImage*, 2018). We now better acknowledge that, despite our methodological precautions to increase generalizability, we cannot completely rule out site effects.

One issue with looking at the centers' differences in graphs metrics is that 1) demographics are strongly associated with centers, notably when considering their age-range and AD risk, and 2) there is a lot of variability in the sample size across sites (CamCAN, FCP and PREVENT-AD being single-site cohorts; while DIAN and ADNI sample were enrolled in 11 different sites [*i.e.* small sample per site]). It is thus particularly difficult to disentangle site from age effects in this context. Consistently, and as previously discussed (see response to Rev.2 – Q2), harmonizing

graph metrics between sites reduces age prediction accuracy, probably due to the fact that systematic differences between centers are due to age range/demographics differences and not scanner.

This point has been clarified in the manuscript.

“While multisite models are known to have larger predictive errors, they have the advantage of being more generalizable to unseen data (to reduce the overfitting of the models).” Page 14-15

“To minimize the possibility of site effects, we included data from a variety of cohorts and sites, validated the model on a completely independent validation set (new site) and applied similar processing methods to all data in order minimize between-site effects. Despite this effort, we cannot exclude the possibility that the age overestimation of most PREVENT-AD participants (Figure 2C) could be the result of a site effect.” Page 18

- **(Rev.2, Q. 9) What is the rationale for the selection of these 26 network measures, and why was the most simple measure (i.e., degree) not included, which influences many of these higher level metrics?**

Response: Because we wanted to use whole-brain metrics, we could not use degree, which is a region- or voxel-wise measure. We did, however, use resilience, which is a measure of the degree distribution across the brain. The 26 measures were chosen based on the availability of whole-brain metrics in the Matlab Brain Connectivity Toolbox.

- **(Rev.2, Q. 10) Networks were thresholded to included only 5% of possible connections, did this result in disconnected networks? If so how was this dealt with?**

Response: We are uncertain what the reviewer means by ‘disconnected networks’, but using a 5% link density ensures that the top 5% of connections were used in each scan, and by definition would not be possible to result in brain networks with no connections. We chose 5% because it has been shown across multiple studies that link densities between 1.5% and 10% give the most biologically plausible networks, and 5% is roughly midway through this range.

- **(Rev.2, Q.10) Please define in text ‘resilience metric’.**

Response: We have added the following sentences in the methods section to further define the “resilience” metric:

“In graph theory, resilience of network G is defined as the relative number of edges that must be removed for the network to lose property P , and is a measure of the

network's robustness to targeted or random attacks. Here, resilience is calculated as the slope of the log-log degree distribution." Page 26

- **(Rev.2, Q. 11)** Graph metrics can be strongly correlated to each other, how does the neural net deal with multi-collinearity issues? In multiple training rounds, what was the proportion of times that the same feature was selected for the selected model of 10 features with 5 units and 2 hidden layers? Please provide this info in a supplemental table

Response: We do not think that our model was strongly affected by multicollinearity issue, both because 1) our feature ranking strategy for determining inputs into our neural network exhibited higher performance than choosing features randomly and 2) feature importance for both SVM and the tree ensemble were mostly in agreement, and it's unlikely that multicollinearity would have had equal influence on these two very different algorithms. Please see Response to Rev.1, Q. 8 for further details.

We determined the number of features going into the neural network by first extracting the metrics' importance from the tree-based and SVM algorithms. We only used the most important metrics as input features into the neural network. Therefore, for models with 5 features, the same 5 features (*i.e.* the 5 most important, *except for the null model for which the 5 features were selected randomly to test the ranking strategy against random features*) were always used. For the models with 10 features, the same 10 features were always used, and so on. Because the proportion of times that the same features are selected for a given number of inputs is always 100%, a supplementary table does not seem necessary. However, we now clarified in the Results and Methods that each graph metric was included only once in the model:

"We built different neural networks with increasing complexity, varying in number of input features (5, 10, 15, 20, or 25 most-important graph metrics, according to the ranking determined previously), hidden layers, and hidden layer units. Importantly, each graph metric was only entered once as input for each neural network architecture tested, and the inputs were kept constant across the model's iterations: features of more complex models always included the features of the simpler ones." Page 8 (see also page 28)

Other issues:

- **(Rev.2, Q. 12)** Could the CDR or MMSE be added to the description of table 2 and possibly table 3 to indicate whether possibly individuals may have been included with mild cognitive impairment?

Response: All individuals were selected to be cognitively unimpaired. DIAN participants' status was based on a global CDR of 0. In PREVENT-AD, normal cognition was defined as CDR of 0 and a Montreal Cognitive Assessment (MoCA) ≥ 24 . In the few cases of ambiguous results (3 participants having a CDR of .5 and 1 participant with a MoCA of 23 in the present sample), participants were further evaluated with a more extensive neuropsychological test battery, which was carefully reviewed by neuropsychologists and physicians to ensure normal cognition.

We now specify this information in the text and include MMSE for DIAN (carriers MMSE = 29.02±1.27, range = 24-30; noncarriers MMSE = 29.36±1.03, range = 26-30), and MoCA for PREVENT-AD (MoCA = 28.11±1.52; range 23-30) in Table 2.

“Baseline data from cognitively unimpaired mutation carriers and non-carriers archived in the DIAN data freeze 10 (January 2009 to May 2016) were used in the present study. All selected individuals had a Clinical Dementia Rating (CDR)⁷⁷ scale of 0.” Page 18-19

“Inclusion criteria included i) being 60 or older; 55 to 59 for individuals who were less than 15 years from the age of their relative at symptom onset, ii) being cognitively normal and iii) no history of major neurological or psychiatric disease. Normal cognition was defined as CDR of 0 and a Montreal Cognitive Assessment (MoCA)⁷⁹ ≥ 24. In the few cases of ambiguous results (3 participants having a CDR of .5 and 1 participant with a MoCA of 23 in the present sample), participants were further evaluated with a more extensive neuropsychological test battery, which was carefully reviewed by neuropsychologists and physicians to ensure normal cognition” Page 19

- (Rev.2, Q. 13) P22 states - “In order to reduce the number of inputs to the model and avoid circularity bias, we searched for the graph metrics that were most reliably predictive of age”: This sentence reads as if first a filtering step was performed to select features based on their dependency on the outcome, which variable is a circular approach and creates a bias. However, the text after that suggest that feature selection was part of the cross-validation step, in which features and model training are performed on ***independent*** data. Please clarify.

Response: Thank you for bringing this lack of clarity to our attention. You are correct in that the first step is a filtering step: we used SVM and tree ensemble to get a sense of which metrics contribute more to age predictions only within the training set. However, we do not believe this is a circular approach because we performed this analysis only on the training data; our validation and final testing data had no influence on this filtering step. Actually, the error observed in the train was similar to the one observed in the validation and the test sets.

RMSE

Training set	14.01
Validation set	13.84
Test set	13.24

We have removed “*and avoid circularity bias*” from the quoted sentence. We also added a sentence in the methods section to clarify this point:

“Feature selection was not part of cross-validation.” Page 27

Reviewer #3 (Remarks to the Author):

This manuscript aims at developing a model able to predict brain aging from resting state functional connectivity. When applying the model to the AD subjects, they found that the pre-symptomatic phase of ADAD is characterized by accelerated functional brain aging. This is not surprise to us. These findings are further precede, detectable fibrillar A β deposition. The study uses a large sample size of healthy control (n=773) participants and the manuscript is well written. I congratulate the authors on these good results. While I think the methods and results are novel and interesting, I have several suggestions that could strengthen the results of the manuscript.

- (Rev.3, Q. 1) When I saw the MAE of the results, especially in the training dataset, it is a little bit large in comparison with previous studies. One possible reason is that the training data come from different sites, hence the MAE is a little large for these unnormalized datasets. It is really difficult for me to accept the post serise result based on this model with large MAE, for the differences between NC and AD might be caused the not right model.

Response: It is true that our model has a higher MAE than those in previous publications (for a non-exhaustive list, see the Table included in the response to Reviewer 1). While the reported measure of prediction accuracy varies from one study to another, it appears that our model provides generally lower accuracy. Such differences might be attributed to the fact that we 1) used rs-fMRI data, 2) included different cohorts in the training set and 3) added an external validation sample. The vast majority of previous papers (beyond the ones reported in the table) used structural data, single-site/cohort data and/or validated their model “only” through cross-validation. Of note, this lower accuracy might present some advantages for clinical group differentiation. In fact, recent findings suggest that moderately fitting brain age models provide more clinically-informative information than tightly-fitting models as they allowed a better differentiation of patients with MCI, AD, schizophrenia and depression (Bashyam et al., *Brain*, 2020). Thus, while counterintuitive, our model’s lower performance might not necessarily imply a lower sensitivity of our model.

First, the use of rs-fMRI could be a greater source of error when compared to structural data, because the measure is noisier. Previous studies on rs-fMRI tend to show lower accuracy when using rs-fMRI. While noisier, our study suggests that it is still possible to predict age from this modality. In addition, the inclusion of multiple datasets in our model is likely to have increased the error. Notably, publications by Liem et al. (*NeuroImage*, 2017) and by Orban et al (*Schizophrenia Research*, 2018) have shown that the use of multiple datasets, was prone to increase the prediction error, but the use of a single dataset dramatically reduces the applicability of the model to other datasets. Finally, using a validation set to define the best model has the beneficial effect of avoiding overfitting that, unfortunately, comes with the cost of reducing the accuracy of the prediction in the training set. By using only one dataset for training and validation or even testing (as done in several previous studies), it is likely that our prediction would have been better, but would have the major pitfall of being cohort-dependent and represent a lack of generalizability.

Thus, if we agree that the model's error is larger than in previous studies, which we now further acknowledge, this cost in terms of prediction's precision also ensures greater generalizability, which is crucial when testing data (Poldrack et al., *JAMA Psychiatry*, 2019).

As for the "post-series" results, we do not think this lower model prediction could have created a false difference between ADAD carriers and non-carriers. If anything, it could have prevented us from finding some associations (as suggested by other reviewers), but we do not think it could have created a false positive result.

The lower precision of our model compared to previous brain age studies has now been further acknowledged:

"Nevertheless, it is important to acknowledge that the error provided by our model is higher than in previous studies.^{3,7,16,32,33,65} This might be due to both the use of rs-fMRI data, which is probably noisier and experiences more dynamic changes than structural data, and the inclusion of different datasets to build our model.^{7,66,67} While multisite models are known to have larger predictive errors, they have the advantage of being more generalizable to unseen data (to reduce the overfitting of the models). As a result, the model output needs to be interpreted with caution and should not be interpreted on its own as reflecting some biological or cognitive state. Rather, the model output should be used to compare brain age characteristics between groups from the same site, as detailed below." Page 14-15

- **(Rev.3, Q. 2)** In the method section, did the authors have tried any harmonization protocol, which might allows the analyses of data collected from different sites and therefore increases the sample size substantially. it would be better if the authors can consider how to harmonize the data from different sites? Or maybe you can consider to train the model by CAMCAN data for it is the largest with N>400. In that way, the authors can consider to validate the results in the other datasets, then investigate the difference between the MCI, AD and NC in ADNI and ADAD dataset

Response: We did not apply any specific harmonization tool on our data. However, we ensured that all data were processed the exact same way for all subjects/sites to reduce inter-site disparities. Thus, all scans went through the same pre-processing pipeline using NIAK (as detailed in the Methods section), any region that was not consistently acquired across subjects and sites (*i.e.* regions that were cut on the images) were removed from the connectivity matrices, and we followed previously published methods for mitigating site effects, such as normalizing and binarizing graphs. By doing so, we expected to reduce the variability caused by the different sites.

Following the reviewer's suggestion, we also applied the ComBat harmonization method on the data, but the results suggest that the use of such data harmonization tools are not appropriate here. See Rev.2, Q.5_response for details.

The methodology initially adopted to limit site effects has been further stressed in the revised version of the manuscript.

“PAD deviation from zero should not be interpreted in isolation due to the potential existence of site/cohort effects; thus, we will only interpret group comparisons within cohorts..” Page 10

“To minimize the possibility of site effects, we included data from a variety of cohorts and sites, validated the model on a completely independent validation set (new site) and applied similar processing methods to all data in order minimize between-site effects.” Page 17

“In order to limit site effects, all functional images were processed in our laboratory (by APB) applying the exact same pipeline and processing steps.” Page 24

The model was developed on multiple sites on purpose, with the objective of increasing the generalizability of the model to new/external datasets (see responses to Reviewers 1 and 2 for details, including response to Rev.2, Q. 8). Developing a model on CamCAN only would provide a site/cohort specific prediction and, therefore, is unsuited for the purpose of the present study (see for example Liem et al., *NeuroImage*, 2017 or Orban et al., *Schizophrenia Research*, 2018, for a demonstration of the advantage of training models on multiple datasets).

Of note, we did not include any patients (MCI/AD) in our analyses. ADNI and DIAN participants included were all cognitively normal. Previous studies already investigated the influence of MCI/AD on brain age (with structural data). Here, we aimed at determining whether biological age was affected earlier on in the disease process; thus, we only considered preclinical AD. While addressing the differences between clinical stages is still an interesting perspective, it was beyond the scope of the present paper. Nevertheless, brain age has been explored in ADNI patients and briefly mentioned, for reference, in the manuscript (see response to the next question for details – Rev.3, Q3).

- (Rev.3, Q. 3). If I am right, I think ADNI has more than 100 subjects, why only 44 subjects were included in the present study. In that way, the results in figure 3 will be enriched by the data from ADNI. Also the results can be enriched by the longitude MCI subjects in ADNI. I strongly suggest the authors can consider to add this part with more well fitted model.

Response: This is right. There were more than 44 ADNI participants with rs-fMRI data (i.e. 155 rs-fMRI data in total from ADNI2). Considering the fact that we included only individuals who were cognitively intact (see response to the previous comment), we excluded all MCI/AD patients from this sample (n=106), which substantially reduced the sample size of eligible data for this cohort (n=49). In addition, 5 participants were excluded due to failure to pass quality control (see Methods for details). As a result, only 44 ADNI participants were included in this specific study (see also answer to Rev. 1 Q.3).

This selection process has been clarified in the text.

“We gathered rs-fMRI data collected in 1624 cognitively unimpaired participants from 18 to 94 years old [...]. Considering our focus on the preclinical phase of AD, individuals with mild cognitive impairment (MCI) or AD dementia were excluded from the present study.” Pages 5-6

“The primary goal of ADNI has been to test whether serial MRI, PET, other biological markers, and clinical and neuropsychological assessment can be combined to measure the progression of MCI and early AD. Considering the focus on preclinical AD, MCI and demented patients were not eligible for the present study.” Page 20

Nevertheless, we explored the influence of cognitive impairment (MCI or dementia) to further address the relevance of the model to AD. To do so, we assessed the differences in PAD (predicted age difference), between the 15 ADNI participants from the test set (all being cognitively unimpaired *APOE4* carriers) and the MCI/AD participants. A total of 100 participants either with mild cognitive impairment or dementia passed quality control and were analyzed. As expected, we found a higher PAD in patients with MCI/AD when compared with cognitively unimpaired ($F_{1,112}=2.85$, $p=.05$; one-tailed statistic controlling for chronological age; see Figure 3 - response). Results are similar when using nonparametric tests (but without controlling for age; Mann-Whitney-U=965, $p=.04$). The higher PAD in clinical AD aligns with our previous results (and previous studies on structural brain age) and suggests that our model could detect advanced aging in AD patients. Nevertheless, this result needs to be taken cautiously, considering the very small sample of cognitively unimpaired individuals.

We added this exploratory analysis as a supplement and briefly mentioned the results in the manuscript.

“Considering our focus on the preclinical phase of AD, individuals with mild cognitive impairment (MCI) or AD dementia were excluded from the present study. In supplementary analyses we nevertheless show that the brain age of ADNI individuals with dementia is overestimated when compared to cognitively normal individuals, which is in line with previous literature^{16,32} and support the validity of our predictive model (see Supplementary for details).” Pages 6

Figure 3 – response: Predicted brain age difference in ADNI *APOE4* controls and ADNI patients with mild cognitive impairment (MCI) and dementia (AD dementia).

- (**Rev.3, Q. 4**) The figures look great and are informative, but the text in figure 1 are too small to be legible and need to be increased.

Response: We are sorry for this issue; the figure has been revised accordingly (see next page).

A. Features (i.e. graph metrics) ranked by importance

B. Root mean square error in the different sets as function of the number of features and network architecture

REVIEWER COMMENTS

Reviewer #1 (Remarks to the Author):

The authors have thoroughly and satisfactorily responded to all of my comments. The methods are now more clearly described and several additional analyses have been conducted to corroborate the robustness of the principal study findings and to put them into a broader perspective. Moreover, the introduction and discussion sections now better frame the specific aims and the complementarity of the study in the context of the existing literature on functional brain changes in presymptomatic AD.

Reviewer #3 (Remarks to the Author):

The authors have addressed all my comments well.

Reviewer #4 (Remarks to the Author):

Major strength: The focus on generalisability and careful separation of training, test, and validation sets is a good example of how to approach machine learning for something like this. Good example of collection of multisite data.

Major weaknesses: First, the rs-fMRI based brain aging model as a “base model”: (1) The rationales for using rs-fMRI for this purpose were insufficient; the literature on some well-developed chronological brain aging models was not fully compared. For example, Cole et al. *Molecular Psychiatry*. (2018) – they reached 88% model fit for the same research question, relative to ~50% in the current manuscript; (2) the application of the graph theory to rs-fMRI is still more controversial compared to the application in dMRI-based modality; relevantly, some aspects of method for the graph matrices are problematic (see below). Together, these limitations may explain the relatively weak predictive value of the current brain age model.

Second: the method (dataset, study design, and analysis) is not suitable for determining if the pathology (although genetics may be fine) is a precursor of brain aging.

What are the noteworthy results?

- Graph theory metrics most predictive of brain age: subgraph centrality, clustering/modularity coefficients, smallworldness. Suggests that segregation/integration critical to age-related functional changes. Good example of using explainable approaches to identify knowledge that may be useful in future.

Will the work be of significance to the field and related fields? How does it compare to the established literature? If the work is not original, please provide relevant references.

- Some good examples of explainable machine learning approach: feature ranking, optimizing model to use fewest number of features possible.
- Good example of building generalizability into the algorithmic development, would be even better with more representative samples.
- Good example of collation of multisite data.

Does the work support the conclusions and claims, or is additional evidence needed?

- limited findings.

Are there any flaws in the data analysis, interpretation and conclusions? - Do these prohibit

publication or require revision?

- Idea that functional changes occur early on but do not accelerate as the disease progresses does not seem possible to support using this dataset that did not include MCI or AD. Functional changes are clearly present later on in the disease.
- Unclear why only cognition analysis was controlled for education.

Is the methodology sound? Does the work meet the expected standards in your field?

- Why neural net? Seems unnecessarily complex to start with neural nets. Minimal justification for using graph theory and neural nets; the methods are presented arbitrarily.
- Unclear why 40% used as threshold after scrubbing. Seems like a low number.
- Some more advanced pre-processing tools (e.g. ICA-AROMA) that have shown promise in recent years for motion correction (Parkes et al., 2018) not used. Scrubbing is good but excludes high motion individuals, further limiting generalizability.
- No justification for choice of sparsity in graphs.
- There may be an issue with collinearity of the predictors (graph theory metrics).
- It is unclear whether binarized or weighted matrices were used for all metrics or only some, where applicable.
- Was there a rationale for using a maximum of 2 hidden layers and 10 units?
- Was the model retrained without the two APO4 carriers that were accidentally included initially? This was not clearly resolved.
- Why are SVM and regression tree ensemble used for feature selection and why neural network for model training?

- Is there enough detail provided in the methods for the work to be reproduced?

- ICBM demographics not presented, unclear how similar it was to the other datasets. Only years of education, important to include race/ethnicity at least.
- Unclear exactly how data were normalized across sites. Assume it was just done at the final stage (normalization of graph theory metrics in individuals). This is valid but a more detailed analysis of site effects may reveal differences in final metrics (I see that this is addressed in the response).

Minor points

- Limited justification for functional > structural approach, and for specific focus on graph theory metrics.
- Limited description of graph theory metrics.
- Description of datasets, particularly justification for splitting of dataset, was confusing.
- Pgs 3-5. The logical progression of the introduction could be laid out more clearly. There are several places where it feels as if the authors jump between concepts without clearly linking them. Specifically, it is not consistently clear why/how certain statements are relevant to a particular point. For example, on pg. 3 the authors mentioned that brain age may complement existing measures of brain integrity without mentioning which measures. Next, the authors point out that lifestyle factors impact brain age, then state that they aim to assess the impact of genetic and pathological factors; however, motivation for studying genetic risk factors is discussed in the subsequent paragraph. Related points seem to be scattered throughout the introduction, so the authors may want to consider reorganizing the introduction to clarify the motivation and objectives of the study.
- The methods and results could be written and organized more clearly. For example, the first paragraph of the Brain age model section is really just feature selection. The subsequent paragraphs describe the actual model. The methods could be clarified by including a table that outlines the steps of training, testing, and validation. An example would be Table 2 in A task-invariant cognitive reserve network by Stern et al. (2018).
- The manuscript would greatly benefit from another round of proofreading – a number of grammatical and punctuation errors, awkward sentences, and formatting inconsistencies

Submission of the second revision of the manuscript NCOMMS-20-21021A, by Gonneaud et al., to be considered for publication in Nature Communications.

We are grateful to the reviewer for the insightful comments and suggestions. Based on Reviewer 4 comments, we have now justified that we used resting state fMRI (rs-fMRI) to derive our predictive model because it is more sensitive for detecting early changes in the preclinical phase of Alzheimer's disease (AD) compared to grey matter atrophy. We have also justified that the error of our brain age model is higher compared to some previous studies because we did not apply any post-hoc correction to reduce the error (unlike many of the studies reported in the table below), we included data from 6 different cohorts (including data from multiple sites increases the error but also improves the generalizability of the model at the same time), and we leveraged rs-fMRI instead of grey matter density because it is better suited for the objectives of studying the preclinical phase of AD. We also justified the use of graph metrics; these features simply outperformed the weighted edges extracted from the matrices during cross-validation. For the same reasons, the neural net was chosen because it outperformed simpler competing models (such as Support Vector Machines [SVM] or decision trees). These decisions were not justified *a priori* and were based on the model performance during cross-validation. Finally, we agreed with the reviewer that the grouping justification, the description of the methods, and the general presentation of the paper could have been improved. In this revised version, we have extensively reorganized the manuscript and improved the clarity of the methods (we have added a new Figure [Figure 1] showing the workflow of the study), improved the display of the figures, and revised the general structure of the paper. The manuscript was also edited to improve the language. Overall, this revised manuscript is substantially improved, and we believe that it addresses all reviewer 4 concerns. Below, we provide an extensive point by point response to each query of the reviewer.

Reviewer #4 (Remarks to the Author):

Major strength: The focus on generalisability and careful separation of training, test, and validation sets is a good example of how to approach machine learning for something like this. Good example of collection of multisite data.

Major weaknesses: First, the rs-fMRI based brain aging model as a “base model”: (1) The rationales for using rs-fMRI for this purpose were insufficient; the literature on some well-developed chronological brain aging models was not fully compared. For example, Cole et al. *Molecular Psychiatry*. (2018) – they reached 88% model fit for the same research question, relative to ~50% in the current manuscript; (2) the application of the graph theory to rs-fMRI is still more controversial compared to the application in dMRI-based modality; relevantly, some aspects of method for the graph matrices are problematic (see below). Together, these limitations may explain the relatively weak predictive value of the current brain age model.

Response: We thank the reviewer for the positive comment on the machine learning approach we adopted, which is an important feature of the proposed paper. We made substantial changes to the introduction and discussion to clarify our approach and better justify our methodological choices.

First, we now better justify our choice of using rs-fMRI in the introduction and discussion. It is important to emphasize that brain age models derived from different modalities may convey different information about accelerated brain aging, as the changes in rs-fMRI are only mildly correlated with grey matter atrophy and are detected earlier in the disease progression. Here, we chose to use rs-fMRI instead of cortical atrophy for this conceptual reason. Evidence demonstrates that rs-fMRI may be detected in preclinical phase of the disease, which is not necessarily the case of brain atrophy. Since the scope of our paper is to study the preclinical phase of AD in both DIAN and PREVENT-AD, we chose to derive our brain age model using

rs-fMRI. We believe that this decision was the right approach to tackle our specific research question given the latest evidence from the literature. Below are some examples of these modifications:

“Prior work has shown that brain changes characteristic of an AD process can be demonstrated two or three decades before symptom onset.^{25,26} Typically, this sequence begins with the accumulation of cerebral beta-amyloid (A β), followed by the deposits of hyperphosphorylated tau (neurofibrillary tangles), metabolic brain alterations and other evidence of neurodegeneration that precede cognitive and functional symptoms.^{25,27} Functional brain alterations revealed by MRI measures of resting state connectivity (rsfMRI) become detectable almost synchronously with A β and tau measured by positron emission tomography (PET) and are therefore evident several years before atrophy can be detected by structural MRI.^{28,29} Conjunction of such functional and biological changes appears to extend throughout the development of AD from its pre-clinical to its dementia stages.²⁷ These findings suggest that MRI measures of resting state functional connectivity may be a more sensitive modality than structural imaging for detection of brain changes in pre-clinical AD. introduction page 4-5.

“Variation in notional “biological aging” has been proposed to account for inter-individual differences in the way people age.⁴⁰ Combined with larger and more available datasets, machine learning methods can improve our understanding of brain function and our ability to predict health trajectories from brain properties. Previous models of brain aging have been informed primarily by characteristics of brain structure.⁴¹ Accelerated structural brain aging has been found in individuals with MCI and AD dementia.^{12,14,15} However, functional brain abnormalities are generally detectable prior to structural changes in the AD continuum, the latter being typically more proximate to the expression of clinical symptoms.^{25,42,43} Here we developed a model that could evidently predict brain age across the entire adult human lifespan (ages 18-94). Discussion page 14.

Second, we now justify why the R^2 of our model is lower than what has been reported in previous studies in the literature. There are three main reasons for this: 1) we use rs-fMRI data instead of structural data (for the reasons stated above), 2) we used 6 datasets (which is a main strength of the proposed study that increases the generalizability of our model) and 3) we did not apply any post-hoc age-bias correction that inflates the model performance. Regarding the specific model mentioned by the reviewer (Cole et al. Molecular Psychiatry, 2018), it was based on structural imaging data, the performance was not assessed in an external validation set (the $R^2 = 0.88$ is reported for the cross-validation procedure within the train set; which drastically increases the likelihood of overfitting compared to our procedure), and an age-bias correction was applied in the train set to maximize the prediction and artificially reduce the error (i.e. applying a post-hoc correction that was not naturally learned by the model).

Each of these three methodological points is detailed below.

1) The expected error when using rs-fMRI data instead of structural data: In a subsequent paper by the same group (Cole et al. Neurobiology of Aging, 2020) they predicted age in the UK BioBank using functional data with a R^2 of .19, while we report a R^2 of .53 in the training set (which is as good as the best prediction in Cole’s paper, obtained using dMRI; R^2 of .53). Importantly, our prediction remained at R^2 of .36 in our test set (See Table below for a list of measures reported in key structural and functional brain age predicted papers, actualised since the previous revision). Together, these findings suggest that our model compares well with other models derived from rs-fMRI. We believe that our methodology is robust and that our model performance is good.

2) Using data from multiple sites: Some differences in our model might also be attributed to the fact that we included different cohorts in the training set and added an external validation sample. The vast majority of previous papers used single-site/cohort data and/or validated their model “only” through cross-validation. Multiple datasets are prone to increase the prediction error, but the use of a single dataset

dramatically reduces the applicability of the model to other datasets (Liem et al, NeuroImage, 2017; Orban et al, Schizophrenia Research, 2018). By using only one dataset for training and validation or even testing (as done in several previous studies), it is likely that our prediction would have been better (as suggested by the performance accuracy increase in the training set with more complex models; Figure 2B) but it would have the major pitfall of being cohort-dependent and represent a lack of generalizability (Poldrack et al., JAMA Psychiatry, 2019). Lastly, a somewhat lower accuracy might also present some advantages for clinical group differentiation: recent findings suggest that moderately fitting brain age models provide more clinically-informative information than tightly-fitting models as they allowed a better differentiation of patients with MCI, AD, schizophrenia and depression (Bashyam et al., Brain, 2020). Therefore, our model's lower performance might not necessarily imply a lower sensitivity to detect early changes in preclinical AD.

3) Using post-hoc age-bias correction: Brain age models are known to overestimate younger ages and underestimate older ages (this is discussed in several papers from JH Cole). This is a common feature in brain age models, which is sometimes accounted for by adjusting the fitted regression line to improve the reported predictive value. Notably, this is done in Cole, *Neurobiology of Aging*, 2020 and de Lange et al, *NeuroImage* 2020. Some methodologists have however argued that we should report the actual prediction of the model (which is what we did) before applying any type of correction because it allows to provide an objective measure of model's performance (i.e. what the model can actually learn and predict). Here, we report the model fit without applying an age correction since the development of the model is a stand-alone result. When testing our specific hypotheses in patients from DIAN, PREVENT-AD, and ADNI, we added the chronological age as a covariate when testing group differences for the PAD to account for the natural overestimation of younger ages and underestimation older ages observed in brain age models. Thus, while we agree with Cole and others that the model bias that can influence the results should be accounted for, we disagree that the model itself should be corrected (which naturally decreases the error, at least in the train set). We clarified these important points in the discussion.

Note that the adjustment of the model's regression line was tested in our study (as in Cole 2018). As expected, such manipulation (artificially) increased the model's accuracy in all sets. Results obtained when addressing brain age in preclinical AD while using this procedure were fully similar to those presented in the manuscript in its present form. While the interpretability of the predicted age appears easier with such adjustment, such method has the disadvantage of modifying the model itself. We truly believe that presenting the non-adjusted data is more appropriate (even if the error is higher).

“Compared with structural predictive models, previous modeling approaches using rs-fMRI data have found higher error.7,19,46,47 These observations could partly be attributable to known characteristics of rs-fMRI data. Such data are typically noisier and experience more dynamic changes than structural data, and they may be more sensitive to multi-site effects. Despite these difficulties, we attempted to derive our brain age model from rsfMRI because this modality appears better suited to study of the pre-clinical phase of AD. An extensive literature suggests that connectivity disruption appears early in the course of sAD as well as in “normal” aging.48–53 Moreover, training, validating, and testing our predictive model across multiple cohorts also increased the error of our model compared to previous studies.3,7,11,13,14 Yet, inclusion of data from different sites should logically improve the generalizability of the model, a key strength when the model is applied to new data from different cohorts.7,54 Finally and importantly, brain age models tend to overestimate younger ages and underestimate older ages.19,55 While some researchers apply an age-bias correction procedure to their model,19 we are showing the non-adjusted model and used chronological age as a nuisance variable in our PAD analyses instead of applying this correction prior to the PAD calculation. In sum, while we recognize that the error of our model is higher than most previous brain age models, it was derived from rs-fMRI data, no age bias correction was applied to test the model accuracy, and we suggest that it is more generalizable than previous models. Crucially, it also appears to be sensitive to the questions of interest here.” Discussion page 16-17.

Table – Response: Accuracy of Brain Age models

Study	Modality	Site	Age-range	Age-bias correction	Reported measure(s) of model performance			
					R	R ²	RMSE	MAE
Present manuscript								
Training set	rs-fMRI	Multiple	18-94			.53	14.01	11.00
Validation set		Single	19-79			.49	13.84	11.90
Test set		Multiple	18-90	no		.36	13.24	11.58
Kaufmann et al., Nature Neurosci , 2019	Struct. MRI	Multiple	3–89	no	.93(women) .94 (men)	-	-	-
Wang et al., PNAS , 2019	Struct MRI	Single	~46-96	no	.85	-	-	4.45
Franke et al., NeuroImage , 2010	Struct MRI	Multiple	20-86	no	.92	-	6.28	4.98
Cole et al., NeuroImage , 2017	Struct. MRI	Multiple	10-90	yes	.57-.96 (range)	.32-.92 (range)	6.31-15.10 (range)	4.16-11.8 (range)
Liem et al., NeuroImage , 2017	rs-fMRI	Single + replication	19-82	no	-	.75-.80 (range)	-	5.25-5.99 (range)
	Struct. MRI			no	-	.62-.83 (range)	-	4.83-7.29 (range)
	Multimodal			no	-	.87	-	4.29
Goyal et al., PNAS , 2019	Metabolic PET (glucose, oxygen consumption, cerebral blood flow)	Multisite (2 cohorts with homogenised parameters)	20-82	yes	.89	-	-	~5.4
Amen et al., JAD , 2018	SPECT	Multiple	0-105	no	-	.73	-	-
Dosenbach et al., Science , 2010	rs-fMRI	Single	6-35	no	-	.55	-	-
Lee et al., Proc IEEE Int Symp Biomed Imaging , 2018	rs-fMRI	Single	8-22	no	.36-.61 (range)			2.15-3.02 (range)
Cole et al., Molecular Psychiatry , 2018	Struct. MRI (WM+GM)	Multiple for the training,	18-90	yes	.94	.88	5.01	6.31
		other site for the test	72±1SD	yes	NA	NA	8.85	7.08
Cole et al., Neurob. Aging 2020	Multimodal	Single site	45-84	yes	.79	.62	-	3.52
	Struct. MRI				.68	.47	-	5.93
	dMRI				.73	.53	-	5.26
	rs-fMRI				.44	.19	-	5.26
de Lange, NeuroImage , 2020	Struct. MRI	Single site	60-85	Without (with)	.46	.30	4.49 (2.32)	3.60 (1.73)
	rs-fMRI			Without (with)	.004	.002	5.16 (1.20)	4.18 (0.90)
	rs-fMRI		60-79	no	.06	.004	4.27	3.48

	rs-fMRI	UK biobank	60-79	no	.04	.002	4.69	3.85
--	---------	------------	-------	----	-----	------	------	------

Second: the method (dataset, study design, and analysis) is not suitable for determining if the pathology (although genetics may be fine) is a precursor of brain aging.

Response: It is unclear what the reviewer means when stating that the method is not suitable for determining if the pathology is a precursor of brain aging. We want to emphasize that mutation carriers in DIAN **are certain to develop AD**, making this an exceptional model to study prospectively their accelerated brain aging in the preclinical phase of the disease. We agree that the genetics aspect of the paper is stronger conceptually and methodologically when compared to the pathology, for which we did not have confirmed autopsy, or any possibility to address the causality in these associations. We have now address that in the discussion. Furthermore, our cross-sectional design does not allow us to know if accelerated brain age precedes the pathology or vice versa (which can be overcome when studying the mutation carriers). We have now revised the introduction and discussion of the paper to stress the importance of the individuals with genetic mutation in DIAN. We have also removed the inference that our results suggest that accelerated brain aging precedes AD pathology. The abstract, introduction and discussion also now more closely focus on asymptomatic ADAD mutation carriers and we interpret the results related to A β and sAD with more caution. Figures have also been updated accordingly (see Figure 3 below).

“AD dementia symptoms appear only after massive, evidently irreversible brain changes. Therefore, a more promising approach, at least in theory, is to prevent such changes. However, AD prevention requires improved understanding of the pre-clinical phase of AD.²⁷ Identification of individuals in this clinically “silent” phase of the disease is challenging because it is mostly unknown who will develop dementia during the lifespan. One way to circumvent this problem is the study of autosomal dominant AD (ADAD), a group of rare genetically determined variants of AD caused by mutations in the amyloid precursor protein (APP), presenilin 1 (PSEN1) or presenilin 2 (PSEN2) genes, all involved in A β production.^{22,28} Because these mutations are fully penetrant, progression to disease is predictable, making ADAD an ideal model for the study of the pre-clinical (i.e., pre-symptomatic) phase of AD.”
Introduction page 5.

“Applying our model in the context of AD, we found evidence of accelerated functional brain aging in individuals in the pre-clinical phase of dominantly inherited AD. ADAD is widely believed to be a disease caused by overproduction of A β , and studies of ADAD have shown that biomarkers such as CSF-A β , start changing in mutation carriers as early as 25 years before symptom onset.²² [...] Our rs-fMRI predictive model implied that functional brain age of ADAD pre-symptomatic mutation carriers (DIAN) exceeded their chronological age by about 10 years (based on the findings in non-carriers). This observation alone suggests that the pre-symptomatic phase of ADAD is accompanied by accelerated brain aging. The relative importance of A β burden on accelerated brain aging was less clear. While no association was found between A β burden when restricted to DIAN mutation carriers, the difference between mutation carriers and non-carriers was stronger (i.e. significant only) in the one with fibrillar A β as detected with PET imaging. The observations of accelerated brain aging in carriers may therefore not be entirely attributable to the accumulation of A β . While A β is often hypothesized to be the starting point of the AD neuropathological cascade,⁵⁹ tau is believed to be more toxic^{60,61} and might therefore be more closely associated with accelerated aging. Mutated genes in ADAD could also have life-long effects on the brain that are not fully dependent on A β accumulation. Consistent with this view, a previous study in PSEN1 mutation carriers from the Columbian cohort showed early changes in brain function before evidence of cerebral A β plaque accumulation.⁶² Finally, we cannot exclude the possibility that some A β -negative individuals would in fact be A β accumulators,^{63,64} or present other forms of A β that cannot be detected through PET. What seems to be clear is that AD genetic mutations influence functional brain properties

in pre-clinical ADAD. The exact mechanisms that drive this accelerated brain aging will need further investigation.” Discussion page 17.

“Several limitations should be mentioned. These relate both to the model and to the cohorts used to test our hypotheses. [...] One obvious limitation of inference from the PREVENT-AD data compared to those from DIAN, is that we cannot know which participants will later develop AD dementia. The lack of evidence for accelerated brain aging in PREVENT-AD APOE $\epsilon 4$ carriers (vs. non-carriers) might reflect nothing more than the known fact that not all APOE $\epsilon 4$ carriers will develop AD dementia (i.e., are in the pre-clinical phase of the disease) while some non-carriers will develop the disease. The subsample of PREVENT-AD participants having $A\beta$ pathology in the test set was also relatively small, which could likely limit inference.” Discussion page 19

Figure 3. Predicted age difference in DIAN and PREVENT-AD

A. Density plot of chronological age vs. predicted age in the test set participants in DIAN. **B.** Brain age is overestimated in autosomal dominant mutation carriers compared to non-carriers. The overestimation in mutation carriers is in part due to $A\beta$, with a difference between mutation noncarriers and $A\beta+$ mutation carriers only, and an association between $A\beta$ load and predicted age difference across the whole cohort. **C.** Density plot of chronological age vs. predicted age in the test set participants in PREVENT-AD **D.** In individuals at risk of sporadic Alzheimer’s disease, brain age is overestimated irrespectively of APOE4 genotype. For B and D the interquartile range (25th Percentile, Median and 75th Percentile), the whiskers (lines indicating variability outside the upper and lower quartiles minimum value) and the individual dots are presented.

What are the noteworthy results?

- Graph theory metrics most predictive of brain age: subgraph centrality, clustering/modularity coefficients, smallworldness. Suggests that segregation/integration critical to age-related functional changes. Good example of using explainable approaches to identify knowledge that may be useful in

future.

Response: We thank the reviewer for highlighting the noteworthy of this approach.

Will the work be of significance to the field and related fields? How does it compare to the established literature? If the work is not original, please provide relevant references.

- Some good examples of explainable machine learning approach: feature ranking, optimizing model to use fewest number of features possible.
- Good example of building generalizability into the algorithmic development, would be even better with more representative samples.
- Good example of collation of multisite data.

Response: We thank the reviewer for this very positive feedback. We are however unsure which more representative samples could have been used, given that we leveraged 6 different cohorts.

We used multiple datasets to create and validate our model, some of these datasets including adults across the lifespan. For our test set we got access to the DIAN data, which is an ideal dataset to study the preclinical phase of AD. We also included a large cohort of individuals with a family history of sAD and validated our model's sensitivity in AD individuals with cognitive impairment in ADNI (third independent dataset to test our model). An ideal cohort of preclinical sAD would have included cognitively normal participants that progressed to dementia, unfortunately such data are extremely rare and do not always include PET and fMRI data at the time the participants were cognitively normal. In fact, we only know of the ADNI cohort that has PET data on few individuals progressing from cognitively normal to dementia and most of them do not have fMRI data.

Does the work support the conclusions and claims, or is additional evidence needed?

- limited findings.

Response: We are unsure what the reviewer's comment is specifically referring to. We have a first part of the results on the brain age model (1), and a second on brain age in preclinical AD (2).

1. As pointed out by the reviewer our model is a "good example of how to approach machine learning for something like this. Good example of collection of multisite data." Our model successfully predicted age across the lifespan in a completely independent lifespan cohort (ICBM) and in our test set, which includes multisite participants from the same cohorts as the training set, but who were not included in it.
2. We provide novel insights on functional brain aging in preclinical AD, and now more clearly synthesise the results by mentioning that: "*Our results showed, first, that pre-symptomatic carriers of ADAD mutations (DIAN cohort) had evidence of accelerated functional brain aging. Importantly, this finding was stronger in individuals who already accumulated significant A β pathology as evidenced by PET imaging. In the cohort at elevated risk of sAD (PREVENT-AD cohort), neither APOE ϵ 4 status nor PET evidence of A β pathology was associated with apparent accelerated brain aging but individuals closer to their parental age of onset tended to show accelerated brain aging. Secondary analyses in a third independent cohort including a small subset of individuals diagnosed with either sAD dementia or MCI (ADNI cohort) confirmed the expected acceleration in functional brain aging in patients vs. cognitively normal older adults, suggesting that functional brain age is accelerated in cognitively impaired individuals with sAD and therefore validating the sensitivity of our model to sporadic AD-related processes. We conclude that asymptomatic persons with strong genetic determinants show a characteristic pattern of functional brain changes that are associated with accelerated biological brain aging. The biological development of AD is therefore characterized by a*

pattern of advanced brain aging that can be detected prior to symptom onset, at least.” Introduction page 7

It is important to stress the fact that we provide evidence of accelerated functional brain aging in **three independent cohorts, the two forms of the disease** and, **not only on clinical, but mainly in preclinical AD** (i.e. between mutation carriers and non-carriers in DIAN, it was associated with expected years to symptom onset in PREVENT-AD, and between cognitively normal and cognitively impaired in ADNI).

Are there any flaws in the data analysis, interpretation and conclusions? Do these prohibit publication or require revision?

- Idea that functional changes occur early on but do not accelerate as the disease progresses does not seem possible to support using this dataset that did not include MCI or AD. Functional changes are clearly present later on in the disease.

Response: We agree with the reviewer and have removed most of the paragraph in which we discussed accelerated vs advanced aging (which was actually suggested by another reviewer in a previous round of revision). Furthermore, as a sensitivity analysis to validate our brain age model in later stage of the disease, we tested if MCI and AD patients from ADNI have a higher PAD when compared to ADNI Controls. We had a total of 100 MCI/AD participants (after quality control) that we compared to 15 Controls ADNI already included in our test set. The majority of ADNI Controls had been included in the training set, explaining the small sample in the test set. As expected, we found a higher PAD in patients with MCI/AD when compared with cognitively unimpaired ($F_{1,112}=2.85$, $p<.05$; one-tailed statistic controlling for chronological age). This result suggests that our model can detect advanced aging in AD patients based on functional changes, as pointed out by the reviewer. We have included this information in the main manuscript (it was previously in the supplementary material) and have changed the term “advanced brain aging” by “accelerated brain aging” which is more commonly used in the literature.

“Secondary analyses in a third independent cohort including a small subset of individuals diagnosed with either sAD dementia or MCI (ADNI cohort) confirmed the expected acceleration in functional brain aging in patients vs. cognitively normal older adults, suggesting that functional brain age is accelerated in cognitively impaired individuals with sAD and therefore validating the sensitivity of our model to sporadic AD-related processes.” Introduction page 7

“In secondary analyses we nevertheless tested whether cognitively impaired individuals with sAD evidenced accelerated brain aging using our functional predictive model that was built solely on cognitively unimpaired individuals” Method page 8.

“Finally, we performed additional post hoc analyses to test whether sAD symptomatic individuals (MCI and dementia) had a higher PAD than asymptomatic individuals at risk of sAD (APOE $\epsilon 4$ carriers). This analysis was not initially planned, and was conducted only in a small subsample of the ADNI dataset (15 asymptomatic APOE $\epsilon 4$ carriers from the test set and 100 symptomatic individuals). The findings do suggest, as expected, increased PAD among individuals with cognitive impairment as compared with asymptomatic individuals at risk of sAD (using parametric, $F_{1,112}=2.85$, $p=.047$, or non-parametric Mann-Whitney- $U=965$, $p=.04$, one-tailed test).” Result page 12

“While our focus was the preclinical phase of the disease, we performed post-hoc analyses using rs-fMRI data from a small subset of ADNI patients. We found accelerated functional aging in persons with symptomatic sAD (MCI or dementia) when compared with others who were asymptomatic, but at increased risk of sAD (APOE $\epsilon 4$ participants from our test set). These additional analyses suggest

accelerated functional brain aging in individuals with clinical sAD and further confirm the validity of our brain age model” Discussion page 19.

- Unclear why only cognition analysis was controlled for education.

Response: We have now removed the analyses on cognition from the revised version since we believe these analyses were out of scope given that the focus of the paper is on cognitively normal individuals. As mentioned previously, we have now included the difference between cognitively normal and cognitively impaired individuals in ADNI as a core finding, which we believe is a more appropriate way to assess if the brain age model captures cognitive impairments.

Is the methodology sound? Does the work meet the expected standards in your field?

- Why neural net? Seems unnecessarily complex to start with neural nets. Minimal justification for using graph theory and neural nets; the methods are presented arbitrarily.

Response: We agree with the reviewer that the model’s rationale, the results and methods sections remained confusing. We have now clarified that graph metrics were chosen because they outperformed models trained directly on the weighted edges of the matrices. Similarly, neural net was used because it outperformed simpler models. For instance, SVM and tree ensemble yielded a rmse of 16.45 and 16.08, respectively, while neural nets improved the model performance (rmse of 14 and 13.24 in the training and test sets, respectively). There was no rationale or justification *per se* to use these specific features or models; the decision was based on how well these models performed in the validation set (which is an independent cohort of participants, from a different site, specifically used to assess the performance prior to testing the PAD in the ‘real’ test set).

Regarding the graph metrics, we agree that we should have better justified our choice. One main advantage of this approach was to reduce the number of inputs into our final model. In our initial analyses (not mentioned in the manuscript), we modeled brain age with inter-regional connectivity values with relative success, but they were difficult to interpret in that connections could be chosen randomly with little influence on performance. The advantage of graph metrics is that every connection is defined by the other connections, which could lead to easier interrogation. Further, we should mention that the age bias mentioned in the first comment (over-estimation of the age of younger adults and under-estimation of the age of older adults) was greater with inter-regional connectivity than with the graph metrics, which also motivated our choice of using the neural network. Overall, we agree with the reviewer and we clarified how our decisions were made.

Part of this information can now be found in the introduction, results and discussion.

“First, to reduce the number of inputs of the model, we searched for graph metrics that most reliably predicted chronological age.10 To do so, training set data was entered in parallel in support vector machine (SVM) and regression tree ensemble models to identify graph metrics with highest weights. The root mean squared error (rmse) for predicted chronological age in SVM and the tree ensemble were 16.45 and 16.08, respectively. [...] We chose the optimal neural net architecture after having built different neural networks with increasing complexity, varying in number of input features (5, 10, 15, 20, or 25 most-important graph metrics, ranked as described previously), hidden layers, and hidden layer units. Importantly, each graph metric was only entered once as input for each neural network architecture tested, and the inputs were kept constant across the model’s iterations, such that features of more complex models always included the features of the simpler ones” Result section page 9

“Of note, the neural net model outperformed the simpler models used in our feature ranking step (rmse = 16.45 for SVM and 16.45 for tree ensemble, see above).” Result section page 11

“To assess information integration in the brain, we relied on global brain function while applying graph metrics.^{6,36} This approach provides a holistic view of brain function that has been shown previously to change through aging and AD.⁴⁴ Graph theory has the advantage that it quantifies and simplifies the many “moving parts” of dynamic systems inasmuch as every connection is defined by its relation to all others. We also used feature selection as an intermediate step to simplify the final model. We suggest that our approach using graph theory and feature selection are steps in the right direction towards interpretability of complex models. We are encouraged that the 10 graph metrics suggested as most important by these algorithms provided much lower error in our final neural network model in comparison to random choice of graph metrics. Of note, models using individual functional connections as inputs are also possible, but such models have been shown to require multiple dozens⁴⁵ or hundreds¹⁰ of functional connections whose inter-relationships are not defined.” Results page 16

- Unclear why 40% used as threshold after scrubbing. Seems like a low number.
- Some more advanced pre-processing tools (e.g. ICA-AROMA) that have shown promise in recent years for motion correction (Parkes et al., 2018) not used. Scrubbing is good but excludes high motion individuals, further limiting generalizability.

Response: We agree with the reviewer that some preprocessing steps could have been done differently or using different tools. Our threshold was held consistent with our pre-processing pipeline used in previous publications of high technical quality (Orban et al. Sci Data 2015; Vogel et al. Brain 2018; Köbe et al. NeuroImage 2021). By doing so, we kept individuals with relatively high motion (since a few had up to 60% of their frames with high motion), with the aim of retaining as many participants as possible. We should also note that, despite this threshold, more than 90.5% of data was retained across all participants (see Table below now in Supplementary material). We were also very careful by ensuring to remove the effect of age that was merely due to motion (there was a correlation between age and frame displacement) before developing the model, using the mean regression technique as described in Geerligts et al, *Human Brain Mapping*, 2017 and on page 28 of the manuscript. “Motion-related noise was further mitigated using the mean regression (‘MR’) technique as outlined previously.⁸⁴ Briefly, the average of all correlation values within the upper diagonal of the correlation matrix was calculated for each subject in the training data. A linear fit between these across-subject average values and the across-subject value at each element of the correlation matrix was generated, creating a slope and intercept term associated with each element of the matrix. The final value used in each element of the correlation matrix was equal to the residual between the MR-model fit and the original correlation value. Importantly, the MR model was created with only the training data.”

The average percentage of frames retained in each cohort is reported in the table below and had been added in Supplementary material.

Supplementary Table 2. Percentage of frames retained from resting-state fMRI scans in each cohort

Cohort	Average % frames retained \pm std
CamCan	86.2 \pm 15.5
FCP-Cambridge	100 \pm 0
DIAN	93.8 \pm 12.0
Prevent-AD	85.0 \pm 17.2
ADNI	80.7 \pm 15.4
ICBM	96.76 \pm 8.0

- **No justification for choice of sparsity in graphs.**

- **It is unclear whether binarized or weighted matrices were used for all metrics or only some, where applicable.**

Response: We chose 5% of sparsity in graphs because it has been shown across multiple studies that link densities between 1.5% and 10% give the most biologically plausible networks, and 5% is roughly midway through this range (Mansour et al., *Sci Rep*, 2016; Cole et al., *Nat Neurosci*, 2013, Power et al., *Neuron*, 2011, 2013). Whenever a metric could be generated with either binarized or weighted matrices, both metrics were calculated, to have all metrics possible available as input.

Only 5 out of the 26 metrics used binarized matrices and out of those 5, only one was retained in the final model (*i.e.* weighted modularity coefficient). As such, we believe the sparsity threshold is not likely to have biased our model in major ways.

This information can now be found in the method section page 29: “*In the case of unweighted metrics, correlation matrices were thresholded at 5% link density, which ensured only the top 5% strongest correlation values were counted as connections in the matrix.*⁸⁵ Only 5 out of the 26 metrics used binarized matrices and out of those 5, only one was retained in the final model (*i.e.* weighted modularity coefficient).”

- **There may be an issue with collinearity of the predictors (graph theory metrics).**

Response: Multicollinearity of variables certainly can be an issue in determining variable importance. However, our feature ranking strategy for determining inputs into our neural network exhibited higher performance than choosing features randomly, suggesting our feature ranks were not heavily influenced by multicollinearity. Additionally, feature importance for both SVM and the tree ensemble were mostly in agreement, and it’s unlikely that multicollinearity would have had equal influence on these two very different algorithms. We added the correlation matrix of the 10 input features in supplementary material, copied below.

“*For reference, the covariance matrix of the 10 selected graph metrics is presented as supplementary material (eFigure 1).*” Result page 10

eFigure 1: Correlation between the 10 graph metrics used as input in the neural network

We also added a paragraph in the discussion section covering the issue of multicollinearity:

“Second, we also cannot exclude the possible influence of collinearity when determining the age predictive graph metrics. The SVM and the tree ensemble were however mostly in agreement, and it’s unlikely that multicollinearity would have had equal influence on these two very different algorithms.”

Discussion page 19

- Why are SVM and regression tree ensemble used for feature selection and why neural network for model training?
- Was there a rationale for using a maximum of 2 hidden layers and 10 units?

Response: Neural network was chosen as the final predictive model given that it outperformed the simpler model (as detailed in a previous comment: the root mean squared errors (rmse) for predicted chronological age in SVM and the tree ensemble were 16.45 and 16.08, respectively, while neural net provided a rmse of 14, or even below for more complex models).

There was no a priori selection of the final neural network architecture, we ran multiple neural network architectures with the same input features in both our training set (multi-site lifespan dataset) and our validation site (external site lifespan dataset, ICBM). As our final model, we chose the one that had the lower error (based on rmse) in the validation set, with the idea that this would be the most generalizable model, and this led to a model with 2 hidden layers of 5 and 2 units, respectively. Furthermore, while even more layers and units could have been tested, our validation step suggests that improving the neural net complexity would have been associated with a better prediction in the training set to the detriment of a worst prediction in the validation set (i.e. overfitting with higher number of units per layer). We should also note that the feature selection to rank graph metrics from the most to least related to age in simpler

models was useful, given that it outperformed the null models (where graph metrics inputs are chosen randomly).

We clarified this point in the result section *“In a second step, we aimed at creating an accurate model requiring the fewest number of features possible. We used training data to generate a neural net model and assessed its accuracy using the validation set. [...] Network models had 5 to 25 input features in increments of 5, entered according to their importance, as determined previously (see above). [...]. Each graph metric was only entered once as input for each neural network architecture tested, and the inputs were kept constant across the model’s iterations; features of more complex models always including the features of the simpler ones. Architecture of the network was also tested with various number of hidden layers (1 or 2) and number of units in the hidden layers (2, 5, 7, or 10). Age was modeled on the training data using the fitnet function with Bayesian regularization backpropagation. Model accuracy was ultimately determined by the root mean squared error (rmse) between actual and predicted age on the validation data, with lower rmse reflecting higher accuracy. Because neural network units are initialized with random values, the rmse changed slightly each time model error was measured. Thus, the best model was determined by the lowest rmse, averaged over three iterations. Once the most accurate validated model was determined, it was applied on unseen data (test set).”*

- Was the model retrained without the two APO4 carriers that were accidentally included initially? This was not clearly resolved.

Response: The model was not retrained to exclude the 2 APOE4 carriers from the PREVENT-AD cohort that should have been kept for the test set. This mis-inclusion was realized in the course of the analyses on preclinical AD, when the model was determined and applied to the test set already. We did not want to bias our results by re-tweaking the model once tested in the test set (tested only once as the validation set was used to assess model performance outside the train set). While this might be a source of some small errors in the model (we found no effect of APOE4 on PAD in the PREVENT-AD), this procedure is a major strength of this model, built in accordance with best practices in machine learning.

We now clarified this in the limitation page 19 *“Several limitations should be mentioned. These relate both to the model and to the cohorts used to test our hypotheses. First, our choice not to update or “tweak” the model after it was used to test our hypotheses (a main strength of our approach) left us with a few small errors when constructing the model (e.g., two PREVENT-AD APOE ε4 carriers were included in the training set). While these oversights were unlikely to have affected the final results (APOE ε4 carriers from other cohorts without genotype data were presumably included in the training set), they nevertheless pose a small threat to the integrity of the model.”*

- Is there enough detail provided in the methods for the work to be reproduced?

ICBM demographics not presented, unclear how similar it was to the other datasets. Only years of education, important to include race/ethnicity at least.

Response: We added the female/male ratio in Table 1 (see below), where the different datasets used are described. Education, APOE status, MMSE/MoCa and the Aβ status were only available/requested for DIAN and PREVENT-AD and they are shown in Table 2. We also added ethnicity in the supplementary material whenever available. Unfortunately, the ICBM shared minimal information, we only had access to the age and sex information. We note however that all participants came from the Montreal area, the vast majority probably being Caucasians. We added a sentence in the limitation section to acknowledge the lack of diversity in the sample (page 20). *“Also, while we made great efforts to increase the generalizability of our predictive model, most of the participants included in this study were Caucasian*

(see Supplementary material), stressing the need to increase diversity in both lifespan and AD cohorts.”

Table 1. Dataset characteristics

Cohorts		Training set	Validation set	Test set
DIAN non-carriers	N	105		29
	Age [range]	38.70 ±11.41 [19-69]	-	38.90 ±11.55 [20-58]
	Sex ratio F/M [F]	57/48 [54%]		18/11 [62%]
DIAN carriers	N		-	125
	Age [range]			34.33 ±9.66 [18-61]
	Sex ratio F/M [F]			68/57 [54%]
PREVENT-AD	N	36		256
	Age [range]	63.5 ±5.08 [55-78]	-	63.51 ±5.37 [55-84]
	Sex ratio F/M [F]	25/11 [69%]		189/67 [74%]
CamCAN	N	408		96
	Age [range]	51.12±18.27 [18-87]	-	55.80±19.30 [18-88]
	Sex ratio F/M [F]	208/200 [51%]		40/56 [42%]
ADNI	N	29		15
	Age [range]	76.41 ±6.60 [66-94]	-	72.73 ±6.70 [65-90]
	Sex ratio F/M [F]	17/12 [59%]		10/5 [67%]
FCP-Cambridge	N	195		
	Age [range]	21.04 ±2.33 [18-30]	-	
	Sex ratio F/M [F]	122/73 [63%]		
ICBM	N		46	
	Age [range]		42.28 ±19.31 [19-79]	
	Sex ratio F/M [F]		29/17 [63]	
Total sample	N	773	46	521
	Age [range]	43.37 ±20.45 [18-94]	42.28 ±19.31 [19-79]	54.49 ±16.25 [18-90]
	Sex ratio F/M [F]	429/344 [55%]	29/17 [63%]	325/196 [62%]

Demographic information by cohort and set. *F* Female, *M* male, *DIAN* Dominantly Inherited Alzheimer Network, *PREVENT-AD* Pre-symptomatic Evaluation of Experimental or Novel Treatments for Alzheimer’s Disease cohort, *CamCAN* Cambridge Centre for Ageing and Neuroscience, *FCP-Cambridge* 1000-Functional Connectomes Project – Cambridge site, *ADNI* Alzheimer’s Disease Neuroimaging Initiative, *ICBM* the International Consortium for Brain Mapping.

Table 2. DIAN and PREVENT-AD test set characteristics

	DIAN mutation non-carriers	DIAN mutation carriers	PREVENT-AD
N	29	125	256
Chronological Age (years; mean ± SD)	38.90 (±11.55)	34.33 (±9.66)	63.51 (±5.37)
Sex Ratio F/M (%)	18/11 (62%/38%)	68/57 (54%/46%)	189/67 (74%/26%)
Education (years; mean ± SD)	14.41 (2.13)	14.89 (±3.10)	15.65 (±3.51)
EYO (years; mean ± SD) ¹	-8.56 (±10.85)	-14.18 (±8.94)	-10.42 (±7.21)
APOE4 carriers (%) ²	11 (38%)	36 (29%)	108 (42%)
Aβ-positive (%) ³	0 (0%)	39 (34%)	14 (22%)
MMSE or MoCA (mean ± SD) ⁴	29.36±1.03	29.02±1.27	28.11±1.52

¹ EYO in DIAN was calculated based on the family mutation-specific age at onset. In some case the family mutation-specific age was unknown, the parental age at onset was used to calculate EYO. EYO of PREVENT-AD was calculated only for individuals with a parental history of AD, data was available for 241 participants.

² APOE genotyping was missing for 1 PREVENT-AD participants

³ Aβ-PET data was missing for 2 mutation non-carriers and 11 mutation carriers in DIAN and 192 PREVENT-AD participants

⁴ Global cognitive functioning was assessed using MMSE in DIAN and MoCA in PREVENT-AD

EYO Estimated Years to Symptom Onset, MMSE Mini-Mental State Examination,

MoCA Montreal Cognitive Assessment, SD standard deviation

Supplementary material e-Methods

“Race/ethnicity from the different cohorts

DIAN

The sample mainly identified as non-Hispanic/White, both for mutation non-carriers (90% from the training set and 100% from the test set) and mutation carriers (80% from the test set).

The ten remaining mutation non-carriers (all from the training set) identified as Hispanic/White (n=4), Hispanic with no further specification (n=2), non-Hispanic/Middle Eastern (n=1) or non-Hispanic/Aboriginal (n=2). Regarding the 24 mutation carriers with a different race/ethnicity, they identified themselves as Hispanic/White (n=10), Hispanic with no further specification (n=6), non-Hispanic/Middle Eastern-North Africa (n=2), Aboriginal (n=1), native Hawaiian or other pacific islanders (n=3), Hispanic/Black or African American (n=1) and non-Hispanic/Asian (n=1).

PREVENT-AD

The sample was mainly white/Caucasian with the exception of 4 participants (2 Hispanics, 1 Haitian and 1 unspecified).

ADNI

The sample mainly identified as non-Hispanic/White (83% of those included in the training set and 93% of those included in the test set), with the exception of 6 subjects (1 Hispanic/White, 1 unknown ethnicity/White and 3 non-Hispanic/not White [1 Black, 1 with more than one race and 1 unknown] in the training set, and 1 Hispanic/white in the test set).

FCP-Cambridge, CamCAN and ICBM

Participants from the FCP-Cambridge were recruited from the Cambridge (MA, USA) area, CamCAN is a population-based cohort recruited within the Cambridge City (UK) area (excluding term-time residents of colleges and universities) and the ICBM cohort was recruited in the Montreal (QC, Canada) area; however further demographic information, including specific information on race/ethnicity, were not provided for these cohorts.

- Unclear exactly how data were normalized across sites. Assume it was just done at the final stage (normalization of graph theory metrics in individuals). This is valid but a more detailed analysis of site effects may reveal differences in final metrics (I see that this is addressed in the response).

Response: Indeed, this was a question raised by previous reviewers. The reviewer is right, the graph metrics were normalized across individuals. Still, following the previous reviewers' suggestions, we applied the ComBat harmonization method on the data (<https://github.com/Jfortin1/ComBatHarmonization/tree/master/Matlab>) to investigate how it could affect the results. ComBat uses empirical Bayes estimation to reduce differences between sites/batches (Fortin et al., *NeuroImage*, 2018; Yu et al., *HBM*, 2018; Pomponio et al., *NeuroImage*, 2020).

We first applied ComBat to the 26 graph metrics used to derive our brain age model by removing the site effects. We then applied the same feature selection procedure as in the main manuscript (i.e. ranking

features based on the best age prediction using ensemble tree and SVM) using the harmonized graph metrics as input. The figure below shows the age prediction from SVM using the original metrics (panel A) vs. the harmonized metrics (panel B). We can clearly see that the harmonized metrics yield a worse age prediction. The reason is because the site or cohort effects are inherently related to the different age composition of the cohorts (e.g. young adults only, lifespan dataset, older adults only), and thus harmonizing by sites blurred the age prediction by correcting for the sites age difference.

These analyses suggest that harmonization of the model inputs across cohorts would not be appropriate for the current study and we believe that the steps we used to mitigate site effects while processing the data were sufficient and more appropriate. Importantly, as described above, evidence suggests that training, validating, and testing using heterogenous data collected from several sites actually drastically increases the generalizability of the predictions (see Orban et al., *Schizophrenia Research*, 2018).

Figure Revision-2. Age prediction from support vector machine models using original graph metrics as input (A) and age prediction from support vector machine models using harmonized graph metrics from ComBat as input. (B)

We have now included this figure as e-Figure 2 and have added the following text in the method section: *“To minimize such possible site effects, we drew on data from a variety of cohorts and sites, validated the model on a completely independent validation set (new site) and applied similar processing methods to all data. No further harmonization procedure was applied. The sites effects are inherently related to the different age composition of the site (or cohort, see Figure 1), and thus harmonizing by sites would have removed the age difference between participants (see Supplementary e-Figure 2 for an example of sites correction using ComBat; <https://github.com/Jfortin1/ComBatHarmonization>).”* Page 20.

Minor points

- **Limited justification for functional > structural approach, and for specific focus on graph theory metrics.**
- **Limited description of graph theory metrics.**

Response: These points have been addressed above. Briefly, the introduction and discussion have been completely revised and the rationale for using rs-fMRI to target the preclinical phase of AD is more clearly justified.

The 26 graph metrics were chosen based on their ability to quantify whole-brain connectivity. Building models using individual functional connections as inputs would also have been possible, but such models require multiple dozens or hundreds of functional connections whose inter-relationships are not defined.

“Graph theory has the advantage that it quantifies and simplifies the many “moving parts” of dynamic systems inasmuch as every connection is defined by its relation to all others.” Page 16.

- Description of datasets, particularly justification for splitting of dataset, was confusing.

Response: We apologize for the confusion and further clarified this aspect in the first part of the Results section, page 8, and in a new figure (Figure 1): *“After processing and quality control, 1340 cognitively unimpaired individuals remained for the analyses. These were divided into a training set of 773 persons (large multi-cohorts dataset covering the lifespan used to build the predictive models), a validation (independent lifespan dataset of 46 persons from ICBM used to test the generalizability of the developed models and select the final model), and one multi-cohort test set (125 DIAN mutation carriers and 29 without a mutation, 256 PREVENT-AD individuals thought to be at enhanced genetic risk of sAD, 96 from CamCAN, and 15 cognitively normal individuals from the ADNI.)”*

- Pgs 3-5. The logical progression of the introduction could be laid out more clearly. There are several places where it feels as if the authors jump between concepts without clearly linking them. Specifically, it is not consistently clear why/how certain statements are relevant to a particular point. For example, on pg. 3 the authors mentioned that brain age may complement existing measures of brain integrity without mentioning which measures. Next, the authors point out that lifestyle factors impact brain age, then state that they aim to assess the impact of genetic and pathological factors; however, motivation for studying genetic risk factors is discussed in the subsequent paragraph. Related points seem to be scattered throughout the introduction, so the authors may want to consider reorganizing the introduction to clarify the motivation and objectives of the study.

Response: We are thankful for the feedback and we agree. We have completely reorganized the introduction so that it aligns more with the overall goal of the study. This reorganization should have overcome the lack of logical progression of the introduction. Please refer to the Introduction of the revised manuscript (pages 4-7).

- The methods and results could be written and organized more clearly. For example, the first paragraph of the Brain age model section is really just feature selection. The subsequent paragraphs describe the actual model. The methods could be clarified by including a table that outlines the steps of training, testing, and validation. An example would be Table 2 in A task-invariant cognitive reserve network by Stern et al. (2018).

Response: The methods and results section have also been completely revised. We now include a Figure (now Figure 1, see also below) that outlines the different steps applied to build the model and would like to thank the reviewer for this great suggestion.

Figure 1. Methodology overview

A. Multiple cohorts covering the lifespan were included in the study. They were separated into a training and validation set, both used to develop the predictive brain age model, and a test set in which our model was applied. **B.** All participants underwent resting-state functional magnetic resonance imaging that was processed with a uniform pipeline. Functional connectivity matrices were generated from the Power atlas, from which graph metrics were calculated. Graph metrics were the input in our brain age model, and thus all possible metrics were of interest. **C.** The first step towards building the model was to rank the different graph metrics from the most to least related to aging in our training set, to determine an order of importance to our model inputs using both support vector machine and regression three ensemble algorithms. Neural networks were then tested to identify the best brain age model. Different architectures were tested, and the model applied in the training set that best generalized to the validation set was chosen as the final model (see Figure 2). **D.** The model was applied to the left-out test set and our measure of interest was the Predicted Age difference (PAD).

- The manuscript would greatly benefit from another round of proofreading – a number of grammatical and punctuation errors, awkward sentences, and formatting inconsistencies

Response: We paid close attention to the manuscript to correct all errors. We also simplified a lot of sentences to make them more straightforward. All the manuscript was carefully revised by a native English speaker.

REVIEWER COMMENTS

Reviewer #4 (Remarks to the Author):

no comment